



# Increase in ocean acidity variability and extremes under increasing atmospheric $CO_2$

**Friedrich A. Burger**[1,2], **Jasmin G. John**[3], **and Thomas L. Frölicher**[1,2]

[1]Climate and Environmental Physics, Physics Institute, University of Bern, Bern, Switzerland
[2]Oeschger Centre for Climate Change Research, University of Bern, Bern, Switzerland
[3]NOAA/Geophysical Fluid Dynamics Laboratory, Princeton, NJ, USA

**Correspondence:** Friedrich A. Burger (friedrich.burger@climate.unibe.ch)

**Abstract.** Ocean acidity extreme events are short-term periods of relatively high $[H^+]$ concentrations. The uptake of anthropogenic $CO_2$ emissions by the ocean is expected to lead to more frequent and intense ocean acidity extreme events, not only due to changes in the long-term mean but also due to changes in short-term variability. Here, we use daily mean output from a five-member ensemble simulation of a comprehensive Earth system model under low- and high-$CO_2$-emission scenarios to quantify historical and future changes in ocean acidity extreme events. When defining extremes relative to a fixed preindustrial baseline, the projected increase in mean $[H^+]$ causes the entire surface ocean to reach a near-permanent acidity extreme state by 2030 under both the low- and high-$CO_2$-emission scenarios. When defining extremes relative to a shifting baseline (i.e., neglecting the changes in mean $[H^+]$), ocean acidity extremes are also projected to increase because of the simulated increase in $[H^+]$ variability; e.g., the number of days with extremely high surface $[H^+]$ conditions is projected to increase by a factor of 14 by the end of the 21st century under the high-$CO_2$-emission scenario relative to preindustrial levels. Furthermore, the duration of individual extreme events is projected to triple, and the maximal intensity and the volume extent in the upper 200 m are projected to quintuple. Similar changes are projected in the thermocline. Under the low-emission scenario, the increases in ocean acidity extreme-event characteristics are substantially reduced. CE1 At the surface, the increases in $[H^+]$ variability are mainly driven by increases in $[H^+]$ seasonality, whereas changes in thermocline $[H^+]$ variability are more influenced by interannual variability. Increases in $[H^+]$ variability arise predominantly from increases in the sensitivity of $[H^+]$ to variations in its drivers (i.e., carbon, alkalinity, and temperature) due to the increase in oceanic anthropogenic carbon. The projected increase in $[H^+]$ variability and extremes may enhance the risk of detrimental impacts on marine organisms, especially for those that are adapted to a more stable environment.

# 1 Introduction

Since the beginning of the industrial revolution, the ocean has absorbed about a quarter of the carbon dioxide ($CO_2$) released by human activities through burning fossil fuel and altering land use (Friedlingstein et al., 2019). Oceanic uptake of anthropogenic $CO_2$ slows global warming by reducing atmospheric $CO_2$ but also leads to major changes in the chemical composition of seawater through acidification (Gattuso and Buddemeier, 2000; Caldeira and Wickett, 2003; Orr et al., 2005; Doney et al., 2009). When $CO_2$ dissolves in seawater, it forms carbonic acid that dissociates into bicarbonate ($[HCO_3^-]$), releasing hydrogen ions ($[H^+]$) and thereby reducing pH ($pH = -\log([H^+])$). The rise in $[H^+]$ is partially buffered by the conversion of carbonate ions ($[CO_3^{2-}]$) to $[HCO_3^-]$. The associated decline in $[CO_3^{2-}]$ reduces the calcium carbonate saturation state $\Omega = [Ca^{2+}][CO_3^{2-}]/\left([Ca^{2+}][CO_3^{2-}]\right)_{sat}$, i.e., the product of calcium and carbonate ion concentrations relative to the product at saturation. Undersaturated waters with $\Omega < 1$ are corrosive for calcium carbonate minerals. Each type of calcium carbonate mineral has its individual saturation state $\Omega$ due to different solubilities, e.g, $\Omega_C$ for calcite and $\Omega_A$ for arag-

onite. Over the last four decades the surface ocean pH has declined by about 0.02 pH units per decade (Bindoff et al., 2020). Continued $CO_2$ uptake by the ocean will further exacerbate ocean acidification in the near future (Caldeira and Wickett, 2003; Orr et al., 2005; Bindoff et al., 2020; Terhaar et al., 2020), with potentially major consequences for marine life (Doney et al., 2009) and ocean biogeochemical cycling (Gehlen et al., 2012).

Superimposed onto the long-term decadal- to centennial-scale ocean acidification trend are short-term extreme events on daily to monthly timescales during which ocean pH and $\Omega$ are much lower than usual (Hofmann et al., 2011; Joint et al., 2011; Hauri et al., 2013). These events can be driven by different processes, such as ocean mixing, biological production and remineralization, mineral dissolution, temperature and air–sea gas exchange variations, or a combination thereof (Lauvset et al., 2020). In eastern boundary upwelling systems, for example, short-term upwelling events and mesoscale processes can lead to low-surface-pH events and to short-term shoaling of the saturation horizon (i.e., the depth between the supersaturated upper ocean and the undersaturated deep ocean; Feely et al., 2008; Leinweber and Gruber, 2013). Ocean pH can also rapidly change as a consequence of microbial activity (Joint et al., 2011). Phytoplankton blooms and accompanying respiration drastically increase the partial pressure of $CO_2$ ($p$$CO_2$) and reduce pH in the thermocline (Sarmiento and Gruber, 2006). Such extreme events may have pH levels that are much lower than the mean pH conditions projected for the near future (Hofmann et al., 2011).

Most of the scientific literature on ocean acidification has focused on gradual changes in the mean state in ocean chemistry (Orr et al., 2005; Bopp et al., 2013; Frölicher et al., 2016; Terhaar et al., 2019b). However, to understand the full consequences of ocean acidification on marine organisms and ecosystem services, it is also necessary to understand how variability and extremes in ocean acidity change under increasing atmospheric $CO_2$ (Kroeker et al., 2020). The ability of marine organisms and ecosystems to adapt to ocean acidification may depend on whether the species have evolved in a chemically stable or a highly variable environment (Rivest et al., 2017; Cornwall et al., 2020). Furthermore, if the frequency and intensity of short-term extreme events strongly increase, in addition to the long-term acidification, some organisms may have difficulties in adapting, especially if key $CO_2$ system variables cross some critical thresholds, e.g., from calcium carbonate supersaturation to undersaturation. Key plankton species such as coccolithophores (Riebesell et al., 2000), foraminifera, and pteropods (Bednaršek et al., 2012) were found to be adversely affected by low carbonate ion concentrations. After only several days of being exposed to waters which are undersaturated with respect to aragonite, some species such as pteropods already show reduced calcification, growth, and survival rates (Kroeker et al., 2013; Bednaršek et al., 2014).

Carbonate system variability also plays a role in shaping the diversity and biomass of benthic communities (Hall-Spencer et al., 2008; Kroeker et al., 2011). In laboratory experiments in which deep-water corals are exposed to low-pH waters for a week, some corals exhibit reduced calcification, while recovery may be possible when the low-pH condition persists for several months, stressing the importance of high-frequency variability and short-term acidification events (Form and Riebesell, 2012). There is also growing evidence that the organism response to variability in ocean acidity could change with ocean acidification (Britton et al., 2016). Therefore, understanding the temporal variability of ocean carbonate chemistry and how that will change is important for understanding the impacts of ocean acidification on marine organisms and ecosystems (Hofmann et al., 2011).

Changes in extremes can arise from changes in the mean, variability, or shape of the probability distribution (Coles, 2001). There exists no general accepted definition of an extreme event beyond the common understanding that an extreme is rare (Pörtner et al., 2020). As a result, many different approaches exist to define extreme events (Smith, 2011). If a relative threshold (e.g., quantile) is used to define an extreme event, it is important to distinguish between extreme events that are defined with respect to a fixed reference period or baseline, or if the reference period or baseline moves with time. If the baseline is fixed, the changes in the mean state as well as changes in variability and higher moments of the distribution contribute to changes in extreme events (e.g., Fischer and Knutti, 2015; Frölicher et al., 2018; Oliver et al., 2018). However, if a shifting baseline is used, changes in the mean state do not contribute to changes in extreme events (e.g., Stephenson, 2008; Seneviratne et al., 2012; Zscheischler and Seneviratne, 2017; Cheung and Frölicher, 2020; Vogel et al., 2020). In this case, changes in extremes arise solely due to changes in variability and higher moments of the distribution (Oliver et al., 2019). This latter definition ensures that values are not considered extreme solely because the baseline changes under climate change (Jacox, 2019; Oliver et al., 2019). Whether extreme events should be defined with respect to a fixed baseline or with respect to a shifting baseline depends on the scientific question. For example, the shifting-baseline approach may be more appropriate when the ecosystems under consideration are likely able to adapt to the mean changes but not to changes in variability (Seneviratne et al., 2012; Oliver et al., 2019). Here, we use both approaches, with a special focus on the analysis of ocean acidity extremes with respect to shifting baselines.

Under continued long-term ocean acidification (i.e., changes in the mean), one can expect that extreme events in $[H^+]$ and $\Omega$, when defined with respect to a fixed reference period or baseline, will become more frequent and intense (Hauri et al., 2013). In addition to the changes in the mean, recent studies suggest that the seasonal cycles in $[H^+]$ and $\Omega$ are also strongly modulated under elevated atmospheric $CO_2$. Higher background concentrations of dissolved inor-

ganic carbon and warmer temperatures produce stronger departures from mean state values for a given change in pertinent physical or chemical drivers for $[H^+]$ and weaker departures for $\Omega$ (Kwiatkowski and Orr, 2018; Fassbender et al., 2018). Other studies have also addressed the changes in the seasonal cycle of $p\text{CO}_2$ (Landschützer et al., 2018; Gallego et al., 2018; McNeil and Sasse, 2016; Rodgers et al., 2008; Hauck and Völker, 2015). Over the 21st century and under a high-greenhouse-gas-emission scenario, Earth system model simulations project that the seasonal amplitude in surface $[H^+]$ will increase by 81 %, whereas the seasonal amplitude for aragonite saturation state ($\Omega_A$) is projected to decrease by 9 % globally on CE2 average (Kwiatkowski and Orr, 2018). Recent observation-based estimates as well as theoretical arguments support these projected increases in seasonality for $[H^+]$ and $p\text{CO}_2$ (Landschützer et al., 2018; Fassbender et al., 2018). Thus, when extremes are defined with respect to a shifting baseline (i.e., mean state changes are neglected), the frequency and intensity of extreme $[H^+]$ events will likely increase due to increases in variability.

Unlike for marine CE3 heat waves (Frölicher et al., 2018; Collins et al., 2020) and extreme sea level events (Oppenheimer et al., 2020), little is known about the characteristics and changes of extreme ocean acidity events and, if so, only on seasonal timescales (Kwiatkowski and Orr, 2018). A global view of how extreme events in ocean chemistry will unfold in time and space and a mechanistic understanding of the relevant processes is missing. This knowledge gap is of particular concern as it is expected that extreme events in ocean acidity, defined with respect to both a fixed and a shifting baseline, are likely to become more frequent and intense under increasing atmospheric $\text{CO}_2$. Given the potential for profound impacts on marine ecosystems, quantifying trends and patterns of extreme events in ocean acidity is a pressing issue.

In this study, we use daily mean output of a five-member ensemble simulation under low- and high-$\text{CO}_2$-emission scenarios of a comprehensive Earth system model to investigate how the occurrence, intensity, duration, and volume of $[H^+]$ and $\Omega$ extreme events change under rising atmospheric $\text{CO}_2$ levels. Extreme events defined with respect to both a fixed preindustrial and a shifting baseline are assessed, but the main focus is on extremes with respect to a shifting baseline and how these are affected by variability changes.

## 2 Methods

### 2.1 Model and experimental design

The simulations used in this study were made with the fully coupled carbon–climate Earth system model developed at the NOAA Geophysical Fluid Dynamics Laboratory (GFDL ESM2M) (Dunne et al., 2012, 2013). The GFDL ESM2M model consists of ocean, atmosphere, sea-ice, and land modules and includes land and ocean biogeochemistry. The ocean component is the Modular Ocean Model version 4p1 (MOM4p1), with a nominal 1° horizontal resolution increasing to 1/3° meridionally at the Equator, with a tripolar grid north of 65° N, and with 50 vertical depth levels. The MOM4p1 model has a free surface, with the surface level centered around about 5 m depth, and the spacing between consecutive levels is about 10 m down to a depth of about 230 m (Griffies, 2009) with increasing spacing below. The dynamical sea-ice model uses the same tripolar grid as MOM4p1 (Winton, 2000). The atmospheric model CE4 version 2 (AM2) has a horizontal resolution of 2° × 2.5° with 24 vertical levels (Anderson et al., 2004). The land model version 3 (LM3) simulates the cycling of water, energy, and carbon dynamically and uses the same horizontal grid as AM2 (Shevliakova et al., 2009).

The ocean biogeochemical and ecological component is version two of the Tracers of Ocean Phytoplankton with Allometric Zooplankton (TOPAZv2) module that parameterizes the cycling of carbon, nitrogen, phosphorus, silicon, iron, oxygen, alkalinity, lithogenic material, and surface sediment calcite (see Supplement in Dunne et al., 2013). TOPAZv2 includes three explicit phytoplankton groups – small, large, and diazotrophs – and one implicit zooplankton group. The ocean carbonate chemistry is based on the OCMIP2 parameterizations (Najjar and Orr, 1998). The dissociation constants for carbonic acid and bicarbonate ions are from Dickson and Millero (1987), which are based on Mehrbach et al. (1973), and the carbon dioxide solubility is calculated according to Weiss (1974). Total alkalinity in TOPAZv2 includes contributions from phosphoric and silicic acids and their conjugate bases. TOPAZv2 also simulates diurnal variability in ocean physics as well as in phytoplankton growth. While diurnal variations in open-ocean pH are therefore simulated to some extent, we do not expect the model to fully capture the high diurnal variability in seawater chemistry, especially in coastal regions with large biological activity (Kwiatkowski et al., 2016; Hofmann et al., 2011), due to its relatively coarse resolution and simple biogeochemical model.

We ran a five-member ensemble simulation covering the historical 1861–2005 period, followed by a high (RCP8.5; RCP: Representative Concentration Pathway) and a low-greenhouse-gas-emission scenario (RCP2.6) over the 2006–2100 period with prescribed atmospheric $\text{CO}_2$ concentrations. RCP8.5 is a high-emission scenario without effective climate policies, leading to continued and sustained growth in greenhouse gas emissions (Riahi et al., 2011). In the GFDL ESM2M model, global atmospheric surface temperature in the RCP8.5 ensemble is projected to increase by 3.24 °C (ensemble minimum of 3.17 °C to ensemble maximum of 3.28 °C) CE5 between the preindustrial period and 2081–2100. The RCP2.6 scenario represents a low-emission, high-mitigation future (van Vuuren et al., 2011) with a simulated warming in the GFDL ESM2M model of 1.21 (1.18–

1.26) °C. The five ensemble members over the historical period were initialized from a multicentury preindustrial control simulation that was extended with historical land use over the 1700–1860 period (Sentman et al., 2011). The five ensemble members were generated by adding different sea surface temperature (SST) disturbances of the order $10^{-5}$ °C to a surface grid cell in the Weddell Sea at 70.5° S, 51.5° W on 1 January 1861 (Wittenberg et al., 2014; Palter et al., 2018). Although the ocean biogeochemistry is not perturbed directly, [$H^+$] and $\Omega$ differences between the ensemble members spread rapidly over the globe. On average, the ensemble members can be regarded as independent climate realizations after about 3 years of simulation for surface waters and after about 8 years at 200 m (Frölicher et al., 2020). Neither the choice of the perturbation location nor the choice of the perturbed variable has a discernible effect on the results presented here (Wittenberg et al., 2014). In addition, an accompanying 500-year preindustrial control simulation was performed.

## 2.2 Analysis

### 2.2.1 Extreme-event definition and characterization

We analyze daily mean data of [$H^+$] and aragonite saturation state $\Omega_A$ in the upper 200 m of the water column. [$H^+$] is on the total scale and hence the sum of the concentrations of free protons and hydrogen sulfate ions. We define an event as a [$H^+$] extreme event when the daily mean [$H^+$] exceeds the 99th percentile, i.e., occurring once every 100 d. Similarly, we define a $\Omega_A$ extreme event when the daily mean $\Omega_A$ falls below the 1st percentile. The percentiles are calculated for each grid cell from daily mean data of the 500-year preindustrial control simulation. In contrast to absolute thresholds, relative thresholds, such as those used here, take into account regional differences in a variable's mean state, variance, and higher moments. Events that are defined based on relative thresholds have the same probability of occurrence across the globe in the period in which they are defined (e.g., preindustrial period; see also Frölicher et al., 2018).

We assess changes in [$H^+$] and $\Omega_A$ extreme events when they are defined with respect to both a fixed preindustrial baseline and a shifting baseline. Under the fixed baseline approach, the secular trends as well as changes in variability and the higher moments of the distribution impose changes in extreme events. Under the shifting-baseline approach, which is the focus of this study, a value is considered extreme when it is much higher or lower than the baseline that undergoes changes due to secular trends in the variable. Thus, changes in the different extreme-event characteristics are only caused by changes in variability and the higher moments of the distributions. To define the extreme events with respect to the shifting baselines, we subtract the secular trends in [$H^+$] and $\Omega_A$ at each grid cell and in each individual ensemble member prior to the calculation of the different extreme-event charac-

teristics based on the preindustrial percentiles (depicted for one grid cell in Fig. 1). The secular trend is calculated as the five-member ensemble mean, which has been additionally smoothed with a 365 d running mean to keep the seasonal signal in the data (further information in Appendix A). The removal of the secular trend ensures that the mean state in the processed data stays approximately constant while day-to-day to interannual variability can change over the simulation period (Fig. 1).

We calculate four extreme-event metrics: (a) the yearly extreme days (in days; number of days per year above the 99th percentile for [$H^+$] and below the 1st percentile for $\Omega_A$), (b) the annual mean duration (in days; the average number of days above the 99th percentile for [$H^+$] and below the 1st percentile for $\Omega_A$ of single events within a year), (c) the annual mean maximal intensity (in nmol kg$^{-1}$ or $\Omega_A$ unit; maximum [$H^+$] or $\Omega_A$ anomalies with respect to the percentile threshold over the duration of a single extreme event and then averaged over all events within a year), and (d) the mean volume covered by individual extreme events in the upper 200 m (in km$^3$; mean volume of 3D clusters of connected grid cells that are above the 99th percentile for [$H^+$] or below the 1st percentile for $\Omega_A$, calculated using the *measure.label* function from the *scikit-image* library for Python for each day; these daily means are then averaged annually). The yearly extreme days, duration, and maximal intensity are calculated for individual grid cells at the surface and at 200 m. While the truncation of extremes between years alters the results for duration and maximal intensity, it allows for the calculation of annual extreme-event characteristics. We focus our analysis not only on the surface, but also on 200 m to study changes in extreme events within the seasonal thermocline. Most organisms susceptible to ocean acidification are found in the upper 200 m, such as reef-forming corals and calcifying phytoplankton.

### 2.2.2 Decomposition of [$H^+$] variability into different variability components

We use three steps to decompose the total temporal variability in [$H^+$] into interannual, seasonal, and subannual variability (Fig. 2). In a first step, we calculate the climatological seasonal cycle from the daily mean data by averaging each calendar day over all years in the time period of interest. Seasonal variability is then identified with the time-series variance of this 365 d long seasonal cycle. The secular trend in the daily mean data has been removed with the five-member ensemble mean before doing the analysis. In a second step, we subtract the seasonal cycle from the data and estimate the spectral density (Chatfield, 1996) of this residual time series using the *periodogram* function from the *scipy.signal* Python library. In a third step, we calculate the variance arising from variations on interannual and subannual timescales from the spectral density to obtain interannual and subannual variability (further information is given in Appendix B). Following

this methodology, subannual variability comprises all variations in daily mean data with periodicities of less than a year that are not part of the seasonal cycle.

### 2.2.3 Taylor expansion of [H$^+$] and $\Omega_A$ variability changes

To understand the processes behind the simulated changes in the variabilities of [H$^+$] and $\Omega_A$, we decompose these changes into contributions from changes in temperature ($T$), salinity ($S$), total alkalinity ($A_T$), and total dissolved inorganic carbon ($C_T$). Assuming linearity, the difference of [H$^+$] from its mean at time step $i$ can be decomposed into contributions from the drivers by employing a first-order Taylor expansion,

$$
\begin{aligned}
H^+(i) - \overline{H^+} \simeq & \left.\frac{\partial H^+}{\partial C_T}\right|_{\overline{C_T}, \overline{A_T}, \overline{T}, \overline{S}} \left(C_T(i) - \overline{C_T}\right) \\
& + \left.\frac{\partial H^+}{\partial A_T}\right|_{\overline{C_T}, \overline{A_T}, \overline{T}, \overline{S}} \left(A_T(i) - \overline{A_T}\right) \\
& + \left.\frac{\partial H^+}{\partial T}\right|_{\overline{C_T}, \overline{A_T}, \overline{T}, \overline{S}} \left(T(i) - \overline{T}\right) \\
& + \left.\frac{\partial H^+}{\partial S}\right|_{\overline{C_T}, \overline{A_T}, \overline{T}, \overline{S}} \left(S(i) - \overline{S}\right),
\end{aligned}
\tag{1}
$$

and analogously for $\Omega_A$. The partial derivatives are evaluated at $\overline{T}$, $\overline{S}$, $\overline{C_T}$, and $\overline{A_T}$, i.e., the temporal mean values of the drivers in the period of interest. While it is important to take into account the climatological total phosphate and total silicate concentrations for calculating the partial derivatives (Orr and Epitalon, 2015), one introduces only small errors by neglecting variations in phosphate and silicate. The partial derivatives in Eq. (1) are evaluated using *mocsy 2.0* (Orr and Epitalon, 2015).

Using the Taylor decomposition (Eq. 1), one can for example express the seasonal variation in [H$^+$] as a function of the drivers' seasonal variations (Kwiatkowski and Orr, 2018). In this study, however, we analyze the time-series variance of [H$^+$] and $\Omega_A$ that also includes variability on other timescales (see Sect. 2.2.2) and the drivers of its changes. From the Taylor approximation (Eq. 1) and the definition of variance (e.g., Coles, 2001), it follows that the variance of [H$^+$] can be written as a function of the partial derivatives with respect to the drivers (sensitivities), the standard deviations of the drivers, and their pairwise correlation coefficients:

$$
\begin{aligned}
\sigma_{H^+}^2 = & \left(\frac{\partial H^+}{\partial C_T}\right)^2 \sigma_{C_T}^2 + \left(\frac{\partial H^+}{\partial A_T}\right)^2 \sigma_{A_T}^2 \\
& + \left(\frac{\partial H^+}{\partial T}\right)^2 \sigma_T^2 + \left(\frac{\partial H^+}{\partial S}\right)^2 \sigma_S^2 \\
& + 2\frac{\partial H^+}{\partial C_T}\frac{\partial H^+}{\partial A_T}\mathrm{cov}\left(C_T, A_T\right) \\
& + 2\frac{\partial H^+}{\partial C_T}\frac{\partial H^+}{\partial T}\mathrm{cov}\left(C_T, T\right) \\
& + 2\frac{\partial H^+}{\partial C_T}\frac{\partial H^+}{\partial S}\mathrm{cov}\left(C_T, S\right) \\
& + 2\frac{\partial H^+}{\partial A_T}\frac{\partial H^+}{\partial T}\mathrm{cov}\left(A_T, T\right) \\
& + 2\frac{\partial H^+}{\partial A_T}\frac{\partial H^+}{\partial S}\mathrm{cov}\left(A_T, S\right) \\
& + 2\frac{\partial H^+}{\partial T}\frac{\partial H^+}{\partial S}\mathrm{cov}\left(T, S\right),
\end{aligned}
\tag{2}
$$

where the pairwise covariances are functions of the standard deviations and correlation coefficients according to $\mathrm{cov}(x, y) = \sigma_x \sigma_y \rho_{x,y}$, and the partial derivatives are again evaluated at the temporal mean values $\overline{T}$, $\overline{S}$, $\overline{C_T}$, and $\overline{A_T}$. This methodology has also been used to propagate uncertainties in carbonate system calculations (Dickson and Riley, 1978; Orr et al., 2018) and to identify drivers of potential predictability in carbonate system variables (Frölicher et al., 2020). Based on Eq. (2) and the analogous result for $\Omega_A$, a change in variance of [H$^+$] and $\Omega_A$ can be attributed to changes in the sensitivities that arise from changes in the drivers' mean states, to changes in the drivers' standard deviations, and to changes in the pairwise correlations between the drivers. We do so by calculating the Taylor series of Eq. (2) (further information in Appendix C). We then identify the [H$^+$] variance change from mean changes in the drivers as the sum of all terms in the expansion that describe the contributions of sensitivity changes to the overall change in variance ($\Delta_s \sigma_{H^+}^2$). Likewise, we identify the contribution from standard deviation changes in the drivers ($\Delta_\sigma \sigma_{H^+}^2$). We further group terms in the expansion that stem from simultaneous changes in the sensitivities and standard deviations ($\Delta_{s\sigma} \sigma_{H^+}^2$) and the remaining terms that arise either from correlation changes alone or mixed contributions from correlation changes and changes in sensitivities and standard deviations ($\Delta_{\rho+} \sigma_{H^+}^2$). Since these four components contain all terms in the Taylor series, they exactly reproduce a change in variance represented by Eq. (2),

$$
\Delta\sigma_{H^+}^2 = \Delta_s \sigma_{H^+}^2 + \Delta_\sigma \sigma_{H^+}^2 + \Delta_{s\sigma} \sigma_{H^+}^2 + \Delta_{\rho+} \sigma_{H^+}^2.
\tag{3}
$$

We also assess the contributions to the four components from $C_T$ alone; from $C_T$ and $A_T$; and from $C_T$, $A_T$, and $T$. The equivalent procedure is also used to decompose variance change in $\Omega_A$. Further information on the decomposition is given in Appendix C.

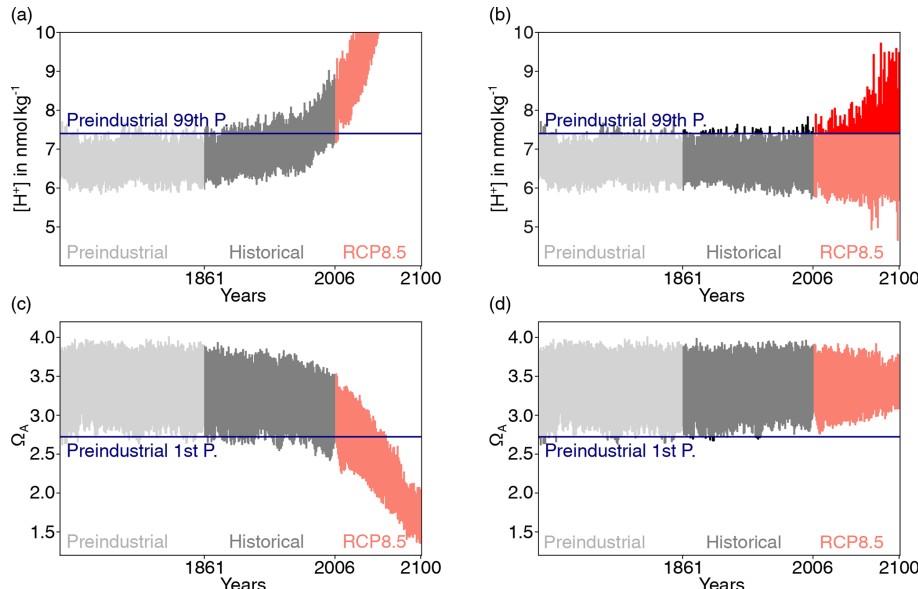

**Figure 1.** Simulated daily mean surface [H$^+$] **(a)** and $\Omega_A$ **(c)** at 40°N and 30°W in the North Atlantic for one ensemble member over the preindustrial period, the 1861–2005 historical period, and the 2006–2100 period under RCP8.5. The same data as in **(a)** and **(c)** but with subtracted ensemble-mean changes with respect to the average of the 500-year preindustrial control simulation is shown in panels **(b)** and **(d)**. For [H$^+$], the preindustrial 99th percentile threshold (horizontal blue line in panels **a** and **b**) is increasingly exceeded even when subtracting the ensemble-mean change, because [H$^+$] variability increases. In contrast, a reduction in $\Omega_A$ variability leads to a reduced undershooting of the preindustrial 1st percentile **(d)**.

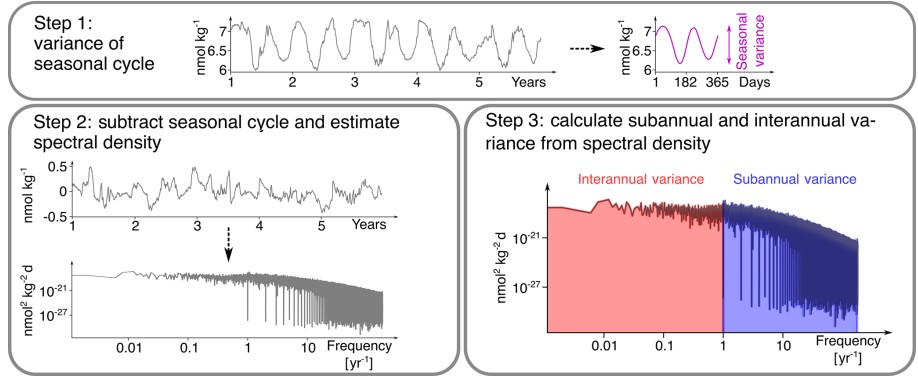

**Figure 2.** The three-step decomposition of [H$^+$] variance into interannual, seasonal, and subannual variance, exemplified for a surface grid cell at 40° N and 30° W in the North Atlantic in the preindustrial control simulation. In a first step, the climatological seasonal cycle is determined (over the whole period, only 5 years are depicted here) and its variance is calculated. Note that the seasonal cycle in this grid cell has two minima and maxima. In a second step, the spectral density of the anomalies with respect to the seasonal cycle is calculated. In a third step, interannual and subannual variance is estimated from the spectral density.

## 2.3 Model evaluation

The focus of our analysis is on changes in variability in [H$^+$] and $\Omega_A$. As observation-based daily mean data of the inorganic carbon chemistry at the global scale are not available, we limit the evaluation of the Earth system model simulation to the representation of the seasonal cycles of [H$^+$] and $\Omega_A$, and especially on its changes over the 1982–2015 period. We developed an observation-based dataset for sur-

face monthly [H$^+$] and $\Omega_A$ using monthly surface salinity, temperature, $p$CO$_2$, and $A_T$ fields. Salinity and temperature data are taken from the Hadley Centre EN.4.2.1 analysis product (Good et al., 2013). $A_T$ is then calculated using the *LIARv2* total alkalinity regression from salinity and temperature (Carter et al., 2018). For $p$CO$_2$, we use the neural-network-interpolated monthly data from Landschützer et al. (2016), which is based on SOCATv4 (Bakker et al., 2016). Although not fully capturing $p$CO$_2$ variability in regions

with only few observations (Landschützer et al., 2016), the $p\mathrm{CO_2}$ dataset appears to be generally well suited for analyzing $p\mathrm{CO_2}$ seasonality and changes therein (Landschützer et al., 2018). An exception is the Southern Ocean, where data-based $p\mathrm{CO_2}$ products are uncertain due to sparse data in winter (Gray et al., 2018). $[\mathrm{H^+}]$ and $\Omega_\mathrm{A}$ are then calculated from salinity, temperature, $A_\mathrm{T}$, and $p\mathrm{CO_2}$ using the CO2SYS carbonate chemistry package (van Heuven et al., 2011). Uncertainties in the derived seasonal cycles for $[\mathrm{H^+}]$ and $\Omega_\mathrm{A}$ that arise from uncertainties in the observation-based input variables are not quantified in this study.

In most regions, the GFDL ESM2M model captures the observation-based mean seasonal cycle in $[\mathrm{H^+}]$ and $\Omega_\mathrm{A}$ quite well, in particular for $\Omega_\mathrm{A}$ (the mean values of the seasonal amplitudes in Fig. 3). However, potential biases in the mean seasonal amplitudes do not directly have an effect on projected changes in extreme events, as we base the extreme-event definition on relative thresholds.

We then compare the simulated ensemble-mean trends in seasonal amplitude with the observation-based estimates (Fig. 3; Appendix D). Similar to the mean seasonal cycle results, the GFDL ESM2M model captures the observed trends in the seasonal $[\mathrm{H^+}]$ and $\Omega_\mathrm{A}$ amplitudes for different latitudinal bands over the 1982–2015 period relatively well. The ensemble-mean trends in the simulated seasonal $[\mathrm{H^+}]$ amplitudes are positive for all latitude bands (Fig. 3, Table 1), consistent with the observation-based estimates. While the estimates for the simulated trends are significantly larger than zero for all latitude bands, this is not the case for the observation-based trends in the equatorial region (10° S–10° N) and the northern low latitudes (10–40° N) (Table 1). The simulated $[\mathrm{H^+}]$ seasonality trends are significantly smaller (with 90 % confidence level) than estimated from observations in the northern high (40–80° N; orange thick lines in Fig. 3a, b) and southern low latitudes (40–10° S; blue thick lines in Fig. 3a, b), where the trends from the model ensemble are $0.031 \pm 0.012\,\mathrm{nmol\,kg^{-1}}$ per decade and $0.035 \pm 0.003\,\mathrm{nmol\,kg^{-1}}$ per decade, compared to the observation-based trends of $0.106 \pm 0.040\,\mathrm{nmol\,kg^{-1}}$ per decade and $0.055 \pm 0.014\,\mathrm{nmol\,kg^{-1}}$ per decade, respectively. The simulated ensemble-mean trends for the remaining latitude bands are not significantly different from the observation-based trend estimates.

For the seasonal amplitude of $\Omega_\mathrm{A}$, we find a significant negative trend in the observation-based data in the northern low latitudes and significant negative trends in the simulations in the northern and southern high latitudes (Table 1). The negative trends in seasonal amplitude in the simulations are significantly different from the observation-based trends in the northern high latitudes ($-0.015 \pm 0.004$ vs. $0.002 \pm 0.009$ $\Omega_\mathrm{A}$ units per decade) and in the southern high latitudes ($-0.012 \pm 0.002$ vs. $0.000 \pm 0.005$ $\Omega_\mathrm{A}$ units per decade).

In summary, taking into account previous evaluations of the mean states of $[\mathrm{H^+}]$ and $\Omega_\mathrm{A}$ and the underlying drivers in the GFDL-ESM2M model (Bopp et al., 2013; Kwiatkowski and Orr, 2018), the model performs well against a number of key seasonal performance metrics. However, the model slightly underestimates past increases in seasonal amplitude of $[\mathrm{H^+}]$, especially in the northern and southern high latitudes. In contrast to the observation-based data, the model also projects negative trends in the $\Omega_\mathrm{A}$ seasonal amplitude there. Nevertheless, the observation-based trends in the northern and especially southern high latitudes are uncertain because wintertime data are sparse there. Even though we lack the daily mean observation-based data to undertake a full assessment, it appears that the GFDL ESM2M model is adequate to assess changes in open-ocean variability of $[\mathrm{H^+}]$ and $\Omega_\mathrm{A}$ and to assess changes in extreme events that arise thereof.

# 3 Results

We first briefly discuss the simulated changes in $[\mathrm{H^+}]$ and $\Omega_\mathrm{A}$ extreme events when these events are defined with respect to a fixed preindustrial baseline period (Sect. 3.1). In Sect. 3.2 and 3.3, these results are contrasted with changes in extremes that are defined with respect to a shifting baseline, i.e., where the secular trends do not alter extreme events. In Sect. 3.4, variability changes are decomposed into sub-annual, seasonal, and interannual variability contributions. The processes leading to variability changes are analyzed in Sect. 3.5.

## 3.1 Global changes in extremes defined relative to a fixed preindustrial baseline

When using the fixed preindustrial 99th and 1st percentiles to define extreme events in $[\mathrm{H^+}]$ and $\Omega_\mathrm{A}$, respectively, large increases in the number of days with $[\mathrm{H^+}]$ and $\Omega_\mathrm{A}$ extremes are projected over the 1861–2100 period in both low- and high-$\mathrm{CO_2}$-emission scenarios (Figs. 4 and A1). Over the historical period, the GFDL ESM2M model projects an increase in yearly extreme days for surface $[\mathrm{H^+}]$ from 3.65 d per year during the preindustrial period to 299 d per year in 1986–2005. By year 2030 and under both $\mathrm{CO_2}$ emission scenarios, the surface ocean is projected to experience a "near-permanent acidity extreme state"; i.e., $[\mathrm{H^+}]$ is above the preindustrial 99th percentile more than 360 d pear year. Likewise, the average duration of events saturates near 365 d, and the intensity of events increases strongly, mainly reflecting the large increase in mean $[\mathrm{H^+}]$ (Fig. A1). A similar but slightly delayed evolution in the number, maximal intensity, and duration of $[\mathrm{H^+}]$ extremes is simulated at 200 m (Fig. A1).

Large increases in yearly extreme days are also projected for $\Omega_\mathrm{A}$ when using a fixed preindustrial 1st percentile as a baseline (Fig. 4b). Similar to $[\mathrm{H^+}]$, the entire surface ocean is projected to approach a permanent $\Omega_\mathrm{A}$ extreme state dur-

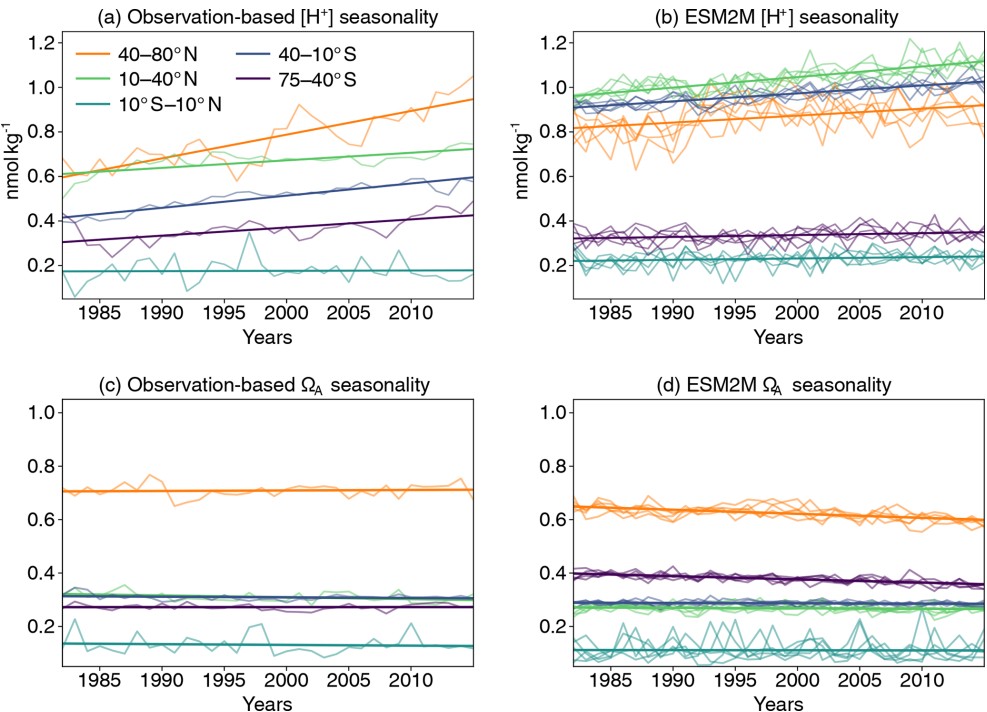

**Figure 3.** Seasonal amplitude of $[H^+]$ (calculated as yearly maximum minus the yearly minimum after subtracting a cubic spline from the data) over the period 1982–2015 averaged over five different latitude bands for the observation-based estimate **(a)** and the GFDL ESM2M model historical (1982–2005) and RCP8.5 (2006–2015) ensemble simulations **(b)**, along with the same for data-based $\Omega_A$ **(c)** and simulated $\Omega_A$ **(d)**. Linear trends in all panels are overlaid as thick lines. The linear trend of the simulated changes is calculated as the mean of the five individual ensemble trends.

**Table 1.** Linear trends in seasonal amplitude of $[H^+]$ (in $nmol\,kg^{-1}$ per decade) and $\Omega_A$ (in $10^{-3}$ per decade) for five latitude bands over the period 1982–2015. Results are shown for the observation-based data (Obs.) and the five-member ensemble mean of the ESM2M model simulations (ESM2M) following the RCP8.5 scenario over 2006–2015. The range ($\pm$) denotes the 90 % confidence interval.

| Latitude | Obs. $[H^+]$ | ESM2M $[H^+]$ | Obs. $\Omega_A$ | ESM2M $\Omega_A$ |
|---|---|---|---|---|
| 40–80° N | $0.106 \pm 0.040$ | $0.031 \pm 0.012$ | $1.9 \pm 8.7$ | $-15.1 \pm 3.8$ |
| 10–40° N | $0.034 \pm 0.034$ | $0.047 \pm 0.005$ | $-6.7 \pm 5.6$ | $-1.8 \pm 2.0$ |
| 10° S–10° N | $0.001 \pm 0.016$ | $0.006 \pm 0.005$ | $-2.8 \pm 10.7$ | $-0.5 \pm 5.3$ |
| 40–10° S | $0.055 \pm 0.014$ | $0.035 \pm 0.003$ | $-2.4 \pm 5.1$ | $-1.2 \pm 1.2$ |
| 75–40° S | $0.037 \pm 0.028$ | $0.009 \pm 0.004$ | $0.1 \pm 4.8$ | $-12.2 \pm 1.7$ |

ing the 21st century under the RCP8.5 scenario. A near-permanent extreme state is projected by year 2062. In contrast to $[H^+]$, a permanent $\Omega_A$ extreme state of the global ocean is avoided under the RCP2.6 scenario.

## 3.2 Global changes in extremes defined relative to a shifting baseline

Next, we investigate changes in $[H^+]$ and $\Omega_A$ extremes when the extreme events are defined with respect to a shifting (time-moving) baseline; i.e., changes in extremes arise only from changes in variability and higher moments of the distributions. The GFDL ESM2M model projects large increases in the number, intensity, duration, and volume of $[H^+]$ ex-

treme events over the 1861–2100 period (Fig. 5). Over the historical period (from the preindustrial period to 1986–2005), the model projects that the number of surface $[H^+]$ extreme days increases from 3.65 d per year to 10.0 d per year (Fig. 5a, ensemble ranges are given in Table 2). The maximal intensity is projected to increase from 0.08 to 0.12 $nmol\,kg^{-1}$ (Fig. 5c, Table 2) and the duration from 11 to 15 d (Fig. 5e). Compared to preindustrial conditions, these changes correspond to a 173 % increase in the number of days per year, a 44 % increase in the maximal intensity, and a 45 % increase in the duration of $[H^+]$ extreme events. The volume of individual events is projected to increase by 20 % over the historical period, from a typical volume of $2.7 \times 10^3\,km^3$, which is

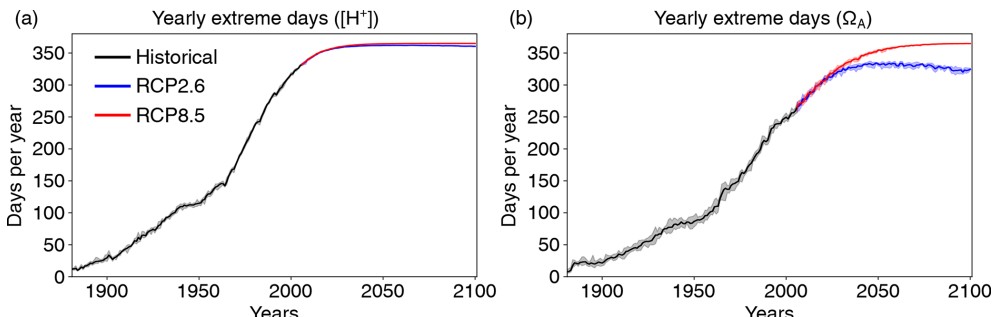

**Figure 4.** Simulated globally averaged yearly extreme days defined with respect to a fixed baseline for $[H^+]$ using the preindustrial 99th percentile **(a)** and for $\Omega_A$ using the preindustrial 1st percentile **(b)**. Shown are changes at the surface over the 1881–2100 period following historical (black lines) and future scenarios, RCP8.5 (red) and RCP2.6 (blue). The thick lines display the five-member ensemble means, and the shaded areas represent the maximum and minimum ranges of the individual ensemble members.

about 0.004 % of the total ocean volume in the upper 200 m (Fig. 5g), to $3.2 \times 10^3$ km$^3$.

Over the 21st century, extreme events in ocean acidity, defined with respect to a shifting baseline, are projected to further increase in frequency, intensity, duration, and volume (Fig. 5). By 2081–2100 under the RCP8.5 scenario, the number of $[H^+]$ extreme days per year at the surface is projected to increase to 50 d (corresponding to a 1273 % increase relative to the preindustrial period). The maximal intensity is projected to increase to 0.38 nmol kg$^{-1}$ (371 % increase), the duration to 32 d (199 % increase), and the volume to $13.9 \times 10^3$ km$^3$ (414 % increase).

At 200 m, the $[H^+]$ extreme events in preindustrial conditions are in general more intense (0.17 nmol kg$^{-1}$; Fig. 5d) and longer lasting (38 d; Fig. 5f) than at the surface. The stronger extreme events are caused by the overall larger variability at 200 m than at the surface in the preindustrial period. The longer duration is connected to the more pronounced contribution from interannual variability (see Sect. 3.4). However, projected relative changes over the historical period and the 21st century are smaller at 200 m than at the surface and with larger year-to-year variations across the ensembles. Under recent past conditions (1986–2005), the number of extreme days per year at 200 m is 4.3 d per year (corresponding to an 18 % increase since the preindustrial period), the maximal intensity 0.20 nmol kg$^{-1}$ (18 % increase), and the duration 46 d (21 % increase). By the end of the 21st century under the RCP8.5 scenario, the number of $[H^+]$ extreme days per year is projected to increase to 32 d per year, the maximal intensity to 0.34 nmol kg$^{-1}$, and the duration to 99 d. Notably, extreme events in $[H^+]$ are projected to become slightly less intense at 200 m than at the surface (0.34 vs. 0.38 nmol kg$^{-1}$) by the end of the century under RCP8.5, even though they were more intense in preindustrial times at depth. In contrast, surface $[H^+]$ extreme events remain shorter in duration at the end of the century than at 200 m.

Under the RCP2.6 scenario, the magnitude of changes in the different $[H^+]$ extreme-event characteristics by the end of

the century is substantially smaller than in the RCP8.5 scenario. This difference is especially pronounced at the surface (blue lines in Fig. 5). There, the number of extreme days per year, maximal intensity, and duration under the RCP2.6 are projected to be 46 % (44–47), 43 % (43–44), and 75 % (73–77) of that under the RCP8.5 scenario. At depth, the differences between the RCP2.6 and RCP8.5 scenario are less pronounced and only emerge in the second half of the 21st century. As opposed to the surface, the number of $[H^+]$ extreme days per year and the maximal intensity at 200 m as well as the volume of events are projected to increase significantly even after the atmospheric $CO_2$ concentration stabilizes in RCP2.6 around year 2050. This delayed response in the subsurface is due to the relatively slow surface-to-subsurface transport of carbon. However, this is not the case for the duration, which slightly decreases in the second half of the 21st century at depth (Fig. 5f). This decrease in duration mainly occurs in the subtropics, where events generally last longer (Fig. A3b). It is connected to an increase in the contribution from high-frequency variability to total variability in those regions over that period.

In contrast to $[H^+]$ extreme events, the number of yearly extreme days in $\Omega_A$ is projected to decrease over the historical period and during the 21st century under both the RCP8.5 and RCP2.6 scenarios (Fig. 6a–b, Appendix Table A1) when the extreme events are defined with respect to a shifting baseline. The number of surface $\Omega_A$ extreme days per year by the end of this century is projected to be 63 % smaller under RCP8.5 and 39 % smaller under RCP2.6 compared to preindustrial conditions (ensemble ranges are given in Table A1). Projected changes at depth are less pronounced than at the surface, again with larger decreases under RCP8.5 than under RCP2.6. Despite this decline in extreme events when defined with respect to a shifting baseline, the long-term decline in the mean state of $\Omega_A$ still leads to more frequent occurrence of extreme low $\Omega_A$ events when defined with respect to a fixed baseline (see Sect. 3.1).

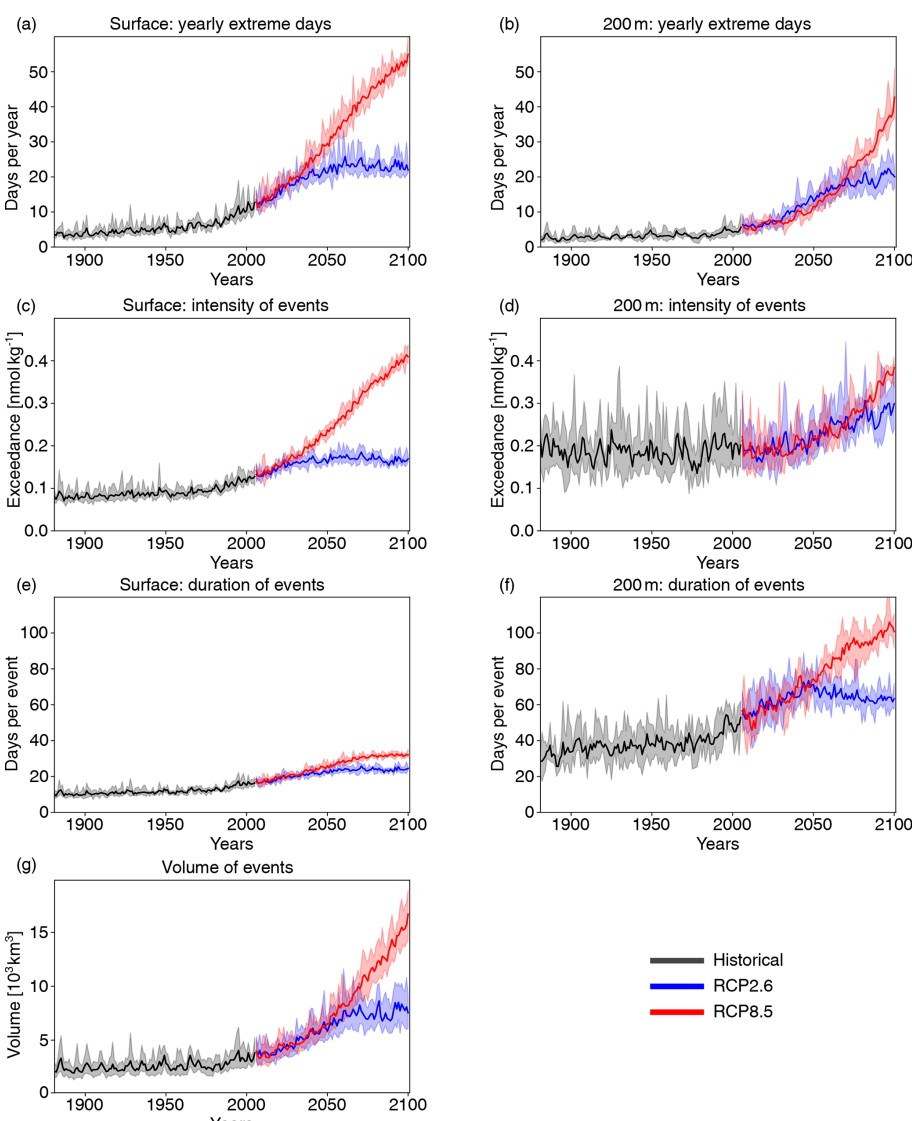

**Figure 5.** Simulated changes in globally averaged [H$^+$] extreme-event characteristics over the 1881–2100 period following historical (black lines) and future RCP8.5 (red) and RCP2.6 (blue) scenarios. The extreme events are defined with respect to a shifting baseline. Yearly extreme days, maximal intensity, and duration are shown for the surface **(a, c, e)** and for 200 m **(b, d, f)**. Volume is shown in **(g)**. The thick lines display the five-member ensemble means, and the shaded areas represent the maximum and minimum ranges of the individual ensemble members.

**Table 2.** Simulated global ensemble-mean [H$^+$] extreme-event characteristics, when extremes are defined with respect to a shifting baseline. Values in brackets denote ensemble minima and maxima. TS1

|  | PI | 1986–2005 | 2081–2100 RCP2.6 | 2081–2100 RCP8.5 |
|---|---|---|---|---|
| Yearly extreme days surf. (days per year) | 3.65 | 9.97 (9.49–10.38) | 22.87 (21.93–23.45) | 50.12 (49.98–50.30) |
| 200 m (days per year) | 3.65 | 4.32 (3.72–5.09) | 19.88 (16.96–22.53) | 32.10 (30.91–34.75) |
| Duration surf. (days) | 10.64 | 15.38 (15.04–15.72) | 23.79 (23.40–24.11) | 31.78 (31.23–32.13) |
| 200 m (days) | 38.00 | 45.95 (42.84–49.96) | 62.94 (60.49–66.11) | 98.66 (95.06–102.01) |
| Maximal intensity surf. (nmol kg$^{-1}$) | 0.08 | 0.12 (0.11–0.12) | 0.17 (0.16–0.17) | 0.38 (0.37–0.39) |
| 200 m (nmol kg$^{-1}$) | 0.17 | 0.20 (0.19–0.21) | 0.28 (0.25–0.30) | 0.34 (0.33–0.34) |
| Volume (km$^3$) | 2709 | 3247 (3082–3451) | 7654 (6873–8464) | 13927 (13 836–14 109) |

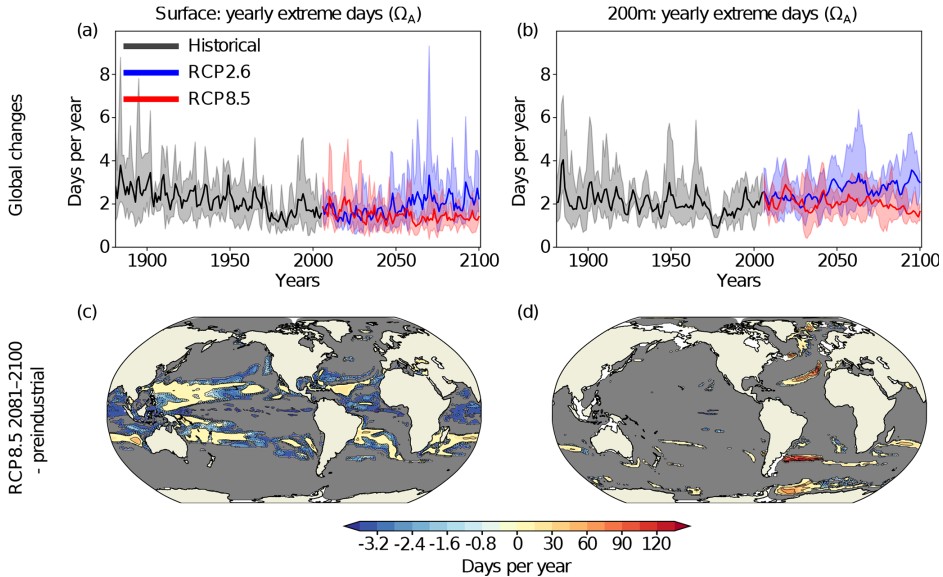

**Figure 6.** Simulated changes in the yearly number of $\Omega_A$ extreme days. The extreme events are defined with respect to a shifting baseline. Panels **(a)** and **(b)** show the globally averaged simulated yearly extreme days in $\Omega_A$ from 1881 to 2100 following historical (black lines) and future RCP2.6 (blue) and RCP8.5 (red) scenarios at the surface **(a)** and 200 m **(b)**. The thick lines display the five-member ensemble means, and the shaded areas represent the maximum and minimum range of the individual ensemble members. Panels **(c)** and **(d)** show the simulated regional changes in yearly extreme days in $\Omega_A$ from the preindustrial period to 2081–2100 under the RCP8.5 scenario at the surface **(c)** and at 200 m **(d)**. Shown are changes averaged over all five ensemble members. The black lines highlight the pattern structure and gray colors represent regions where no ensemble member simulates extremes during 2081–2100.

## 3.3 Regional changes in extremes defined relative to a shifting baseline

Surface [H$^+$] extremes that are defined with respect to a shifting baseline are projected to become more frequent in 87 % of the surface ocean area by the end of the 21st century under the RCP8.5 scenario. However, the projected changes in these ocean acidity extremes are not uniform over the globe (Fig. 7; Appendix Fig. A3). The largest increases in the number of [H$^+$] extreme days per year are projected in the Arctic Ocean (up to +120 d per year), in the subtropical gyres (up to +60 d per year), and in parts of the Southern Ocean and near Antarctica. There are also some regions including the eastern equatorial Pacific and parts of the Southern Ocean where the number of yearly extreme days in surface [H$^+$] is projected to decrease. These are in general also the regions where the seasonality in [H$^+$] is projected to decrease (see Sect. 3.4 below). The largest absolute changes in intensity of surface [H$^+$] extremes (Fig. 7c) are projected for the subtropics, especially in the Northern Hemisphere. For example, events become up to 1 nmol kg$^{-1}$ more intense in the subtropical North Pacific and Atlantic, corresponding roughly to a 10-fold increase in intensity with respect to the preindustrial period. The largest relative increases in intensity are projected for the Arctic Ocean, the North Atlantic, and around Antarctica, where more-than-10-fold increases with respect to the preindustrial period are projected. Re-

gions with large increases in the number of yearly extreme days tend to also show large increases in the duration of extreme events (Fig. 7e). The Arctic Ocean is an exception. Although the number of yearly extreme days increases strongly, the increase in duration is not as pronounced. This discrepancy is because extremes are already long lasting but rare at preindustrial times (Fig. A3). So even though extreme events are projected to occur each year by the end of the century under RCP8.5, the increase in duration is relatively small.

At 200 m, the projected pattern of changes in yearly extreme days generally resembles that at the surface (Fig. 7b). The largest increases in yearly extreme days are projected for parts of the subtropics, the Southern Ocean, and the Arctic Ocean. In contrast to the surface, [H$^+$] extremes at 200 m are projected to become less frequent in the equatorial Atlantic, the northern Indian Ocean, the North Pacific, and in large parts of the Southern Ocean. The regions indicating a decline in [H$^+$] extremes at depth include also some of the eastern boundary current systems, such as the Humboldt, California, and Benguela Current systems. In most of these regions, extreme events are projected to disappear in the RCP8.5 scenario by the end of this century (gray regions in Fig. 7b). The largest increases in subsurface event intensity are projected in the subtropics (Fig. 7d), whereas the duration of [H$^+$] extremes is projected to increase strongly in many regions of the mid-to-high latitudes of both hemispheres (Fig. 7f). The

projected increases in duration at 200 m are much larger than at the surface.

The increase in the number of extreme days per year, the maximal intensity, and the duration is smaller under RCP2.6 compared to RCP8.5 for most of the ocean (Fig. A2). The largest increases in occurrence of extremes under RCP2.6 are simulated for the Arctic Ocean, similar to under RCP8.5, and for parts of the Southern Ocean. The regions in the Southern Ocean where the occurrence of extreme events in [H$^+$] is projected to decrease largely overlap with those for RCP8.5, at the surface and at depth. On the other hand, unlike under RCP2.6, a decrease in extreme-event occurrence is only projected for a small fraction of the tropical oceans under RCP2.6.

While the decline in mean $\Omega_A$ generally leads to lower values in $\Omega_A$ and thus extreme events are becoming more frequent when defined with respect to a fixed preindustrial baseline (Sect. 3.1), extreme events in $\Omega_A$ are projected to become less frequent throughout most of the ocean when defined with respect to a shifting baseline (89 % of surface area under RCP8.5 at the end of the 21st century; Fig. 6c). In many regions, extreme events in $\Omega_A$ are projected to disappear by 2081–2100 under the RCP8.5 scenario (gray regions in Fig. 6c) when defined with respect to a shifting baseline. However, the number of yearly extreme days in $\Omega_A$ is projected to increase by 10 or more in the subtropical gyres, especially in the western parts of the subtropical gyres. At 200 m, no extreme events are projected for most of the ocean during 2081–2100 under RCP8.5 (Fig. 6d).

## 3.4 Decomposition of temporal variability in [H$^+$]

The changes in [H$^+$] extreme events defined with respect to a shifting baseline mainly result from changes in [H$^+$] variability. These variability changes may arise from changes in interannual variability, seasonal variability, and subannual variability. Thus, we decomposed the total variability into these three components (see Sect. 2.2.2). For the preindustrial period, the model simulates generally larger [H$^+$] variance at depth than at the surface (0.42 vs. 0.15 nmol$^2$ kg$^{-2}$, not shown). Seasonality has the largest contribution at the surface (81 % of total variance). At 200 m, interannual variability has the largest contribution (63 %), and also subannual variability is more important compared to the surface (15 % vs. 8 %).

Over the 1861–2100 period under the historical-RCP8.5 forcing, changes in seasonality clearly dominate the overall change in variability at the surface with 87 % contribution to the overall variance change in the global mean (Fig. 8b, d). Changes in interannual variability (3 % contribution to overall variance change; Fig. 8a, d) and subannual variability (10 %; Fig. 8c, d) play a minor role. The largest increases in variability for all three variability types are projected for the northern high latitudes. Around Antarctica and the southern end of South America, large increases in seasonal variabil-

ity are projected (Fig. 8b). In the tropical Pacific and parts of the Southern Ocean, decreases in interannual and seasonal variability are projected (Fig. 8a, b).

In contrast to the surface, changes in interannual and to a lesser extent subannual variability at 200 m are also important for explaining the overall changes in [H$^+$] variability (Fig. 8e, g, h). Changes in interannual variability contribute most to overall variance change at the global scale (with 42 % contribution). Seasonal variability changes are almost equally important (37 %), and changes in subannual variability also contribute substantially to changes in total variability (20 %). The patterns of variability changes are very similar across the three temporal components of variability. The largest increases in [H$^+$] variability are simulated north and south of the Equator. These regions tend to be already more variable during the preindustrial period (see Fig. A3a). However, the model also projects an increase in variability for regions that are less variable during the preindustrial period, such as northern high latitudes. All three temporal components of variability are projected to decrease in the tropics and parts of the Southern Ocean. The variability decrease in those regions is most pronounced for interannual variability (Fig. 8e).

## 3.5 Drivers of [H$^+$] and $\Omega_A$ variability changes

In this section, we investigate the changes in the drivers that cause the variability changes in [H$^+$] and $\Omega_A$. Drivers are carbon ($C_T$), alkalinity ($A_T$), temperature, and salinity. To do so, we attribute changes in [H$^+$] and $\Omega_A$ variability to four factors (see Sect. 2.2.3 for further details): (i) changes in the mean states of the drivers that control the sensitivities ($\Delta_s \sigma^2_{H^+}$), (ii) changes in the variabilities of the drivers ($\Delta_\sigma \sigma^2_{H^+}$), (iii) simultaneous changes in the mean states and variabilities of the drivers ($\Delta_{s\sigma} \sigma^2_{H^+}$; this contribution arises because both mean states and variabilities change and cannot be attributed to either (i) or (ii) alone), and (iv) changes in the correlations between the drivers, also including mixed contributions from correlation changes together with mean state and variability changes ($\Delta_{\rho+} \sigma^2_{H^+}$). In other words, (iv) describes the change in variability that arises because the correlations between the drivers also change, and not only their mean states and variabilities.

The drivers' mean changes between the preindustrial period and 2081–2100 under RCP8.5 cause a strong increase in surface [H$^+$] variability, which is most pronounced in the high latitudes ($\Delta_s \sigma^2_{H^+}$; red line in Fig. 9a, black dashed line in Fig. 9b). On a global average, these variance changes due to the mean changes in the drivers ($\Delta_s \sigma^2_{H^+} = 1.3$ nmol$^2$ kg$^{-2}$) are much larger than the total simulated variance change in [H$^+$] ($\Delta_\sigma \sigma^2_{H^+} = 0.5$ nmol$^2$ kg$^{-2}$, dashed gray or solid black line in Fig. 9a). In general, an increase in mean $C_T$, temperature, and salinity would lead to an increase in $\Delta_s \sigma^2_{H^+}$, whereas an increase in mean $A_T$ would lead to a

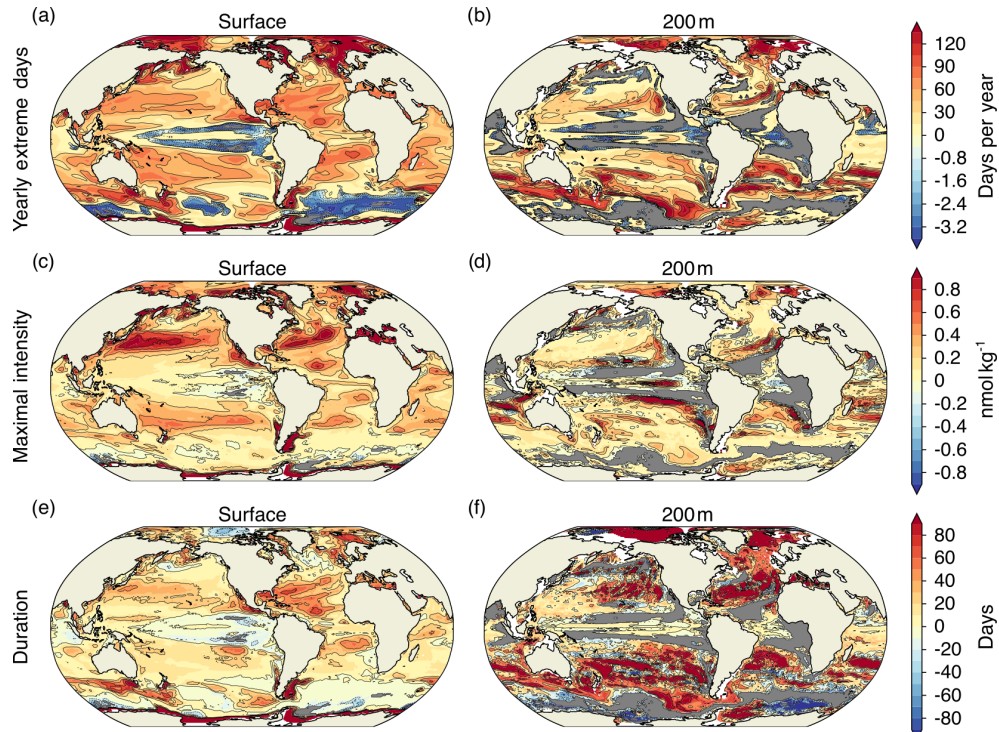

**Figure 7.** Simulated regional changes in [H$^+$] extreme-event characteristics from the preindustrial period to the 2081–2100 period under the RCP8.5 scenario at the surface and at depth for the yearly extreme days **(a, b)**, the maximal intensity of events **(c, d)**, and the duration of events **(e, f)**. The extreme events are defined with respect to a shifting baseline. Shown are changes averaged over all five ensemble members. Gray colors represent areas where no extremes occur during 2081–2100, and the black lines highlight pattern structures.

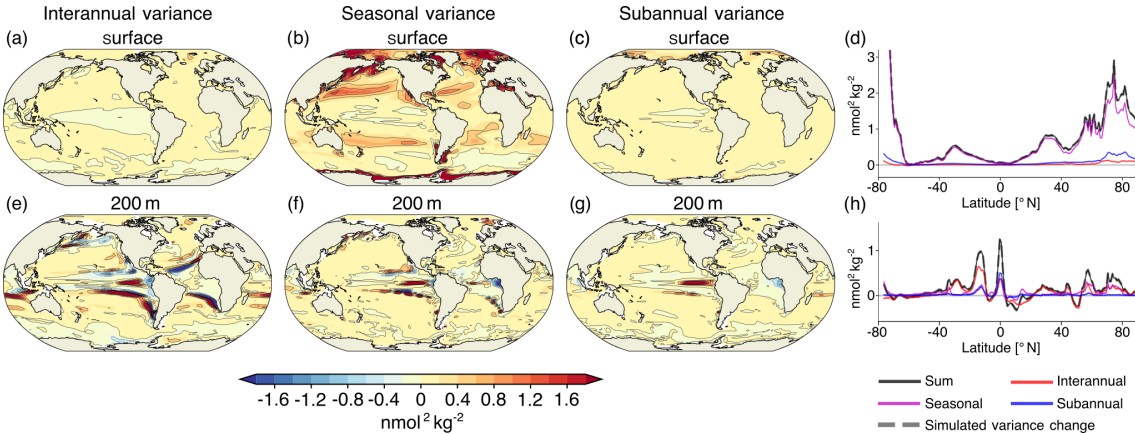

**Figure 8.** Contribution to projected changes in [H$^+$] variance from interannual variability **(a, e)**, seasonal variability **(b, f)**, and subannual variability **(c, g)** between the preindustrial period and the 2081–2100 period under the RCP8.5 forcing at the surface **(a–d)** and at 200 m **(e–h)**. Shown are the ensemble-mean changes. The black lines highlight the pattern structure. Zonal mean contributions are shown for the surface **(d)** and for 200 m **(h)**. The sum of the three components (black lines) accurately reproduces the simulated variance change (gray dashed lines).

decrease. The GFDL ESM2M model projects an increase in mean $C_T$ over the entire surface ocean (Fig. A5a) due to the uptake of anthropogenic $CO_2$ from the atmosphere and therefore an increase in $\Delta_s\sigma^2_{H^+}$ (light blue line in Fig. 9b). In the high latitudes, a relatively small increase in mean $C_T$ leads to

a large increase in $\Delta_s\sigma^2_{H^+}$, because [H$^+$] is more sensitive to changes in $C_T$ due to the low buffer capacity there. Decreases in mean $A_T$ further contribute to the increase in $\Delta_s\sigma^2_{H^+}$ in the high latitudes (green line in Fig. 9b). In the low-to-mid latitudes and in particular in the Atlantic Ocean, mean surface

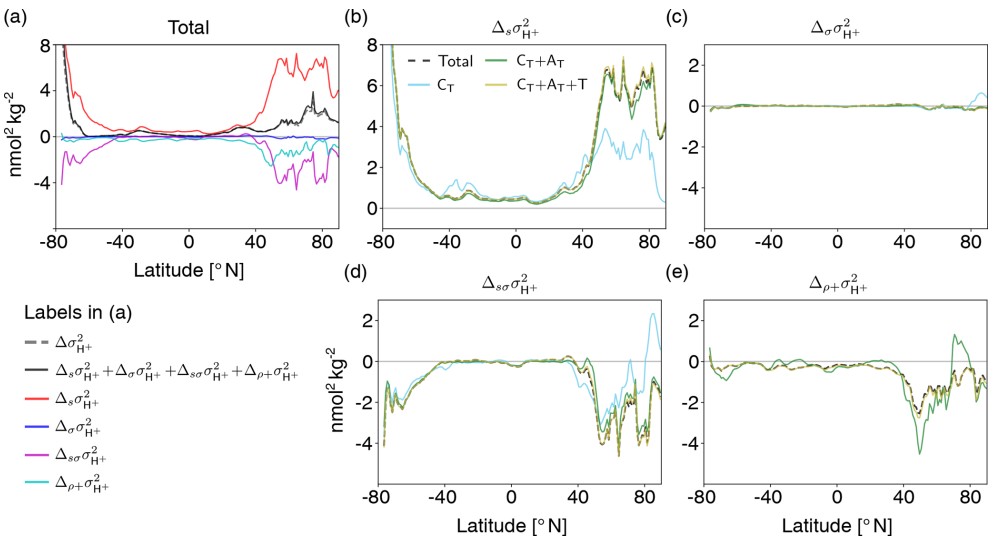

**Figure 9.** Decomposition of surface [H$^+$] variability changes into different drivers ($C_T$, $A_T$, temperature, and salinity). Shown are changes from the preindustrial period to 2081–2100 following the RCP8.5 scenario. The simulated change in [H$^+$] variance ($\Delta\sigma^2_{H+}$) is decomposed into the contribution from changes in the sensitivities that arise from changes in the drivers' mean values ($\Delta_s\sigma^2_{H+}$), the contribution from changes in the drivers' standard deviations ($\Delta_\sigma\sigma^2_{H+}$), the contribution from simultaneous changes in the sensitivities and the drivers' standard deviations ($\Delta_{s\sigma}\sigma^2_{H+}$), and the contribution from correlation changes alone together with simultaneous changes in correlations and sensitivities and standard deviations ($\Delta_{\rho+}\sigma^2_{H+}$) **(a)**. The small mismatch between the sum of the components (black line) and simulated variance change (gray dashed line) arises because the decomposition is based on Eq. (2), which is an approximation to simulated [H$^+$] variance. The contributions to these components from changes in $C_T$ alone (light blue lines); from changes in $C_T$ and $A_T$ (green lines); and from $C_T$, $A_T$, and temperature (gold lines) are shown in panels **(b)**–**(e)** TS2. The dashed black lines in panels **(b)**–**(e)** show the total components that contain contributions from all four drivers.

$A_T$ is projected to increase (Fig. A5) and therefore dampens the overall increase in $\Delta_s\sigma^2_{H+}$ (green line in Fig. 9b). The changes in $A_T$ are largely due to changes in freshwater cycling that also manifest in salinity changes (Fig. A5, Carter et al., 2016). Increases in temperature additionally increase $\Delta_s\sigma^2_{H+}$, mainly in the northern mid-to-high latitudes (gold line in Fig. 9b), but the overall impact of mean changes in temperature, and especially salinity, is small.

Why is the increase in $\Delta\sigma^2_{H+}$ (gray dashed or black solid line in Fig. 9a) smaller than the increase from the mean changes (i.e., $\Delta_s\sigma^2_{H+}$; red line in Fig. 9a)? In the high latitudes, the projected change in the variability of the drivers (Fig. A6) contributes negatively to the [H$^+$] variability change and counteracts to some degree the increase in $\Delta_s\sigma^2_{H+}$. These variability changes alone have a small imprint on $\Delta_\sigma\sigma^2_{H+}$ (blue line in Fig. 9a; black dashed line in Fig. 9c), but the variability changes dampen the increases from the mean changes ($\Delta_{s\sigma}\sigma^2_{H+}$, magenta line in Fig. 9a, black dashed line in Fig. 9d). The latter contribution is large in the high latitudes, where mean changes alone would lead to a strong increase. In the high latitudes, decreases in $C_T$ variability (Fig. A6a) together with increases in mean $C_T$ (Fig. A5a) can explain much of the negative contribution from $\Delta_{s\sigma}\sigma^2_{H+}$ (light blue line in Fig. 9d). In the northern high latitudes, mean and variability changes in $A_T$ are also impor-

tant for $\Delta_{s\sigma}\sigma^2_{H+}$ (green line in Fig. 9d). The additional contribution from changes in the correlations between the drivers ($\Delta_{\rho+}\sigma^2_{H+}$; cyan line in Fig. 9a) also tends to contribute negatively to [H$^+$] variability changes, especially in the North Atlantic, and changes in correlations with temperature play an important role (gold line in Fig. 9e). In summary, the increase in [H$^+$] variability at the surface is mainly caused by increases in mean $C_T$ attenuated by decreases in $C_T$ variability in the high latitudes. Mean changes in $A_T$ reinforce the increase in [H$^+$] variability in the northern high latitudes but dampen the increase in the low latitudes.

At 200 m, the projected increase in $\Delta\sigma^2_{H+}$ (gray dashed or black solid line in Fig. 10a) is also a result of the large increase due to the mean changes in the drivers ($\Delta_s\sigma^2_{H+}$; red line in Fig. 10a; dashed black line in Fig. 10b) and the decrease due to the interplay between mean changes and decreases in the variability ($\Delta_{s\sigma}\sigma^2_{H+}$; magenta line in Fig. 10a, black dashed line in Fig. 10d). Similar to the surface, the changes in mean and variability of $C_T$ are the most important drivers of changes (light blue lines in Fig. 10b, d). Increases in mean $A_T$ partially compensate for the increase in [H$^+$] variability due to the increase in mean $C_T$ (green lines in Fig. 10b, d). Changes in [H$^+$] variability due to changes in temperature and salinity are small. In contrast to the surface, the individual compensating contributions to

[H$^+$] variability change from mean changes and simultaneous mean and variability changes in the drivers, in particular those in $C_T$, are much larger at 200 m. The global average variance change due to the mean changes in the drivers ($\Delta_s\sigma_{H^+}^2 = 3.7$ nmol$^2$ kg$^{-2}$) is much larger than the overall simulated variance change ($\Delta\sigma_{H^+}^2 = 0.1$ nmol$^2$ kg$^{-2}$). The contribution from changes in the correlations between the drivers is overall small (cyan line in Fig. 10a) and stems mainly from changes in the correlation between $C_T$ and $A_T$ (Fig. 10e). Taken together, the increase in [H$^+$] variability at 200 m mainly arises from the balance between increases in mean $C_T$ and decreases in $C_T$ variability. Increases in mean $A_T$ dampen these changes.

Unlike for [H$^+$], both mean changes ($\Delta_s\sigma_\Omega^2$; red lines in Fig. 11) and variability changes in the drivers ($\Delta_\sigma\sigma_\Omega^2$: blue lines in Fig. 11) lead to a decrease in $\Omega_A$ variability ($\Delta\sigma_\Omega^2$; black dashed lines in Fig. 11). At 200 m, variability changes are even the dominant driver for reductions in $\Omega_A$ variability. Simultaneous changes in means and variabilities ($\Delta_{s\sigma}\sigma_\Omega^2$; purple lines in Fig. 11) contribute positively and dampen the reduction in $\Omega_A$ variability from mean and variability changes alone. Mean and variability changes in $C_T$ are the main drivers for changes in $\Omega_A$ variability as indicated by the tight relation between the dashed and solid red, blue, and purple lines in Fig. 11, in particular at 200 m. An exception is the northern high latitudes, where $A_T$ changes also play a substantial role at the surface (not shown). Correlation changes in the drivers ($\Delta_{\rho+}\sigma_\Omega^2$; cyan lines in Fig. 11) are of secondary importance and have the largest imprint in the northern mid- to high latitudes at the surface.

## 4 Discussion and conclusions

We provide a first quantification of the historical and future changes in extreme events in ocean acidity by analyzing daily mean three-dimensional output from a five-member ensemble simulation of a comprehensive Earth system model. In our analysis, we focus on changes in high-[H$^+$] and low-$\Omega_A$ extreme events that are defined with respect to a shifting baseline, where changes in extremes arise from changes in daily to interannual variability. Secular trends in the mean state were removed from the model output before analyzing extremes under this approach. We show that such extreme events in [H$^+$] are projected to become more frequent, longer lasting, more intense, and spatially more extensive under increasing atmospheric CO$_2$ concentration, both at the surface and also within the thermocline. Under RCP2.6, the increase in these extreme-event characteristics is substantially smaller than under RCP8.5. The increase in [H$^+$] variability is a consequence of increased sensitivity of [H$^+$] to variations in its drivers. It is mainly driven by the projected increase in mean $C_T$ and additionally altered by changes in $C_T$ variability and $A_T$ mean and variability as well as changes in the correlations between the drivers. In contrast to [H$^+$], variability of

$\Omega_A$ is projected to decline in the future. Therefore, extreme events in $\Omega_A$ are projected to become less frequent in the future when defined with respect to a shifting baseline. The reason for the decline in variability is that $\Omega_A$, unlike [H$^+$], becomes less sensitive to variations in the drivers with the mean increase in $C_T$. Furthermore, the projected reductions in the drivers' variabilities, mainly in $C_T$, further reduce $\Omega_A$ variability.

The analysis of extreme events defined with respect to fixed preindustrial percentiles reveals that the secular trends in [H$^+$] and $\Omega_A$ are so large that they lead to year-round or almost-year-round extreme events in the upper 200 m over the entire globe by the end of the 21st century, even under the low-emission scenario RCP2.6. Extreme events are no longer temporally and spatially bounded events that arise due to the chaotic nature of the climate system but describe a permanent new state. Under the fixed baseline approach, the relative contribution of changes in variability or higher moments of the distribution to the changes in the number of extremes is small. For example, the number of yearly extreme days for surface [H$^+$] over the 1986–2005 period under the shifting-baseline approach is only 3.8 % of that when defining the extreme events with respect to a fixed preindustrial baseline. This fraction differs regionally, reaching more than 10 % in the North Pacific, the North Atlantic, and the Arctic Ocean. However, we recall here that the changes in the number of [H$^+$] extremes when defined with respect to a shifting baseline are large. These changes in variability may need to be taken into account when assessing the impacts of ocean acidity changes on marine organisms, especially when organisms are likely to adapt to the long-term mean changes but not to changes in variability.

We use the 99th percentile of the distribution from a preindustrial simulation for the definition of the extreme [H$^+$] events (i.e., a one-in-a-hundred-days event at preindustrial levels), but the results may depend on the choice of this threshold. We tested the sensitivity of our results under the shifting-baseline approach by using also the 99.99th percentile threshold (i.e., a one-day-in-27.4-years event at preindustrial levels). The relative increase in the numbers of extreme [H$^+$] days per year is larger for these rare extremes (Fig. 12). For example, nearly every second day with [H$^+$] exceeding the 99th percentile (red solid lines in Fig. 12) is also a day with [H$^+$] exceeding the 99.99th percentile (red dotted lines in Fig. 12) by the end of the 21st century under RCP8.5, both at the surface and at depth. As a result of this large relative increase in rare extremes, the model projects as many days with [H$^+$] exceeding the 99.99th percentile by the end of the century under RCP8.5 (red dotted lines in Fig. 12) as it projects days exceeding the 99th percentile under RCP2.6 (blue solid lines in Fig. 12).

The projected increase in [H$^+$] variability and decrease in $\Omega_A$ variability also alters the occurrence of extreme events based on absolute thresholds. An often used threshold is $\Omega_A = 1$, below which seawater is corrosive with re-

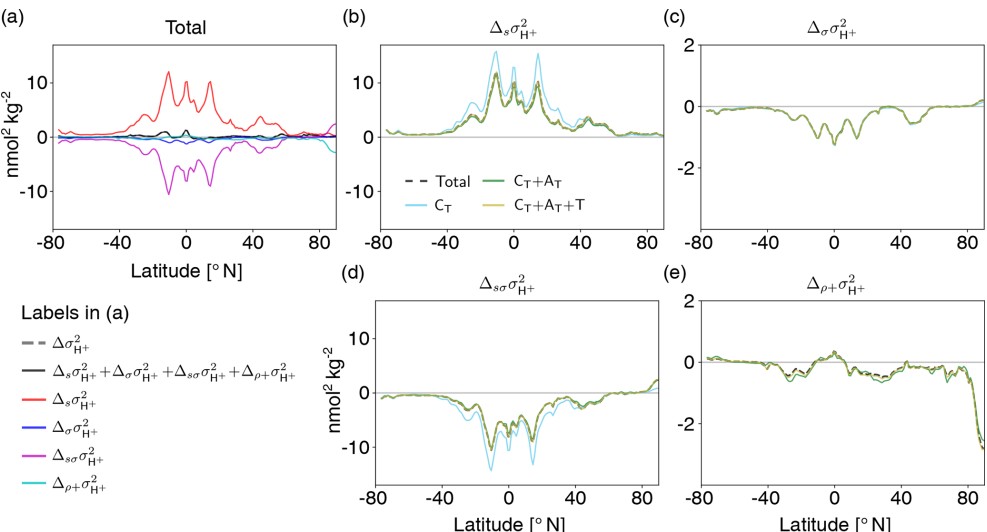

**Figure 10.** Decomposition of [H$^+$] variability changes at 200 m into different drivers ($C_\mathrm{T}$, $A_\mathrm{T}$, temperature, and salinity). Shown are changes from the preindustrial period to 2081–2100 following the RCP8.5 scenario. The simulated change in [H$^+$] variance ($\Delta\sigma_{\mathrm{H}^+}^2$) is decomposed into the contribution from changes in the sensitivities that arise from changes in the drivers' mean values ($\Delta_s\sigma_{\mathrm{H}^+}^2$), the contribution from changes in the drivers' standard deviations ($\Delta_\sigma\sigma_{\mathrm{H}^+}^2$), the contribution from simultaneous changes in the sensitivities and the drivers' standard deviations ($\Delta_{s\sigma}\sigma_{\mathrm{H}^+}^2$), and the contribution from correlation changes alone together with simultaneous changes in correlations and sensitivities and standard deviations ($\Delta_{\rho+}\sigma_{\mathrm{H}^+}^2$) **(a)**. The contributions to these components from changes in $C_\mathrm{T}$ alone (light blue lines); from changes in $C_\mathrm{T}$ and $A_\mathrm{T}$ (green lines); and from $C_\mathrm{T}$, $A_\mathrm{T}$, and temperature (gold lines) are shown in panels **(b)**–**(e)**. The dashed black lines in panels **(b)**–**(e)** show the total components that contain contributions from all four drivers. CE6

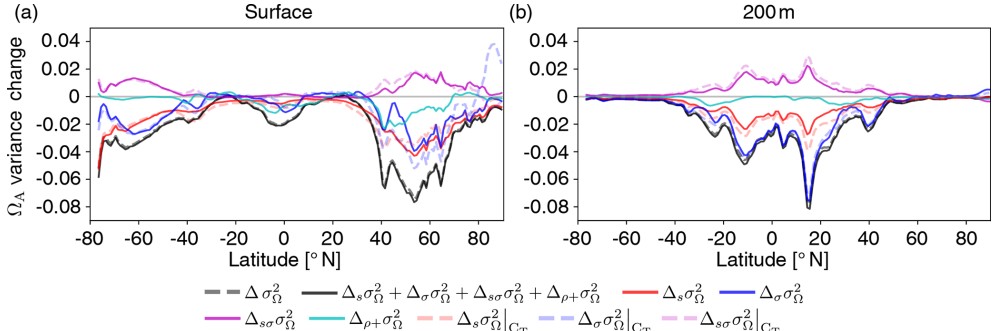

**Figure 11.** Decomposition of $\Omega_\mathrm{A}$ variability changes into different drivers. The simulated zonal mean contribution to variance changes in $\Omega_\mathrm{A}$ (black dashed lines, $\Delta\sigma_\Omega^2$) from the preindustrial period to 2081–2100 (RCP8.5) at the surface **(a)** and at 200 m **(b)**. Shown is the contribution from sensitivity changes (due to mean changes in the drivers) (red lines, $\Delta_s\sigma_\Omega^2$), standard deviation changes in the drivers (blue lines, $\Delta_\sigma\sigma_\Omega^2$), simultaneous changes in sensitivities and standard deviations (purple lines, $\Delta_{s\sigma}\sigma_\Omega^2$), and all contributions that involve changes in the drivers' correlations (cyan lines, $\Delta_{\rho+}\sigma_\Omega^2$). Furthermore, contributions from mean changes, standard deviation changes, and simultaneous mean and standard deviation changes in $C_\mathrm{T}$ alone are shown (dashed red, blue, and purple lines, respectively).

spect to the calcium carbonate mineral aragonite (Morse and Mackenzie, 1990). We assess the influence of the general decline in $\Omega_\mathrm{A}$ variability at the time where a grid cell falls below $\Omega_\mathrm{A} = 1$ for the first time. To do so, we compare these times within the historical and RCP8.5 ensemble to those for the hypothetical case where $\Omega_\mathrm{A}$ variability stays at the preindustrial level but mean $\Omega_\mathrm{A}$ undergoes the ensemble-mean evolution. We find that the decline in $\Omega_\mathrm{A}$ variability, which is observed in the historical and RCP8.5 ensemble,

leads to an average delay of the first occurrence of undersaturation by about 11 years at the surface and about 16 years at 200 m. At the surface, these delays of undersaturation occur throughout the high latitudes (Fig. 13a). At depth, the delays are most pronounced in the tropics (Fig. 13b), but delays also occur in the high latitudes. Assuming unchanged seasonality, McNeil and Matear (2008) found that seasonal aragonite undersaturation of surface waters in the Southern Ocean may occur 30 years earlier than annual mean aragonite undersat-

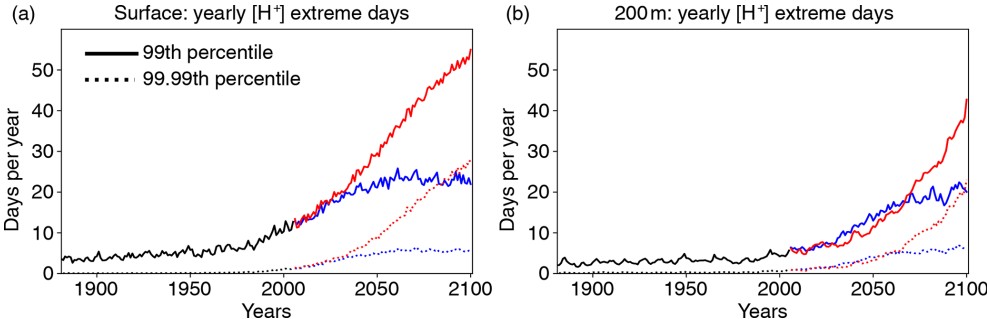

**Figure 12.** Globally averaged number of yearly extreme days for [H$^+$] over the historical (black lines), RCP2.6 (blue), and RCP8.5 (red) simulations for the preindustrial 99th (solid lines) and 99.99th percentile (dotted lines) at the surface **(a)** and 200 m **(b)**. The extreme events are defined with respect to shifting baselines.

uration. However, our simulation shows that the reduction in $\Omega_A$ variability delays the onset of undersaturation by about 10 to 15 years in the Southern Ocean relative to a hypothetical simulation where variability does not change. Therefore, changes in variability need to be taken into account when projecting the onset of seasonal undersaturation, especially in the high latitudes and in the thermocline of the tropics.

Previous studies have shown that the seasonal cycle of surface ocean $p$CO$_2$ will be strongly amplified under increasing atmospheric CO$_2$ (Gallego et al., 2018; Landschützer et al., 2018; McNeil and Sasse, 2016) and that a similar amplification is expected for surface [H$^+$] (Kwiatkowski and Orr, 2018). Here we show that the changes in the seasonal cycle of [H$^+$] translate into large increases in short-term extreme acidity events at the surface as well as at 200 m, when these events are defined with respect to a shifting baseline. In addition to earlier studies, we also show that changes in subannual variability, which are only partially resolved by monthly mean data, contribute to changes in extreme events in [H$^+$] under increasing atmospheric CO$_2$. Furthermore, we show that the average duration of extreme events at the surface and in recent past conditions (1986–2005) is about 15 d. To resolve such events that last for days to weeks, it is necessary to use daily mean output. Currently, ocean carbonate system variables from models that participate in the sixth phase of the Coupled Model Intercomparison Project are routinely stored with a monthly frequency on the Earth system grid (Jones et al., 2016). We therefore recommend storing and using high-frequency output to study extreme events in the ocean carbonate systems.

Even though we consider our results as robust, a number of potential caveats remain. First, the horizontal resolution of the ocean model in the GFDL ESM2M model is rather coarse and cannot represent critical scales of small-scale circulation structures (e.g., Turi et al., 2018). In addition, the biogeochemical processes included in the GFDL ESM2M model are designed for the open ocean but do not capture the highly variable coastal processes (Hofmann et al., 2011). High-resolution ocean models with improved pro-

cess representations are therefore needed to explore variability in ocean carbonate chemistry, especially in coastal regions and smaller ocean basins, such as the Arctic (Terhaar et al., 2019a, b). Observation-based carbonate system data with daily mean resolution would also be necessary to thoroughly evaluate the models' capability to represent day-to-day variations in carbonate chemistry. Secondly, our results, in particular at the local scale, might depend on the model formulation. As the mean increases in $C_T$ mainly drive the increases in [H$^+$] variability (see Fig. 9b), we expect that models with larger oceanic uptake of anthropogenic carbon show larger increases in [H$^+$] variability than models with lower anthropogenic carbon uptake. The GFDL ESM2M model matches observation-based estimates of historical global anthropogenic CO$_2$ uptake relatively well but still has difficulties in representing the regional patterns in storage (Frölicher et al., 2015). Therefore, the exact regional patterns of $C_T$ changes may differ from model to model. Further studies focusing on the physical processes that lead to the regional $C_T$ changes may help to better constrain the regional patterns in variability changes. In addition, it is currently rather uncertain how [H$^+$] and $\Omega_A$ variability changes as a result of changes in the drivers' variabilities. We have demonstrated that this factor is particularly important for $\Omega_A$ and for [H$^+$] at depth. It is well known that current Earth system models have imperfect or uncertain representations of ocean variability over a range of timescales (Keller et al., 2014; Resplandy et al., 2015; Frölicher et al., 2016). A possible way forward would be to assess variability changes and changes in ocean acidity extreme events within a multimodel ensemble, which would likely provide upper and lower bounds. Finally, it is assumed that physical and biogeochemical changes in the ocean will also increase diurnal variability. In particular in coastal areas, such diurnal variations can have amplitudes that are much larger than the projected changes over the 21st century (Hofmann et al., 2011). However, the GFDL ESM2M model does not fully resolve the diurnal variability. Future studies with Earth system models that resolve diurnal processes are needed to quantify changes in diurnal vari-

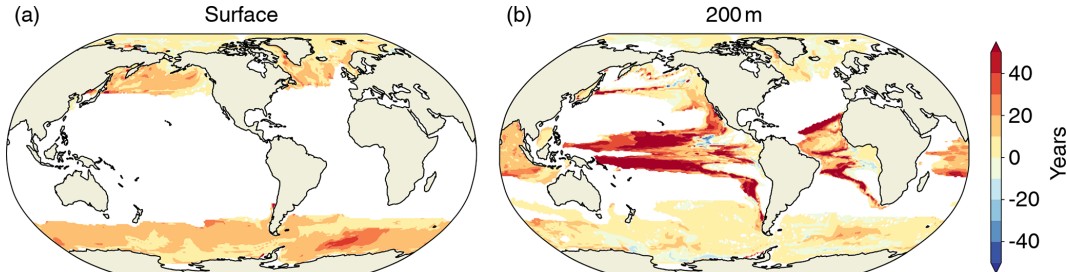

**Figure 13.** The difference in years between the first occurrence of aragonite undersaturation in the historical and RCP8.5 ensemble and a hypothetical simulation where variability does not change over the 1861–2100 period, but only the mean changes. Positive values (yellow and red) indicate a delayed onset of undersaturation resulting from declines in $\Omega_A$ variability.

ability and the impacts of these changes on extreme acidity events.

Our results may also have important consequences for our understanding of the impacts of ocean acidification on marine organisms and ecosystems. The projected increase in the frequency and the duration of ocean acidity extremes implies that marine organisms will have less time to recover from high-[H$^+$] events in the future. Organisms that cannot adapt to the large long-term changes in mean [H$^+$] will likely be the most impacted. However, even if organisms may be able to adapt to the long-term increase in [H$^+$], the large projected increase in [H$^+$] extreme events due to changes in variability may push organisms and ecosystems to the limits of their resilience, especially those organisms that are commonly accustomed to a more steady environment. The risks for substantial ecosystem impacts are aggravated by the fact that the frequency and intensity of marine heat waves are also projected to substantially increase (Frölicher et al., 2018), which also negatively impact marine ecosystems (Wernberg et al., 2016; Smale et al., 2019). The interactions of intensified multiple stressors have the potential to influence marine ecosystems and the ocean's biogeochemical cycles in an unprecedented manner (Gruber, 2011). However, further research is needed to understand the combined impacts of short-term ocean acidity extremes and marine heat waves on marine ecosystems.

In conclusion, our analysis shows that [H$^+$] and $\Omega_A$ in the upper 200 m are projected to be almost permanently under extreme conditions by the end of the 21st century when extremes are defined relative to preindustrial baselines. Even when accounting for the changes in the long-term mean, short-term extreme events in [H$^+$] are projected to become more frequent, to last longer, to be more intense, and to cover larger volumes of seawater due to increases in [H$^+$] variability, potentially adding to the stress on organisms and ecosystems from the long-term increase in ocean acidity.

## Appendix A: Identifying and removing the secular trend in the model data

In Sect. 3.2 and 3.3, we analyze the changes in extreme events in [H$^+$] and $\Omega_A$ that arise from day-to-day to inter-
5 annual variability changes in these variables. We therefore need to remove the secular trends from the data prior to analysis. We estimate the secular trend in a simulation from the five-member ensemble mean, assuming that subannual and interannual to decadal variations in the individual ensemble
10 members are phased randomly and do not imprint on the ensemble mean because they average out. A larger ensemble size would be necessary for this assumption to perfectly hold. However, this potential source of error does not qualitatively alter our results. We remove the seasonal cycle, here
15 defined as the 365 d long mean evolution over the course of a year, from the ensemble mean by smoothing the ensemble mean with a 365 d running mean filter, i.e., by calculating the convolution of the time series with a rectangular window of length 365 and height 1/365. This filter also removes vari-
20 ability on subannual and interannual timescales and thereby also reduces the error we make due to the small ensemble size that is discussed above. We then subtract the running-mean-filtered ensemble mean from the five ensemble members to remove the secular trend in the individual ensemble members.

**Table A1.** Simulated global ensemble-mean $\Omega_A$ extreme-event characteristics, when extremes are defined with respect to a shifting baseline. Values in brackets denote ensemble minima and maxima. TS3

|  | PI | 1986–2005 | 2081–2100 RCP2.6 | 2081–2100 RCP8.5 |
|---|---|---|---|---|
| Yearly extreme days surf. (days per year) | 3.65 | 1.75 (1.50–2.20) | 2.24 (1.86–2.93) | 1.36 (1.09–1.69) |
| 200 m (days per year) | 3.65 | 1.98 (1.51–2.77) | 3.01 (2.28–3.71) | 1.72 (1.38–2.02) |
| Duration surf. (days) | 19.70 | 17.84 (16.84–18.92) | 19.37 (18.07–21.13) | 29.28 (27.37–32.57) |
| 200 m (days) | 38.61 | 66.06 (59.74–18.92) | 98.71 (89.01–109.01) | 111.56 (106.62–122.70) |
| Maximal intensity surf. ($\times 10^{-3}$) | 2.92 | 3.42 (3.26–3.64) | 3.21 (3.07–3.48) | 1.51 (1.42–1.63) |
| 200 m ($\times 10^{-3}$) | 3.26 | 4.96 (3.87–6.67) | 7.90 (6.05–11.06) | 6.02 (2.85–9.13) |
| Volume (km$^3$) | 3640 | 3158 (2888–3460) | 3662 (3021–4215) | 3378 (3086–3714) |

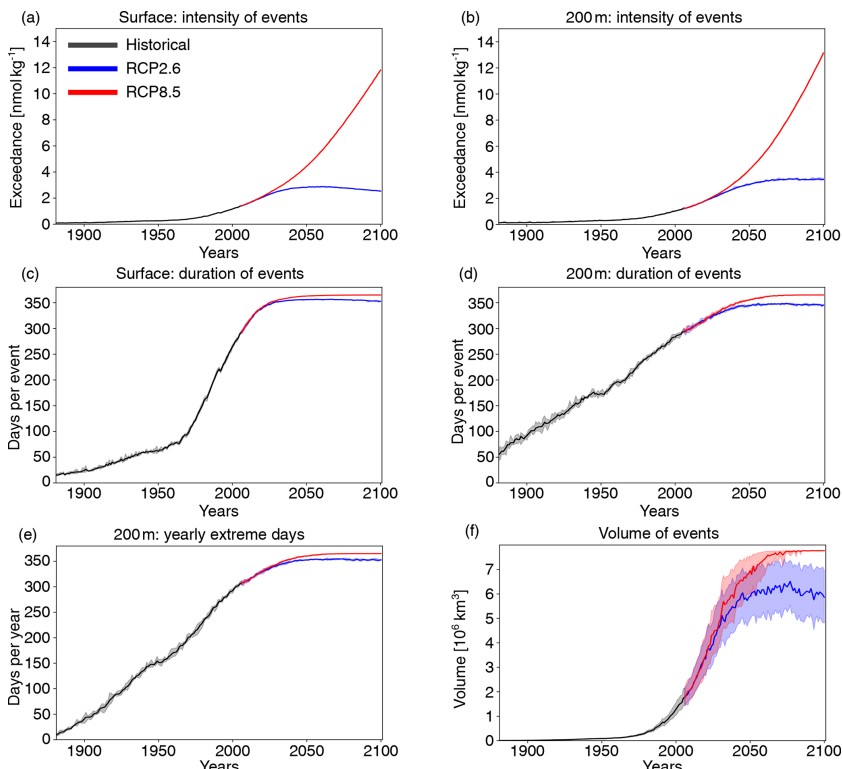

**Figure A1.** Simulated globally averaged changes in [H$^+$] extreme events defined with respect to the fixed preindustrial baseline. Shown are changes over the 1861–2100 period following historical (black lines) and future RCP8.5 (red) and RCP2.6 (blue) scenarios for maximal intensity at the surface **(a)** and at 200 m **(b)**, duration at the surface **(c)** and at 200 m **(d)**, yearly extreme days at 200 m **(e)**, and volume in the upper 200 m **(f)**. The thick lines display the five-member ensemble means, and the shaded areas represent the maximum and minimum ranges of the individual ensemble members.

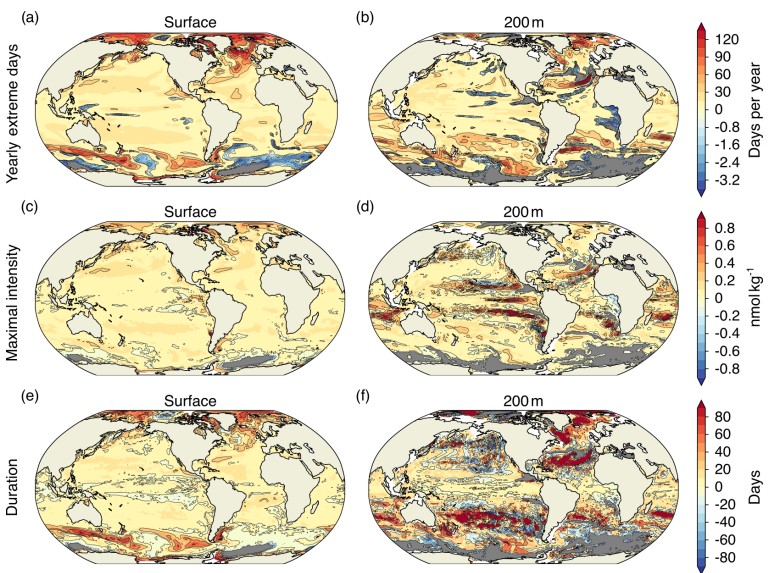

**Figure A2.** Simulated regional changes in [H$^+$] extreme-event characteristics between the preindustrial period and 2081–2100 following the RCP2.6 scenario. The extreme events are defined with respect to shifting baselines. Shown are the changes in yearly extreme days **(a, b)**, maximal intensity **(c, d)**, and duration **(e, f)**. Left panels show changes for the surface, whereas right panels show changes for 200 m. Shown are changes averaged over all five ensemble members. The black contours highlight the pattern structures. Gray areas represent areas with no extremes during 2081–2100.

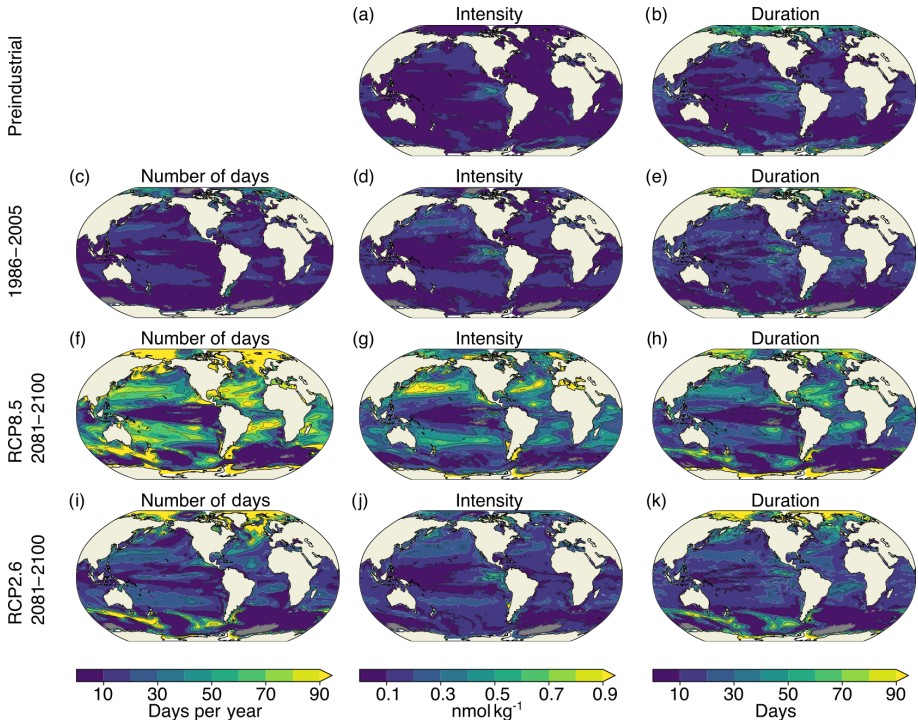

**Figure A3.** Simulated characteristics of surface [H$^+$] extreme events for the preindustrial period **(a, b)**, 1986–2005 ensemble mean **(c–e)**, RCP8.5 2081–2100 ensemble mean **(f–h)**, and RCP2.6 2081–2100 ensemble mean **(i–k)**. The extreme events are defined with respect to shifting baselines. Gray colors represent regions where no ensemble member simulates extremes. The black contours highlight the pattern structures.

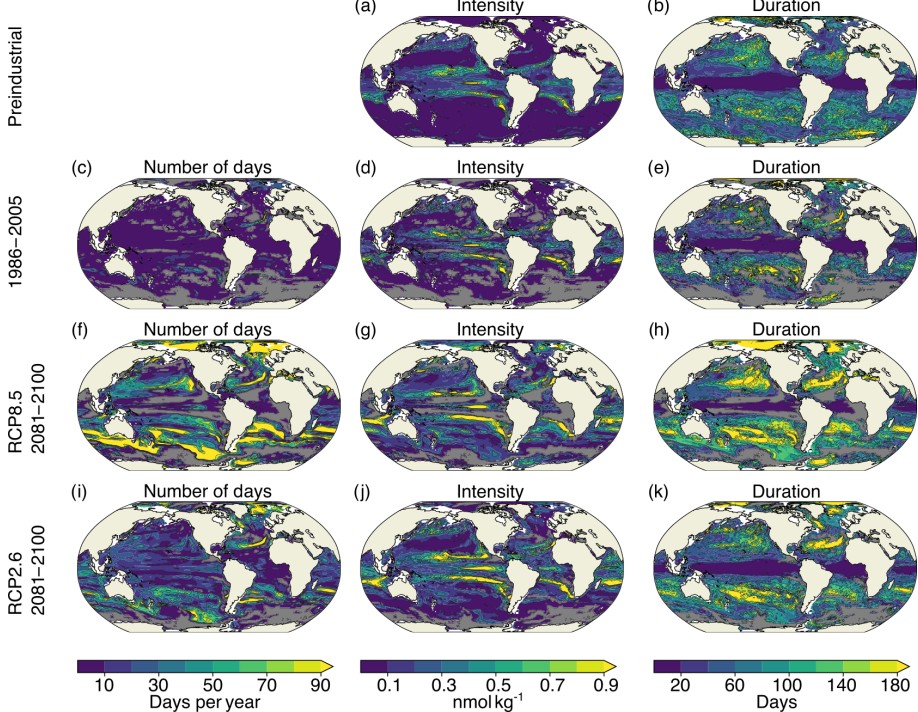

**Figure A4.** Simulated characteristics of [H$^+$] extreme events at 200 m for the preindustrial period **(a, b)**, 1986–2005 ensemble mean **(c–e)**, RCP8.5 2081–2100 ensemble mean **(f–h)**, and RCP2.6 2081–2100 ensemble mean **(i–k)**. The extreme events are defined with respect to shifting baselines. Gray colors represent regions where no ensemble member simulates extremes. The black contours highlight the pattern structures.

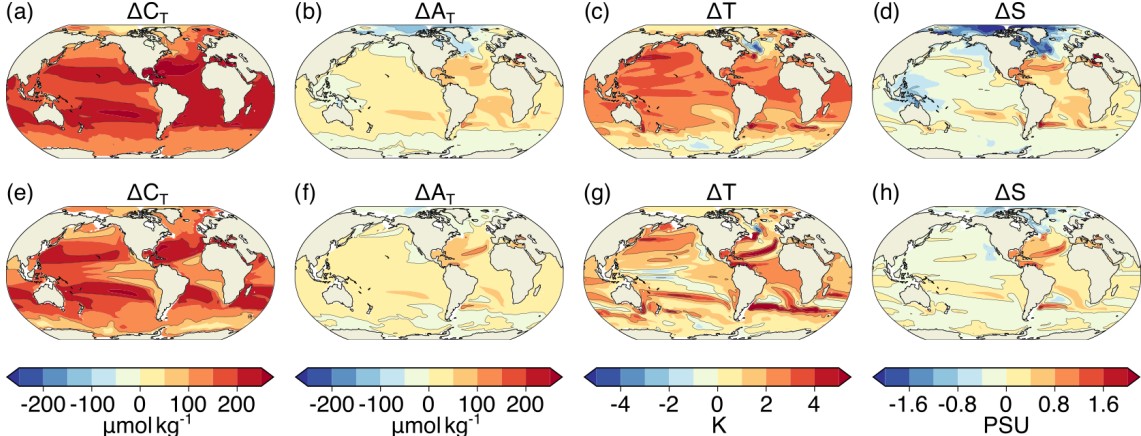

**Figure A5.** Simulated ensemble-mean changes in mean $C_T$ **(a, e)**, $A_T$ **(b, f)**, $T$ **(c, g)**, and $S$ **(d, h)** from the preindustrial period to 2081–2100 following the RCP8.5 scenario. Shown are changes for **(a–d)** the surface and **(e–h)** at 200 m. The black contours highlight the pattern structures.

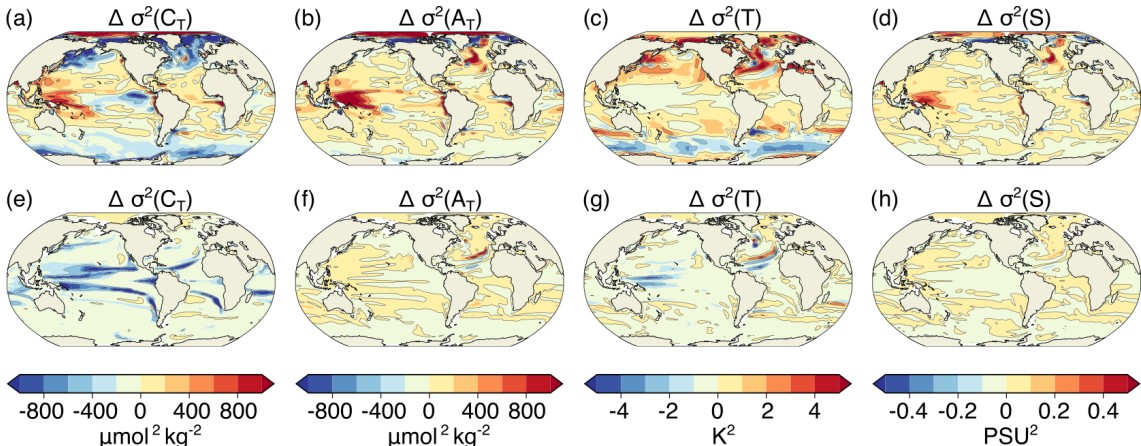

**Figure A6.** Simulated ensemble-mean changes in the variances of $C_T$ **(a, e)**, $A_T$ **(b, f)** $T$ **(c, g)**, and $S$ **(d, h)** from the preindustrial period to 2081–2100 under the RCP8.5 scenario. Shown are changes for **(a–d)** the surface and **(e–h)** at 200 m. The black contours highlight the pattern structures.

## Appendix B: Identifying interannual and subannual variability

The spectral density describes how the variance in a time series is distributed over different frequencies $\nu_j$. It is proportional to the absolute value squared of the discrete Fourier transformation (DFT) of the time series. Defining the spectral density only for positive frequencies, it is given by

$$f(\nu_j) = 2\frac{\Delta t^2}{T}\left|\sum_{k=1}^{N} x_k \cdot \exp\left(-i2\pi\nu_j \cdot \Delta t\, k\right)\right|^2, \tag{B1}$$

with $N$ the number of time steps, $x_k$ the values of the time series at each time step, $\Delta t$ the time interval between two time steps, $T = N \cdot \Delta t$, and the frequencies $\nu_j = j/T$. The autocovariance is the inverse Fourier transform of the spectral density (Wiener–Khintchine theorem, Chatfield, 1996). In the continuous case, the theorem states

$$\gamma(\tau) = \int_{-\infty}^{\infty} \tilde{f}(\nu)\exp(i2\pi\nu\tau)\mathrm{d}\nu, \tag{B2}$$

with the autocovariance function $\gamma(\tau)$ and the spectral density $\tilde{f}$ defined for positive and negative frequencies. Since the two-sided spectral density, $\tilde{f}$, is a real and even function, one can also use

$$\gamma(\tau) = \int_{0}^{\infty} f(\nu)\cos(2\pi\nu\tau)\mathrm{d}\nu \tag{B3}$$

with the one-sided spectral density $f = 2 \cdot \tilde{f}$ that is used in this text. As a consequence, the variance within the time series, given by the autocovariance at lag zero, is obtained by integrating the spectral density over all positive frequencies, $\sigma^2 = \int_0^\infty f(\nu)\mathrm{d}\nu$. For a discrete time series, where the maximal resolved frequency is given by $\nu_{\max} = 1/2\Delta t$, the identity reads

$$\sigma^2 = \sum_{j=0}^{N/2} f(\nu_j)\frac{1}{N\Delta t}. \tag{B4}$$

Based on this equation, one can separate the contributions to variance from low-frequency and high-frequency variations. In this study, we determine interannual variability and subannual variability. Interannual variability is calculated by summing over the contributions to variance from all frequencies up to a cycle of once per year, i.e., by evaluating the sum up to $i_{\mathrm{cut}}$ for which $\nu_{\mathrm{cut}} = 1/365\,\mathrm{d}^{-1}$. Accordingly, subannual variability is obtained by evaluating the sum from $i_{\mathrm{cut}} + 1$ to $N/2$. Prior to this separation, the seasonal variability is removed from the data by subtracting the 365 d climatology.

## Appendix C: Decomposition of [H⁺] variance change

Following Eq. (2) in the main text, the variance in [H$^+$] (or $\Omega_{\mathrm{A}}$) can be approximated as a function of the four sensitivities

$$\boldsymbol{s} = \left(\frac{\partial H^+}{\partial A_{\mathrm{T}}}, \frac{\partial H^+}{\partial C_{\mathrm{T}}}, \frac{\partial H^+}{\partial S}, \frac{\partial H^+}{\partial T}\right)^{\mathsf{T}} \tag{C1}$$

that in turn depend on the mean values of the drivers, the four standard deviations of the drivers

$$\boldsymbol{\sigma} = \left(\sigma_{A_{\mathrm{T}}}, \sigma_{C_{\mathrm{T}}}, \sigma_S, \sigma_{\mathrm{T}}\right)^{\mathsf{T}}, \tag{C2}$$

and the six pairwise correlation coefficients, in matrix notation given by

$$\boldsymbol{\rho} = \begin{pmatrix} 1 & \rho_{AC} & \rho_{AS} & \rho_{AT} \\ \rho_{AC} & 1 & \rho_{CS} & \rho_{CT} \\ \rho_{AS} & \rho_{CS} & 1 & \rho_{ST} \\ \rho_{AT} & \rho_{CT} & \rho_{ST} & 1 \end{pmatrix}. \tag{C3}$$

Based on this notation, we can rewrite Eq. (2) of the main text as

$$\sigma_{\mathrm{H}^+}^2 = \sum_{i=1}^{4}\sum_{j=1}^{4} s_i s_j \sigma_i \sigma_j \rho_{ij}. \tag{C4}$$

We use Eq. (C4) and decompose the variability change between the preindustrial period and 2081–2100 into the contributions from changes in $s$, $\sigma$, and $\rho$ based on a Taylor expansion. Since [H$^+$] variance represented by Eq. (C4) is a polynomial of fifth order in these variables, its Taylor series has five nonvanishing orders. We use the drivers' standard deviations instead of their variances for the decomposition. With the latter, the Taylor expansion would have infinite terms and could not be decomposed exactly as it is done in the following. However, it would asymptotically lead to the same decomposition of [H$^+$] variance change into $\Delta_s\sigma_{\mathrm{H}^+}^2$, $\Delta_\sigma\sigma_{\mathrm{H}^+}^2$, $\Delta_{s\sigma}\sigma_{\mathrm{H}^+}^2$, and $\Delta_{\rho+}\sigma_{\mathrm{H}^+}^2$ that is presented below. Furthermore, it should be noted that the resulting decomposition of [H$^+$] variance change only approximates the simulated variance change because it is based on Eq. (C4), which itself is based on a first-order Taylor expansion of [H$^+$] with respect to the drivers.

In the following, all terms of the Taylor series are given. We denote the sum of first-order terms that contain changes in the four sensitivities $\Delta s_{1,\dots 4}$ by $\Delta_s^{(1)}\sigma_{\mathrm{H}^+}^2$, the sum of second-order terms that contain changes in the sensitivities and standard deviations by $\Delta_{s\sigma}^{(2)}\sigma_{\mathrm{H}^+}^2$, and so on.

The first order is given by $\Delta^{(1)}\sigma_{\mathrm{H^+}}^2 = \Delta_s^{(1)}\sigma_{\mathrm{H^+}}^2 + \Delta_\sigma^{(1)}\sigma_{\mathrm{H^+}}^2 + \Delta_\rho^{(1)}\sigma_{\mathrm{H^+}}^2$ with

$$\Delta_s^{(1)}\sigma_{\mathrm{H^+}}^2 = 2\sum_{k=1}^{4}\sum_{j=1}^{4} s_j \sigma_k \sigma_j \rho_{kj} \Delta s_k,$$

$$\Delta_\sigma^{(1)}\sigma_{\mathrm{H^+}}^2 = 2\sum_{k=1}^{4}\sum_{j=1}^{4} s_k s_j \sigma_j \rho_{kj} \Delta \sigma_k,$$

$$\Delta_\rho^{(1)}\sigma_{\mathrm{H^+}}^2 = \sum_{k=1}^{4}\sum_{l=1}^{4} s_k s_l \sigma_k \sigma_l \Delta \rho_{kl}. \tag{C5}$$

The second order contains

$$\Delta_{ss}^{(2)}\sigma_{\mathrm{H^+}}^2 = \sum_{k=1}^{4}\sum_{l=1}^{4} \sigma_k \sigma_l \rho_{kl} \Delta s_k \Delta s_l,$$

$$\Delta_{\sigma\sigma}^{(2)}\sigma_{\mathrm{H^+}}^2 = \sum_{k=1}^{4}\sum_{l=1}^{4} s_k s_l \rho_{kl} \Delta \sigma_k \Delta \sigma_l,$$

$$\Delta_{s\sigma}^{(2)}\sigma_{\mathrm{H^+}}^2 = 2\sum_{k=1}^{4}\sum_{l=1}^{4}$$
$$(s_l \sigma_l \rho_{kl} \Delta s_k \Delta \sigma_k + s_l \sigma_k \rho_{kl} \Delta s_k \Delta \sigma_l),$$

$$\Delta_{s\rho}^{(2)}\sigma_{\mathrm{H^+}}^2 = 2\sum_{k=1}^{4}\sum_{l=1}^{4} s_l \sigma_k \sigma_l \Delta s_k \Delta \rho_{kl},$$

$$\Delta_{\sigma\rho}^{(2)}\sigma_{\mathrm{H^+}}^2 = 2\sum_{k=1}^{4}\sum_{l=1}^{4} s_k s_l \sigma_l \Delta \sigma_k \Delta \rho_{kl}. \tag{C6}$$

The third-order terms read

$$\Delta_{ss\sigma}^{(3)}\sigma_{\mathrm{H^+}}^2 = 2\sum_{k=1}^{4}\sum_{l=1}^{4} \sigma_l \rho_{kl} \Delta s_k \Delta s_l \Delta \sigma_k,$$

$$\Delta_{s\sigma\sigma}^{(3)}\sigma_{\mathrm{H^+}}^2 = 2\sum_{k=1}^{4}\sum_{l=1}^{4} s_l \rho_{kl} \Delta s_k \Delta \sigma_k \Delta \sigma_l,$$

$$\Delta_{ss\rho}^{(3)}\sigma_{\mathrm{H^+}}^2 = \sum_{k=1}^{4}\sum_{l=1}^{4} \sigma_k \sigma_l \Delta s_k \Delta s_l \Delta \rho_{kl},$$

$$\Delta_{\sigma\sigma\rho}^{(3)}\sigma_{\mathrm{H^+}}^2 = \sum_{k=1}^{4}\sum_{l=1}^{4} s_k s_l \Delta \sigma_k \Delta \sigma_l \Delta \rho_{kl},$$

$$\Delta_{s\sigma\rho}^{(3)}\sigma_{\mathrm{H^+}}^2 = 2\sum_{k=1}^{4}\sum_{l=1}^{4}$$
$$(s_l \sigma_k \Delta s_k \Delta \sigma_l \Delta \rho_{kl} + s_l \sigma_l \Delta s_k \Delta \sigma_k \Delta \rho_{kl}). \tag{C7}$$

The fourth order reads

$$\Delta_{ss\sigma\sigma}^{(4)}\sigma_{\mathrm{H^+}}^2 = \sum_{k=1}^{4}\sum_{l=1}^{4} \rho_{kl} \Delta s_k \Delta s_l \Delta \sigma_k \Delta \sigma_l,$$

$$\Delta_{ss\sigma\rho}^{(4)}\sigma_{\mathrm{H^+}}^2 = 2\sum_{k=1}^{4}\sum_{l=1}^{4} \sigma_l \Delta s_k \Delta s_l \Delta \sigma_k \Delta \rho_{kl},$$

$$\Delta_{s\sigma\sigma\rho}^{(4)}\sigma_{\mathrm{H^+}}^2 = 2\sum_{k=1}^{4}\sum_{l=1}^{4} s_l \Delta s_k \Delta \sigma_k \Delta \sigma_l \Delta \rho_{kl}. \tag{C8}$$

And the fifth order is given by

$$\Delta_{ss\sigma\sigma\rho}^{(5)}\sigma_{\mathrm{H^+}}^2 = \sum_{k=1}^{4}\sum_{l=1}^{4} \Delta s_k \Delta s_l \Delta \sigma_k \Delta \sigma_l \Delta \rho_{kl}. \tag{C9}$$

We identify the variance change from changes in the sensitivities as

$$\Delta_s \sigma_{\mathrm{H^+}}^2 = \Delta_s^{(1)}\sigma_{\mathrm{H^+}}^2 + \Delta_{ss}^{(2)}\sigma_{\mathrm{H^+}}^2, \tag{C10}$$

the change from standard deviation changes as

$$\Delta_\sigma \sigma_{\mathrm{H^+}}^2 = \Delta_\sigma^{(1)}\sigma_{\mathrm{H^+}}^2 + \Delta_{\sigma\sigma}^{(2)}\sigma_{\mathrm{H^+}}^2, \tag{C11}$$

the change from simultaneous changes in sensitivities and standard deviations as

$$\Delta_{s\sigma} \sigma_{\mathrm{H^+}}^2 = \Delta_{s\sigma}^{(2)}\sigma_{\mathrm{H^+}}^2 + \Delta_{ss\sigma}^{(3)}\sigma_{\mathrm{H^+}}^2$$
$$+ \Delta_{s\sigma\sigma}^{(3)}\sigma_{\mathrm{H^+}}^2 + \Delta_{ss\sigma\sigma}^{(4)}\sigma_{\mathrm{H^+}}^2, \tag{C12}$$

and that from correlation changes and mixed contributions that include correlation changes as

$$\Delta_{\rho+} \sigma_{\mathrm{H^+}}^2 = \Delta_\rho^{(1)}\sigma_{\mathrm{H^+}}^2 + \Delta_{s\rho}^{(2)}\sigma_{\mathrm{H^+}}^2 + \Delta_{\sigma\rho}^{(2)}\sigma_{\mathrm{H^+}}^2$$
$$+ \Delta_{ss\rho}^{(3)}\sigma_{\mathrm{H^+}}^2 + \Delta_{\sigma\sigma\rho}^{(3)}\sigma_{\mathrm{H^+}}^2 + \Delta_{s\sigma\rho}^{(3)}\sigma_{\mathrm{H^+}}^2$$
$$+ \Delta_{ss\sigma\rho}^{(4)}\sigma_{\mathrm{H^+}}^2 + \Delta_{s\sigma\sigma\rho}^{(4)}\sigma_{\mathrm{H^+}}^2 + \Delta_{ss\sigma\sigma\rho}^{(5)}\sigma_{\mathrm{H^+}}^2. \tag{C13}$$

Finally, we calculate the analogs of Eqs. (C10)–(C13) that only take into account changes in $C_{\mathrm{T}}$; changes in $C_{\mathrm{T}}$ and $A_{\mathrm{T}}$; and changes in $C_{\mathrm{T}}$, $A_{\mathrm{T}}$, and $T$. This is done by calculating $\Delta s_{1,\dots 4}$ only based on mean changes in the considered variables and by setting the standard deviation changes in variables and correlation changes in pairs of variables that are not considered to zero.

## Appendix D: Comparison of simulated ensemble-mean trends in seasonal amplitude to observation-based trends

We construct confidence intervals for the observation-based slope estimates following Hartmann et al. (2013). For the simulations, we use the arithmetic average of the five ensemble-member slope estimates, $\hat{b}_k$, as the estimator,

$$\hat{\bar{b}} = \frac{1}{5} \sum_{k=1}^{5} \hat{b}_k, \tag{D1}$$

with estimated variance

$$\hat{\sigma}_{\bar{b}}^2 = \frac{1}{5^2} \sum_{k=1}^{5} \hat{\sigma}_{b_k}^2. \tag{D2}$$

We then construct the confidence interval for $\hat{\bar{b}}$ as

$$(\hat{\bar{b}} - q \cdot \hat{\sigma}_{\bar{b}}, \hat{\bar{b}} + q \cdot \hat{\sigma}_{\bar{b}}), \tag{D3}$$

with $q$ the $(1 + p)/2$ quantile (we use $p = 0.9$) of the $t$ distribution with $5 \cdot (N - 2)$ degrees of freedom. We correct the sample size $N$ (34, the number of years we use for the fits) to a reduced sample size $N_r$ when we find positive lag-one autocorrelation in the residuals of the fits (data – linear regression model). Lag-one autocorrelation is estimated as the average of the five ensemble-member lag-one autocorrelation estimates,

$$\hat{\bar{\rho}} = \frac{1}{5} \sum_{k=1}^{5} \hat{\rho}_k, \tag{D4}$$

and we obtain $N_r = N \cdot (\hat{\bar{\rho}} - 1)/(\hat{\bar{\rho}} + 1)$. Positive $\hat{\bar{\rho}}$ is only found in the northern high latitudes. This is in contrast to the observation-based case, where we find large positive $\hat{\rho}_o$ (up to 0.7) in the residuals of all latitude bands besides the tropical region.

For testing the significance of a difference between the simulation slope estimate $\hat{\bar{b}}$ and the observation-based estimate $\hat{b}_o$, we use Welch's test, which assumes different variances for the two estimates (Andrade and Estévez-Pérez, 2014). The variance of the simulation slope estimate is calculated by dividing the ensemble-averaged slope variance by the ensemble size (Eq. D2) and is hence smaller than the observation-based slope variance. If the absolute value of the test statistic

$$\frac{\hat{\bar{b}} - \hat{b}_o}{\sqrt{\hat{\sigma}_{\bar{b}}^2 + \hat{\sigma}_o}} \tag{D5}$$

is larger than the $(1 + p)/2$ quantile of the $t$ distribution with (Andrade and Estévez-Pérez, 2014)

$$\frac{\left(\hat{\sigma}_{\bar{b}}^2 + \hat{\sigma}_{b_o}^2\right)^2}{\hat{\sigma}_{\bar{b}}^4 / (5 \cdot (N_r - 2)) + \hat{\sigma}_{b_o}^4 / (N_{r,o} - 2)} \tag{D6}$$

degrees of freedom, we consider the observation-based and simulation slope to be different from each other with a confidence level of $p = 0.9$.

*Data availability.* The GFDL ESM2M model data underlying the figures and analyses are available under https://zenodo.org/record/4032577 (Burger, 2020).

*Author contributions.* FAB and TLF designed the study. FAB performed the simulations, assisted by TLF and JGJ. FAB performed the analysis and wrote the initial manuscript. All authors contributed to the writing of the paper.

*Competing interests.* The authors declare that they have no conflict of interest.

*Disclaimer.* The work reflects only the authors' view; the European Commission and their executive agency are not responsible for any use that may be made of the information the work contains.

*Acknowledgements.* Friedrich A. Burger and Thomas L. Frölicher thank the CSCS Swiss National Supercomputing Centre for computing resources. The authors thank Elizabeth Drenkard, Fortunat Joos, and Jens Terhaar for discussions and comments; Rick Slater for the help in porting the ESM2M model code to CSCS; and James Orr, Sarah Schlunegger, Jean-Pierre Gattuso, and one anonymous reviewer for their excellent and insightful reviews.

*Financial support.* This research has been supported by the Swiss National Science Foundation (grant no. PP00P2_170687) and Horizon 2020 (COMFORT, Our common future ocean in the Earth system – quantifying coupled cycles of carbon, oxygen, and nutrients for determining and achieving safe operating spaces with respect to tipping points (grant no. 820989)).

*Review statement.* This paper was edited by Jean-Pierre Gattuso and reviewed by James Orr, Sarah Schlunegger, and one anonymous referee.

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

**Remarks from the language copy-editor**

**CE4**    Could you please provide an independent official source for "atmospheric model" and "land model" as proper nouns so the capitalization can be adjusted in accordance with our standards? I can't find a source for either and it is our standard to lowercase general terms even when they describe a model. For example, the "Whole Atmosphere Community Climate Model" is spelled with uppercase initials, as it is the name of a particular model, but "climate–chemistry model" and "general circulation model" are lowercased because they refer to a type of model and are, therefore, descriptive terms.

**Remarks from the typesetter**