# Peer review of "Increase in ocean acidity variability and extremes under increasing atmospheric $CO_2$"

_Biogeosciences, 2020_

## Referee Comment (RC1) · Anonymous Referee #1 · 6 Feb 2020

General comments:

This is a nice manuscript that assesses extreme chemistry variability in ensemble projections of an Earth system model. The manuscript is both interesting and timely. My comments mainly relate to improvements I think the authors could make in understanding the different drivers of carbonate chemistry variability. Particularly, I'd like to see more on the physical processes driving the differences between the RCPs and the projected frequency/intensity/duration of extreme variability events.

Another general issue I think is how the authors choose to define extreme events. I would be more comfortable calling these "extreme variability events" given that mean trends have been removed. This is particularly an issue when they discuss saturation state and often give the impression that extreme events are projected to decline.

Finally, given that the carbonate chemistry decompositions apparently do not sum, I'm not convinced of their value. I suggest removing this analysis if it can't be properly validated.

Specific comments:

L21-23. This could be better explained. I suggest a sentence or two more, including a full definition of omega.

L43. I don't think Hofmann et al., 2011 is really relevant here as they don't assess organism adaptation/acclimation under variable chemistry regimes. Many other papers do, with mixed findings, for example see: (Rivest et al., 2017; Cornwall et al., 2020)

L45-48. The authors are being too concise here. Explain what you mean by undersaturation. I understand that you might have to refer to aragonite versus calcite but that should probably be already mentioned anyway.

L77. Its not very clear what "residual" means in this context, as it hasn't been defined yet.

L90. What is the depth of the first ocean level? This will useful to know, as it will be a major determinant of surface variability.

L120. What is potential vegetation? Are you referring to a coupled terrestrial carbon cycle? It is not clear what relevance this has.

L142. Do you mean that extremes that last over a change in year are split in two?

L143-145. It would be worth saying something about why this upper ocean region is so important e.g. location of most reef forming corals, calcifying phytoplankton etc.

Fig 1. Legend. This second line of this needs clarifying. I guess you're subtracting the ensemble mean change not the ensemble mean. You should say what reference years are used to calculate this ensemble mean change.

L160-165. The methods here are quite convoluted. An illustrative figure highlighting the different steps in the approach would really benefit readers. This could go in the main text or appendices.

L185-186. This suggests something is wrong with the decomposition. At any rate I'm not sure you can call this a decomposition if the separate terms don't sum. How far off summing is the decomposition? If it's not working its value is highly questionable and probably shouldn't be included.

L206-209. Has something like Table A1 been published elsewhere? I would move Table A1 into the main text. It's a nice finding that your observation-based product shows H+ seasonal amplitude increases and variable trends in omega seasonal amplitude. This forms a nice link between this work and the Landschützer et al., 2018 and Kwiatkowski & Orr, 2018 papers.

L239. What is meant by coherently here?

L255-256. Is this true? In Fig 3c/d this appears true under RCP8.5 but the opposite seems to occur under RCP2.6. Indeed the projections at 200m under RCP2.6 are very interesting and quite different across metrics. Any idea why the duration is so much more responsive? Some more detail here would be great. Are there reductions in stratification post 2040 in RCP2.6? I wonder if greater vertical mixing under mitigation might be driving the halt/slight decline in the duration of events and the difference in lags across metrics.

L275-279. More care needs to be made when making these sort of statements as the mean decline in OmegaA has been removed. I would call this variability/extreme variability not extreme events/extreme days.

L299-300. Could this be because the areas of upwelling are moving polewards in the model (see Rykaczewski et al., 2015)? Poleward of these grey regions there appears to be a general increase in extremes, which fits this narrative. Can you check upwelling

or some proxy of this in the model?

L313. Extreme "variability" events would perhaps be more accurate (here and elsewhere in the manuscript).

Section 3.4 As stated above, it's hard to have confidence in the decomposition if it doesn't sum to the model realisation. Maybe the authors would be better to focus on physical processes (upwelling/mixing/ice loss) and how they might explain changes in variability.

L394-395. Some further detail is needed here I think.

Section 4. I recommend dividing this into a few small subsections.

L411-414. But presumably once omega<1 is reached, the ocean spends a greater amount of time undersaturated when this reduced variability is taken into account. Can you comment on this?

Fig 11. This is an interesting figure.

L424-425. This is computationally a big ask. Maybe the community could get by with some daily statistics output at monthly resolution?

L448-449. This seems off topic.

L471. I think you need to clarify some of these definitions. To most people seasonal cycles are a form of sub-annual variability.

Technical comments:

L36. A reference is needed at the end of this paragraph. Some of those already cited in this paragraph would suffice.

L39. The formatting of multiple references here is different to elsewhere in the manuscript.

Fig 1. Legend. Mention that this is for the surface ocean.

L221-224. It would be clearer to discuss model performance at capturing mean seasonal cycles before discussing trends in seasonal cycles.

L272-273. I would rephrase this. 200m is not really the deep ocean.

Fig 5. Labels (extreme days/intensity/duration) on left of this figure would make it easier to read.

336. "during the" preindustrial

References

Cornwall, C. E., Comeau, S., DeCarlo, T. M., Larcombe, E., Moore, B., Giltrow, K., Puerzer, F., D'Alexis, Q. and McCulloch, M. T.: A coralline alga gains tolerance to ocean acidification over multiple generations of exposure, Nat. Clim. Chang., 1–4, doi:10.1038/s41558-019-0681-8, 2020.

Kwiatkowski, L. and Orr, J. C.: Diverging seasonal extremes for ocean acidification during the twenty-first century, Nature Climate Change, 8(2), 141–145, doi:10.1038/s41558-017-0054-0, 2018.

Landschützer, P., Gruber, N., Bakker, D. C. E., Stemmler, I. and Six, K. D.: Strengthening seasonal marine CO 2 variations due to increasing atmospheric CO 2, Nature Climate Change, 8(2), 146, doi:10.1038/s41558-017-0057-x, 2018.

Rivest, E. B., Comeau, S. and Cornwall, C. E.: The Role of Natural Variability in Shaping the Response of Coral Reef Organisms to Climate Change, Curr Clim Change Rep, 3(4), 271–281, doi:10.1007/s40641-017-0082-x, 2017.

Rykaczewski, R. R., Dunne, J. P., Sydeman, W. J., García‐Reyes, M., Black, B. A. and Bograd, S. J.: Poleward displacement of coastal upwelling-favorable winds in the ocean's eastern boundary currents through the 21st century, Geophysical Research Letters, 42(15), 6424–6431, doi:10.1002/2015GL064694, 2015.

---

## Referee Comment (RC2) · James Orr (Referee) · 11 Feb 2020

**General comments:**

This manuscript has the potential to become the first published peer-reviewed study on extreme events in ocean acidification, something that would nicely complement recent studies focused on projected changes in marine heatwaves. The subject is highly relevant for publication in Biogeosciences, the analysis is original, and the authors have clearly devoted considerable effort. Before it can be published though, more work seems needed to make the analysis more accurate and to better communicate these results to the larger community.

For the analysis, my main concern is that in the deconvolution of drivers, the authors'

equation for the Taylor expansion of the variance (Equation 2) is flawed. The bad news is that the units for the first 4 terms on the right-hand side (RHS) do not check. Those units should each be identical to the units for the sole term on the LHS, i.e., the total variance in $(nmol/kg)^2$, but they are not. To have the right units, the sensitivities (partial derivatives) in the first 4 terms would each need to be squared. That modification will change the balance between terms. Given that error, it is not surprising that the authors say that "these contributions can not be separated into summable terms" (line 186). The good news is that the 6 final terms in that equation do have the right units; they are correct as is. Moreover, by making these modifications, the authors should be able to get the terms to add up. With the squared sensitivities, this equation has already been given correctly in previous work such as for uncertainty propagation of $CO_2$ system variables (Dickson and Riley, 1978; Orr et al., 2018) and for analysis of the variance of seasonal to interannual variability (Ericson et al., 2018).

When using the corrected equation, the authors will need to demonstrate quantitatively that all the terms on the RHS add up to the value on the LHS. That could be done very clearly by showing zonal means on the same plot for each RHS term, the sum of all RHS variance terms, and the actual simulated value for the total variance of [H+].

A second flaw with the analysis is that when comparing different contributions, the authors usually compare standard deviations, not variances (e.g., in Figs 7, 8, 9, and A5, and the 4 equations in Appendix C). Although perhaps more intuitive because of the units, comparing the standard deviations of the different components leads to a false impression of relative importance. It is only the variances of the components that linearly add up to the total variance. When minor components are compared in terms of their standard deviations, they appear overly important in ters of their contribution to the total variance. The authors should make all comparisons in terms of variances, not standard deviations.

A third flaw with the analysis is that changes in the mean state (trend) have been removed and seldom enter into the discussion. Because most of the future change

in both [H+] and $\Omega_A$ will be due to changes in the mean state, this neglect leads the authors to make statements that make little sense, such as the following:

L152: "changes in different extreme event characteristics are only caused by variability"

L301-302: "extreme [H+] days are projected to disappear in the RCP8.5 scenario by the end of the century"

L309: "the occurrence of extremes is projected to decrease"

L314: "$\Omega_A$ extreme events are projected to disappear by 2081-2100"

L316-317: "No extreme events are projected for most of the ocean during 2081-2100 under RCP8.5"

Hence there is a communication problem that is directly tied to the authors' peculiar meaning for "extreme event". Unless the authors can bring the mean state back into the picture, they cannot legitimately use the term "extreme event". They are currently focusing only on a diagnostic for changes in variability. To help remedy the problem, I would like to see the authors provide a quantitative analysis of the contributions of each of the temporal components, including the change in the mean state (trend), to the overall change in the maxima. They might be able to do this without repeating their entire analysis procedure, simply by computing the variance due to the change in the mean state and adding that to the total of the other variance contributions that have already been computed. This analysis would clearly demonstrate the dominance of the change in the mean state and properly put the authors' other results into context.

The study of marine heatwaves by Froelicher et al. (2018) included the change in the mean state as part of the analysis, so it is even more unclear to me as to why the same was not done here for the analysis of extreme events in [H+] and $\Omega_A$. Moreover, the relative importance of the change in the mean state relative to other temporal fluctuations (subannual, monthly, interannual) seems to be much larger for [H+] and $\Omega_A$ relative to SST. Its prominent role needs greater emphasis in this study by Burger et al.

**Specific comments:**

L2: Please define what is meant by by "acidity". Acidity is a term used by aquatic chemists to refer to base neutralizing capacity, just as alkalinity is used to refer to acid neutralizing capacity. Acidity does not a priori refer to [H+] just as alkalinity does not refer to [OH-]. The meaning of the authors ([H+]) differs and has to be defined. If acidity is to be used, I would be more comfortable with the term "free acidity" when referring only to [H+] because that is only part of the total acidity that is not bound up in other ions that react with [OH-].

L3: It is unclear what is meant by "mean ocean acidification". Ocean acidification involves changes in many $CO_2$ system variables simultaneously.

L28: Please define "short term". Also add a hyphen (see Global changes section below).

L61-62: The sentence refers to changes in [H+] and $\Omega_A$ and cites references most of which only addressed the seasonal cycle of $pCO_2$. Please separate the references so that the readers know which one(s) actually addressed [H+] and $\Omega_A$.

L76: "daily output" is ambiguous. Please say "daily mean output" if that is what you mean. If more frequent, say something like "6 hourly output". This is an important point because the analysis does not seem to include the potentially large diurnal cycle if it is based on daily mean output.

L77: It seems ambiguous to write about "daily variability". That could be misunderstood by readers to mean "diurnal variability", which the authors did not consider because they are presumably using daily mean output.

L79-81: Please delete the last 3 lines. It is much nicer to end the Introduction with the aim of the study. Moreover, subsequently diluting the aim by following that with an outline of the paper makes no sense, particularly when that outline follows the standard IMRAD format (Introduction, Methods, Results, and Discussion) which is what is

expected anyway.

L98: I suspect that the GFDL-ESM2M model does not simply use the K1 and K2 directly from Mehrbach et al. (1973), unlike what is stated, but rather the K1 and K2 from the Mehrbach data after being converted to the total hydrogen scale by either Dickson and Millero (1987) or Lueker et al. (2000). Please clarify. Another important point that is not mentioned which concerns the model's total alkalinity equation. Does the total alkalinity equation in the GFDL-ESM2M model include contributions from phosphoric and silicic acid systems. Many models do not, and this can bias $pCO_2$ in the high latitudes by 10 ppm or so. [H+] would also be affected.

L100: This mention of diurnal variability seems irrelevant if the authors are using only daily mean output, which seems to be the case. Please delete.

L140: The mention of the software seems too vague. What function was used from the library?

L148-149: It is not clear to me why each ensemble member is detrended with the ensemble mean trend rather than the trend in the individual ensemble member. I understand that it is easy conceptually just to calculate the ensemble mean trend and use that. But one could also use a spline to detrend each ensemble member. Anomalies relative to the ensemble mean trend will be larger than those relative to the individual member trend. They would contain differences due to the different trends in the ensemble members as well as differences due to the other variability.

L151: Please reword. It is not that the mean state is constant. Rather, the trend was removed.

L159: It says that both the standard deviation and variance are used. As mentioned in the general comments, I think that the comparison should be made only with the variance.

Figure 1: caption - change "subtracted" to "removed". If the ensemble mean were

simply subtracted, the mean of the preindustrial period would be zero rather than 7.3. Using "removed" gives more leeway in the meaning.

L164: "daily variability" is a confusing term. It could be taken to mean "diurnal variability" by some readers. I think the authors should stick with the term "subannual" and define that carefully as meaning "with the seasonal cycle removed but ignoring diurnal variability."

L166: It is dangerous to simply compare contributions from the different standard deviations, which do not add up linearly and give a biased impression of their contributions to the total variability. Rather, use the variance of the different components, which add linearly to make up the total variance.

L168: Please clarify what is meant by "changes".

L172: Equation 1 does not seem to be used. Please just delete it and remove any mention of it in the text.

L182: Equation 2 is wrong. In the first 4 terms, the partial derivatives should be squared. This fix will change the balance of their contributions.

L186: Since your Eq. 2 is wrong, it is not surprising that the terms do not sum up to match the total simulated variance. This match is key if we are to believe the deconvolution analysis.

L209: Table A1 should be brought into the main body of the text and given as Table 1.

L215-221:

- It is unclear how trends and the error bars were computed for the data based estimates

- What is the statistical significance of the model vs. data-based comparisons? In some cases, they do not appear to differ statistically, e.g., for $\Omega_A$ in the northern

high latitudes.

L229: Is there not some question about the data-based estimate of seasonal variability in the Southern Ocean where winter data is sparse?

L230: I would recommend being more cautious, by saying something like "it appears that the GFDL-ESM2M model is adequate to assess..."

L238: What is meant by intensity?

L239: Table A2 should be brought into the main body of the paper and given as Table 2. This would allow the authors to simplify the text to mention only the mean changes and avoid giving the uncertainty ranges in parentheses). That simplification would make reading easier. Alternatively they could just give the ranges and not the mean changes. More generally, the first 4 paragraphs of this section (3.1) suffer from trying to cram too much information into every place where any numerical fact is mentioned. Writing complex things such as "0.20 nmol kg-1 (0.19-0.21 nmol kg-1; 18

L240: Please tell us what fraction of the total volume above 200 m is represented by the 2.7 x$10^3 km^3$.

L254-255: "is connected to" is vague. The increased contribution of interannual variability is not a mechanistic explanation. Previous work has discussed why changes in [H+] in the subsurface are larger than those at the surface (Orr, 2011; Resplandy, 2015). Perhaps their chemical explanation explains the longer duration of extreme variability mentioned here as well.

L270-271: Given the large uncertainty between ensemble members, can the authors really say that there is a statistically significant increase in extreme days per year, intensity, and volume of events? I see no subplot for the Volume for the 200-m analysis.

Section 3.2 and 3.3: There are many statements particularly in these sections that make no sense because when addressing "extreme events", one must include the

dominant contribution from changes in the mean state.

Section 3.4: This section will need to be completely rewritten after the fixing the first 4 terms in Eq. 2 and reevaluating the associated contributions. In addition that analysis should include a more quantitative assessment than just showing maps of the different contributions. A comparison of zonal means would be a good start. One could also compare contributions to the variance with stacked bar charts with different parts of each bar indicating different temporal components, similar to those shown in Kwiatkowski and Orr (2018).

L390: It is not clear what "nonlinear dependence" refers to. The authors use a 1st order Taylor expansion, which is linear in terms of the variances.

L396-405: The comparison of the results for the 99th and 99.9 percentiles of "extremes" is also confused by the general neglect of the effect of the mean state, which is dominant. This paragraph should be rewritten with that in mind or otherwise deleted.

L410-415: There seems a missed opportunity to compare the authors' results in terms of the effect of changes in variability to those from McNeil and Matear (2008) who assumed that seasonal variability was unchanged but advanced the time at which the $\Omega_A$=1 threshold was reached relative to previous work that did not address seasonal variability.

L416-418: The discussion about arbitrary thresholds for [H+] is too vague. More details would be needed if it is to be kept.

L424-425: The text states, "It is therefore critical to uses daily temporal output to assess extreme events in ocean biogeochemistry." I believe that this statement tends to oversell the importance of the variability of the daily-mean output variability. Currently the authors results indicate that daily-mean variability of [H+] and $\Omega_A$ is generally a minor, second-order concern in surface waters. That is actually good news for future analysis, because saving daily mean output is not feasible on a routine basis, especially

considering the many CMIP models and scenarios.

On the other hand, the contribution to the total variance from diurnal variations, a component that the authors do not quantify, may be much larger especially in coastal regions. An assessment of that variability in the observations and in high-resolution models should remain a priority.

L447-449: The discussion about the "partial" diurnal cycle in TOPAZv2 (GFDL-ESM2M) does not seem relevant because by using daily means, the authors do not assess diurnal variability. This sentence should simply be deleted. It will only confuse the reader.

L455-456: It seems misleading to lead off this sentence with "While coastal species may be adapted to large variability in ocean acidity...". Although already large variability of [H+] is seen in coastal regions, that variability will also grow as atmospheric $CO_2$ continues to invade the ocean and coastal-water buffer capacities also decline.

L464-465: As written, this first statement in the final paragraph was known already before this study was undertaken based on previously published papers concerning the future increase in the mean state and the future growth in seasonal variations of [H+]. This new study confirms those findings. It should not say "our analysis reveals", although it could say something like it "confirms previous findings ..."

L467: What is "mean" ocean acidification?

**Technical concerns:**

Global changes:

- change "200 m depth" to "200 m"

- change "point in time" to "time"

- change "points in time" to "times"

- MISSING HYPHENS: the terms "short term", "deep water", and "sea ice" should all include a hyphen when they modify a noun

L38: change "The vast majority of" to "Most of the". Delete "so far"

L44:

- delete "one might expect that".

- change "exhibit" to "have"

- It is not "carbonate chemistry" that will cross critical thresholds. That is like saying "physics will cross thresholds". You could instead say "key $CO_2$ system variables"

L46: change "negatively impacted" to "adversely affected".

L49: change "undergo a decline in" to "exhibit reduced"

L63:

- change "suggest" to "project".

- change "is projected to" to "will"

L69:

- change "extreme" to "extremes"

- Please separate citations so that we know which ones address heatwaves and which ones address sea-level rise.

L78: change "imprint on the occurrence" to "affect"

L84: change "are performed" with "were made"

L104:

- change "1000 year" to "1000-year".

- "a new computing infrastructure" is vague. Please clarify your intended meaning.

L108: insert "the average" before "atmospheric"

L111: insert "average" before "atmospheric"

L120:

- change "500 year" to "500-year"

- delete "long"

- what is "potential vegetation"

L121: make the same first 2 changes as in the previous line but for "220 year long"

L124:

- write "daily mean" instead of "daily"

- add "the aragonite saturation state" before the symbol $\Omega_A$

- delete the 2nd sentence

L202: change "is" to "appears to be"

L238: change "typical" to "average"

[Figure]

L266: delete "blue lines in". That color information should only be given in the Figure or its caption.

Figure 2: The choice of colors for the lines could be improved. It is hard to distinguish light green vs. dark green and blue vs. purple. Yellow is difficult to see on a white background. Are these colors good for colorblind people?

Figure 3: There is too much white space and repeated information. To remedy these problems, I'd suggest to start by labeling the subpanels as a matrix, i.e., with row labels and column labels. Thus you could remove the title for each subpanel and use that as the row label of the matrix. The row labels should only be given on the left (the connection with the right column is obvious). And column headings "Surface" and "200 m" should only be given at the top. The numbers on the tick marks can be kept for all subpanels, but the x-axis label should only be kept on the bottom row. The current y-axis labels should all be removed. The same approach should be taken for all subsequent figures with more than 2 subplots. The figures will then be more compact, less verbose and redundant, and readers will intuitively understand the setup of your multi-panel figures (without going to the figure caption, repeatedly).

L283-284: delete "but show distinct spatial patterns" (redundant).

L295: the meaning of "reoccur" is unclear.

L421: Please provide separate citations for the papers that addressed $pCO_2$ and those that addressed [H+].

L476: I like the choice of the authors in Appendix A to use "subannual" instead of "daily" variability. Please do the same in the main body of the paper.

Appendices: Please provide an equation number for each equation listed in the appendices.

Appendix C: The equations should be written in terms of variances rather than standard deviations.

Tables A1 to A3 do not follow the formatting standards used by Biogeosciences. See the author guidelines and previously published BG papers.

Colorbar: On all the maps with diverging scales, there seems to be an excessive amount of yellow and it is difficult to assess where the "zero" line is. A better choice would be a color bar with very pale blue and very pale red next to each other in the center of the color distribution.

**References**

Dickson, A.G., Millero, F., 1987. A comparison of the equilibrium constants for the dissociation of carbonic acid in seawater media. Deep-Sea Res. Vol. 34, 1733–1743.

Dickson, A.G., Riley, J.P., 1978. The effect of analytical error on the evaluation of the components of the aquatic carbon-dioxide system. Mar. Chem. 6, 77–85.

Ericson, Y., Falck, E., Chierici, M., Fransson, A., Kristiansen, S., Platt, S. M., et al., 2018. Temporal variability in surface water $pCO_2$ in Adventfjorden (West Spitsbergen) with emphasis on physical and biogeochemical drivers. J. Geophys. Res.: Oceans, 123, 4888–4905. doi:10.1029/2018JC014073

Frölicher, T. L., Fischer, E. M., Gruber, N. (2018). Marine heatwaves under global warming. Nature, 560(7718), 360-364.

Kwiatkowski, L., and Orr, J. C. Diverging seasonal extremes for ocean acidification during the twenty-first century, Nature Climate Change 8, 2, 141–145, doi:10.1038/s41558-017-0054-0, 2018.

Lueker, T.J., Dickson, A.G., Keeling, C.D., 2000. Ocean $pCO_2$ calculated from dissolved inorganic carbon, alkalinity, and equations for K1 and K2: validation based on laboratory measurements of $CO_2$ in gas and seawater at equilibrium. Mar. Chem. 70, 105–119.

McNeil, B. I., Matear, R. J., 2008. Southern Ocean acidification: A tipping point at

450-ppm atmospheric $CO_2$. Proc. Nat. Acad. Sci., 105(48), 18860-18864.

Orr, J. C. Recent and future changes in ocean carbonate chemistry, In: Gattuso, J.-P. and Hansson, L. (eds.), Ocean Acidification, Oxford Univ. Press, 41–66, 2011.

Orr, J. C., Epitalon, J. M., Dickson, A. G.,  Gattuso, J. P., 2018.  Routine uncertainty propagation for the marine carbon dioxide system. Marine Chemistry, 207, 84-107.

Resplandy, L., Bopp, L., Orr, J. C.,  Dunne, J. P., 2013. Role of mode and intermediate waters in future ocean acidification: Analysis of CMIP5 models. Geophys. Res. Lettr., 40(12), 3091-3095.

---

## Referee Comment (RC3) · Sarah Schlunegger (Referee) · 29 Feb 2020

Burger et. al. presents a clear and logical assessment of projected changes in the variability of ocean acidity and their impact on acidic extremes. The study uses an Earth System Model (ESM) which they demonstrate has historical fidelity with observed acidification trends. The study contrasts future acidity extremes under low- and high-emissions scenarios, all relative to 99th percentile extremes as defined in the preindustrial control simulation. Changes in the duration, frequency, intensity and volume extent of extreme events are presented. Drivers of change in [H+] variability are partitioned into contributions from changes in the mean state and variability of carbon concentrations, temperature, alkalinity and salinity. As noted by other reviewers, the exact decomposition used is flawed and needs revision before publication.

Other than this obvious issue, I have only three broad suggestions which would ready the manuscript for publication in Biogeosciences.

1. As a first assessment of changes in OA extremes, this paper has the opportunity to present simple, conceptual explanations for why those changes are occurring. This is done adequately for the case of [H+] but not for the case of Omega. Why is Omega variability decreasing? Is this largely driven by changes in carbonate chemistry, the way it is for [H+], or are changes in ocean dynamics involved? This has implications for the robustness of the results across other ESMs.

2. Further discussion is needed of why variability changes at the surface differ from those at 200m depth and of why there is strong compensation between the contributions of increasing carbon concentrations and decreasing carbon variability to [H+] variability at depth (9e vs 9f, and to a lesser extent at the surface, 8e and 8f). The striking spatial patterns in Figure 9e and 9f should also be explained.

3. Include discussion of model uncertainty. For example, results that are a direct chemical consequence of invading anthropogenic carbon (e.g. increasing sensitivity of [H+] to drivers) are more likely to be more robust across models than results that are a consequence of changes in ocean physics. ESM2M has less warming and a stronger ocean carbon sink than the majority of CMIP5 models – how might the results presented compare to a model with higher climate sensitivity and less ocean carbon uptake models?

The manuscript is a significant contribution to the field, addresses the pressing issue of ocean acidification, and does so with excellent visualization and explanation. I strongly recommend publication (after the listed revisions are implemented).

————————-

Additional comments and stylistic suggestions

————————-

L5: number of days where? At each point? On average? Overall occurrence any-where?

L28: replace 'or' with 'and/or'

L33: define saturation horizon – e.g. Saturation generally decreases with increasing depth, with transition from saturated to undersaturated referred to as the saturation horizon.

L33: add 'can' in front of 'also' . . . 'can also'

L38: remove 'vast'

L40: add 'also' in front of necessary. . . 'also necessary'

L45: . . .'critical threshold, like undersaturation.'

L50: sentence rearrangement for clarity: 'In laboratory experiments in which deep-water corals are exposed to low-pH waters for a week. . .'

L53: Start sentence with 'Therefore,'

L61: '. . .at which higher background concentrations of dissolved inorganic carbon (DIC or C_T) and warmer temperatures produce stronger departures from mean state val-ues for a given change in pertinent physical or chemical drivers.'

L103: Detail about spin-up and control is distracting.

L110: omit 'and', replace with comma

L120: 'year-long' – however these detail is distracting so potentially remove all together.

Figure 1.

» The figure caption (and methods) say that the ensemble mean is removed – however what is removed is actually the departure of the ensemble mean from the preindustrial state. If the ensemble mean itself were removed, then all values at year 1861 would be

~zero. Update description or replot.

» Panel b is presumably derived from panel a, however it is confusing that the variability in panel a contracts during the 21st century, however it is shown to increase in panel b. Please add an explanation in the caption or text.

L230: remove 'we consider'

L240-245: Further and continued clarification that values given are a global average of local changes. With each presentation of a value, indicate it is a global average.

L244: change 'single events' to 'individual events'

L255: Why is high-frequency variability larger at the surface whereas low-frequency variability larger at depth? ... Presumably because the surface has direct contact with the atmosphere, and its chaotic, high-frequency (weather) variability, whereas the deeper ocean experiences stronger low-frequency variability as the ocean acts as a low pass filter on the atmospheres' stochastic forcing (e.g. Hasselmann 1976). Add/include explanation.

L298: insert 200 meters... ' In contrast to the surface, 200 meter [H+] extremes'...

L357-361: $C\_T$ increases due to invasion of anthropogenic carbon. But why does ALK change? ALK is not influence by gas-exchange, but by biology or circulation. Briefly explain ALK changes and give reference. 378-379: [H+] variability changes at depth are larger than at surface – is this mostly due to the fact that background $C\_T$ concentrations increase with depth, however increased $C\_T$ itself is associated with decreased susceptibility to external variability? Figure 8 e) vs f) indicates this is the case. Provide commentary.

Figure 9.

» Panel 9b and 9c bear strikingly similar patterns and magnitudes – where the changes in the mean-state of the drivers is largest and positive, changes in the variability are

also largest and negative. And this is mostly coming from C_T mean (increasing) and C_T variability (decreasing). Is this directly understandable through carbonate chemistry? In a higher DIC world, DIC variations are actually smaller? Or is it that the physical, surface to deep gradient in DIC is weakening with surface invasion of anthropogenic carbon and therefore mixing-related variability is reduced?

» Explain this near-perfect anti-correlation.

» Also, the structures of strong mean and variability changes look dynamical. Potentially the expansion of the subtropical gyres. Explain the spatial structure of the patterns of strong drivers.

» In the appendix, include figures of the pre-industrial and year 2100 concentrations of C_T at surface and 200m depth. The patterns in 9e and 9f are likely related to both the starting concentrations of C_T and the storage of C_T. Figure A4a and A4e show the change in C_T, which resembles the strong patterns of 9e and 9f. Synergies of anthropogenic carbon and background concentrations of C_T may help explain strong structures of 9e and 9f. The strongest temperature changes (A4g) also resemble the strong patterns of 9e and 9f, indicating either dynamical changes or thermal changes are involved.

Line 393: Is the decrease in Omega variability a direct consequence of carbonate chemistry? Or of changes in the variability of the drivers or physical state (e.g. vertical gradient hypothesis stated before)? This is an important distinction, as carbonate chemistry is well constrained (little model uncertainty) whereas dynamical changes have much larger model uncertainty. If reductions in Omega variability are chemically deterministic, then this feature will be relatively robust across models, as most use the same (e.g. OCMIP2, Najjar et al. 2007) protocols for carbonate chemistry calculations. But if it is dynamical, it may not be. Furthermore, if it is chemically driven, and if it is related to the storage of anthropogenic carbon and/or concentration of natural carbon, then contemporary observations and reconstructions of the distribution of natural and

anthropogenic carbon in the ocean could reveal where changes in Omega variance are likely to occur.

————————-

References

————————-

Hasselmann, K. 1976. "Stochastic Climate Models Part I. Theory, Tellus" 28 (6): 473–85. https://doi.org/10.3402/tellusa.v28i6.11316.

Najjar, R G, X Jin, F Louanchi, O Aumont, K Caldeira, S C Doney, J C Dutay, et al. 2007. "Impact of Circulation on Export Production, Dissolved Organic Matter, and Dissolved Oxygen in the Ocean: Results from Phase II of the Ocean Carbon-Cycle Model Intercomparison Project (OCMIP-2)." Global Biogeochemical Cycles 21 (3): n/a–n/a. https://doi.org/10.1029/2006GB002857.

———————————————

---

## Author Comment (AC1) · 21 Apr 2020

Dear reviewer,

attached you can find our reply to your comments on the manuscript. The document also includes the applied changes in the manuscript. We thank for the positive and encouraging feedback.

Best regards, Friedrich Burger

Please also note the supplement to this comment:
https://www.biogeosciences-discuss.net/bg-2020-22/bg-2020-22-AC1-supplement.pdf

---

## Author Comment (AC2) · 21 Apr 2020

**Increase in ocean acidity variability and extremes under increasing atmospheric CO$_2$**
**Response to reviewers' comments**

Friedrich A. Burger, Thomas L. Frölicher, and Jasmin G. John

March 2020
* * *
**Reviewer 1**

**General comments**:

This is a nice manuscript that assesses extreme chemistry variability in ensemble projections of an Earth system model. The manuscript is both interesting and timely.

→ We thank the reviewer for the positive and encouraging feedback.

My comments mainly relate to improvements I think the authors could make in understanding the different drivers of carbonate chemistry variability. Particularly, I'd like to see more on the physical processes driving the differences between the RCPs and the projected frequency/intensity/duration of extreme variability events.

→ In this study, we show for the first time how extreme variability events change under increasing CO$_2$ and we also identify the individual drivers of changes, i.e. changes in temperature, salinity, carbon and/or alkalinity. An in-depth quantification of the physical processes that cause the changes in the individual drivers would clearly be an interesting additional analysis, but this is beyond the scope of this study. Nevertheless, we have now addressed some particular questions on physical processes driving the extremes events, such as that raised for lines 255-256, and we have added a sentence to the discussion section highlighting that an analysis of the physical processes would be an important next step.

Another general issue I think is how the authors choose to define extreme events. I would be more comfortable calling these "extreme variability events" given that mean trends have been removed. This is particularly an issue when they discuss saturation state and often give the impression that extreme events are projected to decline.

→ We thank the reviewer for this suggestion. We have changed throughout the manuscript "extreme events" to "extreme variability events". We also added a new Figure 11 that compares the changes in extreme events due to changes in variability with changes in extreme events that are caused by both changes in variability as well as changes in long-term ocean acidification.

Finally, given that the carbonate chemistry decompositions apparently do not sum, I'm not convinced of their value. I suggest removing this analysis if it can't be properly validated.

→ We think that our statement on lines 185-186 "Unfortunately, these contributions can not be separated into summable terms because [H$^+$] standard deviation is a nonlinear function of those." may have led to confusion. Therefore, we clarify our decomposition approach in the following:

Variance of [H$^+$] as a function of the sensitivities (partial derivatives), standard deviations, and correlations of the drivers involves terms of the form $s_i s_j \sigma_i \sigma_j \rho_{ij}$, so products of five variables ($s_i$ denotes the sensitivity with respect to variable $i$ here. $i$ could be for example C$_T$ and $j$ could be for example A$_T$).

Because of this large degree of nonlinearity, one can not confidently estimate changes in such a product by applying a first-order Taylor expansion on it. To state this was the intention in lines 185-186. Instead we added changes in sensitivities, standard deviations, and correlations in three steps and analyzed how much of the total variability change can be explained additionally by e.g. also taking into account variability changes on top of the sensitivity changes. This procedure included the nonlinearities in Equation 2 by construction and was thus exact. We have now streamlined the text to make the method more transparent.

However, in response to reviewer 3, we have also extended our approach and decompose variance change by applying a full fifth-order Taylor decomposition on Equation 2. While the previous approach did not allow to track how much variability change in $[\text{H}^+]$ arises from variability changes in the drivers alone and how much from the interaction between sensitivity and variability changes, the new approach allows to quantify this aspect. The new decomposition is exact since we take into account all non-vanishing orders of the Taylor series. We then group these terms of the Taylor series into four groups: (1) contributions of sensitivity changes to the overall change in variance, (2) contribution from standard deviation changes in the drivers, (3) simultaneous changes in the sensitivities and standard deviations (these can neither be attributed to sensitivity changes nor to variability changes alone), and (4) all terms that include changes in the correlations. The method is explained in the revised section 2.2.3 and in more detail with the full decomposition in appendix C.

Nevertheless, the results of this decomposition are consistent with the previous one in the sense that (1) is the same as 'All Means' in previous Figures 8+9, (2)+(3) (the sum) is identical to 'All Variabilities', and (4) is identical to 'Phasing' in previous Figures 8+9. The partitioning of 'All Variabilities' into (2) and (3) allows to understand more accurately what role variability changes in the drivers play for $[\text{H}^+]$ variance.

In response to a comment by reviewer 3, we now also decomposed variance changes for $\Omega_A$ and added a plot showing the zonal mean decomposition (new Figure 10).

In summary, we have revised method section 2.2.3, results section 3.4 and appendix C to clarify our carbonate chemistry decomposition approach.

**Specific comments:**

L21-23. This could be better explained. I suggest a sentence or two more, including a full definition of omega.

$\rightarrow$ We now write: 'The rise in $[\text{H}^+]$ is partially buffered by the formation of $[\text{HCO}_3^-]$ from $[\text{CO}_3^{2-}]$. The associated decline in $[\text{CO}_3^{2-}]$ reduces the calcium carbonate saturation state $\Omega = [\text{Ca}^{2+}]\,[\text{CO}_3^{2-}]/\left([\text{Ca}^{2+}]\,[\text{CO}_3^{2-}]\right)_{\text{sat}}$, i.e. the product of calcium and carbonate ion concentrations relative to the product at saturation. Undersaturated waters with $\Omega < 1$ are corrosive for calcium carbonate minerals.'

L43. I don't think Hofmann et al., 2011 is really relevant here as they don't assess organism adaptation/acclimation under variable chemistry regimes. Many other papers do, with mixed findings, for example see: (Rivest et al., 2017; Cornwall et al., 2020)

$\rightarrow$ Thank you for these relevant references. We now cite Rivest et al. (2017) and Cornwall et al. (2020).

L45-48. The authors are being too concise here. Explain what you mean by undersaturation. I understand that you might have to refer to aragonite versus calcite but that should probably be already mentioned anyway.

$\rightarrow$ We changed 'undersaturation' to 'aragonite undersaturation' and added additional explanation above on different calcium carbonate minerals and their saturation states, also introducing aragonite and calcite.

L77. Its not very clear what "residual" means in this context, as it hasn't been defined yet.

$\rightarrow$ Following the recommendation by reviewer 2, we renamed 'residual daily variability' to 'subannual' variability throughout the manuscript.

L90. What is the depth of the first ocean level? This will useful to know, as it will be a major determinant

of surface variability.

→ The MOM4p1 model has a free surface and the depth of the first ocean grid level is centered at around 5 m. We clarified: 'The MOM4p1 model has a free surface and the surface level is centered around about 5 m depth and the spacing between consecutive levels is about 10 m down to a depth of about 230 m (Griffies,2009).'

L120. What is potential vegetation? Are you referring to a coupled terrestrial carbon cycle? It is not clear what relevance this has.

→ Potential vegetation refers to a terrestrial carbon cycle setup without land-use change. However, we agree with the reviewer that this is not relevant in the context of the paper and we deleted this additional information.

L142. Do you mean that extremes that last over a change in year are split in two?

→ Yes, this is correct. No changes are made to the manuscript.

L143-145. It would be worth saying something about why this upper ocean region is so important e.g. location of most reef forming corals, calcifying phytoplankton etc.

→ We added a sentence to motivate our choice: "We focus our analysis not only on the surface, but also on 200 m depth to study changes in extreme events within the thermocline, where most organisms susceptible to ocean acidification are found, such as reef-forming corals and calcifying phytoplankton."

Fig 1. Legend. This second line of this needs clarifying. I guess you're subtracting the ensemble mean change not the ensemble mean. You should say what reference years are used to calculate this ensemble mean change.

→ Thank you for the pointer. We have modified the sentence to: '(b,d) Same as (a,c), but the ensemble-mean change with respect to the average of the 500-year long preindustrial control simulation has been subtracted.'

L160-165. The methods here are quite convoluted. An illustrative figure highlighting the different steps in the approach would really benefit readers. This could go in the main text or appendices.

→ While we agree that this is a good idea in principle, we prefer not to add another figure since we already added two additional figures in response to comments by reviewers 2 and 3 and prefer to keep the number of figures at a reasonable level. However, we clarified that we used three steps to assess whether changes in low or high frequency variability cause changes in extreme variability events and their characteristic.

L185-186. This suggests something is wrong with the decomposition. At any rate I'm not sure you can call this a decomposition if the separate terms don't sum. How far off summing is the decomposition? If it's not working its value is highly questionable and probably shouldn't be included.

→ We have revised the carbonate chemistry decomposition. Please see more details above.

L206-209. Has something like Table A1 been published elsewhere? I would move Table A1 into the main text. It's a nice finding that your observation-based product shows $[H^+]$ seasonal amplitude increases and variable trends in omega seasonal amplitude. This forms a nice link between this work and the Landschützer et al., 2018 and Kwiatkowski & Orr, 2018 papers.

→ We thank the reviewer for this comment and we agree that this has, to our knowledge, not been published elsewhere yet. Reviewer 2 has made a similar comment. We have moved the Table from the Appendix into the main text.

L239. What is meant by coherently here?

→ We clarified in the method section the calculation of the volume of individual extreme events: 'mean volume of clusters of connected grid cells that are above the $99^{th}$ percentile'. We therefore deleted the word coherently here.

L255-256. Is this true? In Fig 3c/d this appears true under RCP8.5 but the opposite seems to occur under

RCP2.6. Indeed the projections at 200m under RCP2.6 are very interesting and quite different across metrics. Any idea why the duration is so much more responsive? Some more detail here would be great. Are there reductions in stratification post 2040 in RCP2.6? I wonder if greater vertical mixing under mitigation might be driving the halt/slight decline in the duration of events and the difference in lags across metrics.

→ Yes, the changes (in a relative sense) are larger at the surface than at 200m depth both for the RCP8.5 and for RCP2.6, as can be seen from Table A2 (now in the main text as Table 2). We clarified this by adding the word 'relative': "However, projected relative changes over the historical period and the 21$^{st}$ century are smaller at 200 m than at surface...". Global mean event duration at 200m depth is declining under RCP2.6 in the second half of the 21$^{st}$ century mainly due to the decrease in event duration in the subtropics. There, the reduction in duration is connected to reductions in the contribution from interannual variability to total variability and partially also to reductions in total variability. We have added this to section 3.1 and write there: "This decrease in duration mainly occurs in the subtropics, where events generally last long (Figure A2b). It is connected to an increase in the contribution from high-frequency variability to total variability in those regions over that period.". We think that the mechanism proposed by the reviewer (increased vertical mixing under mitigation leading to increased high frequency variability at subsurface and thereby to reduced event duration) is interesting. However, a complete analysis of the physical processes behind the changes in duration under RCP2.6 is beyond the scope of the study. We added a sentence to the discussion section that further studies are needed which focus on the physical processes.

L275-279. More care needs to be made when making these sort of statements as the mean decline in OmegaA has been removed. I would call this variability/extreme variability not extreme events/extreme days.

→ We have changed the notation of extreme events to extreme variability events throughout the manuscript and we also added a clarifying sentence in this section: "It should be noted that, despite this decline in extreme variability events, the long-term decline in the mean state of $\Omega_A$ still leads to more frequent occurrence of low values in $\Omega_A$ (see Discussion section)"

L299-300. Could this be because the areas of upwelling are moving polewards in the model (see Rykaczewski et al., 2015)? Poleward of these grey regions there appears to be a general increase in extremes, which fits this narrative. Can you check upwelling or some proxy of this in the model?

→ Except maybe in the California Current Region, Figure 5b does not show a general increase in extremes polewards of the EBUS. No changes are made to the manuscript.

L313. Extreme "variability" events would perhaps be more accurate (here and elsewhere in the manuscript).

→ We agree and changed it throughout the MS. Please also see reply above.

Section 3.4 As stated above, it's hard to have confidence in the decomposition if it doesn't sum to the model realisation. Maybe the authors would be better to focus on physical processes (upwelling/mixing/ice loss) and how they might explain changes in variability.

→ Section 3.4 has been revised. Please see above for detail. Although we agree that the investigation of the underlying physical processes is interesting, we believe that it is out of the scope of the present study. We added a sentence in the discussion section stating that there is more work to do on the underlying physical processes.

L394-395. Some further detail is needed here I think.

→ We rewrote the sentence to "Extreme variability events in $\Omega_A$ are projected to become less frequent in the future. It is because $\Omega_A$, unlike [H$^+$], becomes less sensitive to variations in the drivers with the mean increase in C$_T$. Furthermore, the projected reductions in the drivers' variabilities, mainly in C$_T$, significantly add to the reduced occurrence of $\Omega_A$ variability extremes."

Section 4. I recommend dividing this into a few small subsections.

→ Many thanks for the suggestion. However, we decided to keep the structure of the discussion and conclusion section as is.

L411-414. But presumably once omega<1 is reached, the ocean spends a greater amount of time undersaturated when this reduced variability is taken into account. Can you comment on this?

→ This is certainly true. We now discuss this aspect in the new discussion paragraph about extreme events defined by a fixed baseline. We write: "Interestingly, the GFDL ESM2M projects that surface mean $[H^+]$ overshoots the preindustrial $99^{th}$ percentile in year 1975 on global average. Thereafter, higher variability actually reduces the number of extreme event days that are above the preindustrial percentile. Surface mean $\Omega_A$ falls below the preindustrial $1^{st}$ percentile in year 1990. After that, lower variability further increases the number of extreme event days below the preindustrial percentile."

Fig 11. This is an interesting figure.

→ Many thanks!

L424-425. This is computationally a big task. Maybe the community could get by with some daily statistics output at monthly resolution?

→ Our analysis shows that the average duration of an extreme $[H^+]$ event at surface is about 10 days at preindustrial and 15 days at present-day. It can therefore not be represented by monthly data. We modified the text to: "In addition to earlier studies, we also show that changes in subannual variability contribute to changes in extreme $[H^+]$ variability events under increasing atmospheric $CO_2$ and that the average duration of extreme variability events at the surface and at present-day is about 15 days. It is therefore critical to use daily temporal output to assess extreme events in ocean acidity."

L448-449. This seems off topic.

→ We agree and removed the sentence.

L471. I think you need to clarify some of these definitions. To most people seasonal cycles are a form of sub-annual variability.

→ We added a clarification on the definition of the seasonal cycle: 'the seasonal cycle, here defined as the 365-day long mean evolution over the course of a year'

**Technical comments:**

L36. A reference is needed at the end of this paragraph. Some of those already cited in this paragraph would suffice.

→ We added Hofmann et al., 2011.

L39. The formatting of multiple references here is different to elsewhere in the manuscript.

→ We fixed the formatting issue.

Fig 1. Legend. Mention that this is for the surface ocean.

→ Mentioned at beginning of caption:' Simulated daily surface $[H^+]$ (a) and $\Omega_A$ (c)'

L221-224. It would be clearer to discuss model performance at capturing mean seasonal cycles before discussing trends in seasonal cycles.

→ Following the recommendation, we changed the order and now discuss the mean seasonal cycles before discussing the trends.

L272-273. I would rephrase this. 200m is not really the deep ocean.

→ Changed to 'surface-to-deep ocean transport" to "surface-to-subsurface transport'

Fig 5. Labels (extreme days/intensity/duration) on left of this figure would make it easier to read.

→ We have followed the reviewers recommendation, also for the analogous supplementary figure A1.

336. "during the" preindustrial

→ Changed.
* * *
**Reviewer 2: James Orr**

**General comments**:

This manuscript has the potential to become the first published peer-reviewed study on extreme events in ocean acidification, something that would nicely complement recent studies focused on projected changes in marine heatwaves. The subject is highly relevant for publication in Biogeosciences, the analysis is original, and the authors have clearly devoted considerable effort. Before it can be published though, more work seems needed to make the analysis more accurate and to better communicate these results to the larger community.

→ We thank the reviewer for this positive statement and the careful and detailed review. We appreciate the suggestions on how to improve the manuscript.

For the analysis, my main concern is that in the deconvolution of drivers, the authors' equation for the Taylor expansion of the variance (Equation 2) is flawed. The bad news is that the units for the first 4 terms on the right-hand side (RHS) do not check. Those units should each be identical to the units for the sole term on the LHS, i.e., the total variance in (nmol/kg) 2 , but they are not. To have the right units, the sensitivities (partial derivatives) in the first 4 terms would each need to be squared. That modification will change the balance between terms. Given that error, it is not surprising that the authors say that "these contributions can not be separated into summable terms" (line 186). The good news is that the 6 final terms in that equation do have the right units; they are correct as is. Moreover, by making these modifications, the authors should be able to get the terms to add up. With the squared sensitivities, this equation has already been given correctly in previous work such as for uncertainty propagation of CO2 system variables (Dickson and Riley, 1978; Orr et al., 2018) and for analysis of the variance of seasonal to interannual variability (Ericson et al., 2018). When using the corrected equation, the authors will need to demonstrate quantitatively that all the terms on the RHS add up to the value on the LHS. That could be done very clearly by showing zonal means on the same plot for each RHS term, the sum of all RHS variance terms, and the actual simulated value for the total variance of [H+].

→ We thank the reviewer for pointing us to this issue. We apologize that we have made a typo in equation 2. The sensitivities in the first four terms on the RHS should be squared (we have had applied the formula correctly, it was an unfortunate typo that made it's way into the formula in the manuscript.). In response to the reviewers' comments, we have revised all sections in the manuscript on the carbonate chemistry decomposition. More details can be found above in response to the third general comment by reviewer 1. We appreciate the proposal of adding zonal mean plots and included those in the figures. They show that the representation of $[\text{H}^+]/\Omega_\text{A}$ variance by Equation 2 works reasonably well. The Taylor decomposition of Equation 2 itself is exact since it contains all terms. The references mentioned by the reviewer are now cited in the manuscript.

A second flaw with the analysis is that when comparing different contributions, the authors usually compare standard deviations, not variances (e.g., in Figs 7, 8, 9, and A5, and the 4 equations in Appendix C). Although perhaps more intuitive because of the units, comparing the standard deviations of the different components leads to a false impression of relative importance. It is only the variances of the components that linearly add up to the total variance. When minor components are compared in terms of their standard deviations, they appear overly important in terms of their contribution to the total variance. The authors should make all comparisons in terms of variances, not standard deviations.

→ We agree that Equation 2 suggests to use variance as variability measure. We therefore changed all variability analyses such that only variances are used throughout the manuscript.However, we would like to note here that using variances instead of standard deviations does not change one of our main conclusions that

changes in mean $C_T$ are mainly responsible for changes in extreme variability events in $[H^+]$. Furthermore, it does not changes our conclusions on the frequency decomposition of variability changes.

A third flaw with the analysis is that changes in the mean state (trend) have been removed and seldom enter into the discussion. Because most of the future change in both [H+] and $\Omega_A$ will be due to changes in the mean state, this neglect leads the authors to make statements that make little sense, such as the following: L152: "changes in different extreme event characteristics are only caused by variability"

$\rightarrow$ We have included a new Figure 11 in the Discussion section that also shows changes in extreme events due to changes in the mean state in addition to changes in variability. We have also added a new paragraph that discusses this new figure. In addition, we have clarified throughout the manuscript that in our study, changes in different extreme event characteristics are only caused by changes in variability. We now also call these extreme events 'extreme variability events' to clearly distinguish from extreme events that are caused by long-term ocean acidification.

L301-302: "extreme [H+] days are projected to disappear in the RCP8.5 scenario by the end of the century"

$\rightarrow$ We have clarified: "In most of these regions, extreme variability events are projected to disappear in the RCP8.5 scenario by the end of this century"

L309: "the occurrence of extremes is projected to decrease"

$\rightarrow$ We have clarified: "The regions in the Southern Ocean where the occurrence of extreme variability events is projected to decrease largely overlap with those for RCP8.5, at surface and at depth."

L314: "$\Omega_A$ extreme events are projected to disappear by 2081-2100"

$\rightarrow$ We have clarified: "extreme variability events in $\Omega_A$ are projected to disappear by 2081-2100"

L316-317: "No extreme events are projected for most of the ocean during 2081-2100 under RCP8.5"

$\rightarrow$ Changed to: 'no extreme variability events are projected for most of the ocean during 2081-2100 under RCP8.5'

Hence there is a communication problem that is directly tied to the authors' peculiar meaning for "extreme event". Unless the authors can bring the mean state back into the picture, they cannot legitimately use the term "extreme event". They are currently focusing only on a diagnostic for changes in variability. To help remedy the problem, I would like to see the authors provide a quantitative analysis of the contributions of each of the temporal components, including the change in the mean state (trend), to the overall change in the maxima. They might be able to do this without repeating their entire analysis procedure, simply by computing the variance due to the change in the mean state and adding that to the total of the other variance contributions that have already been computed. This analysis would clearly demonstrate the dominance of the change in the mean state and properly put the authors' other results into context.

$\rightarrow$ As pointed out above, we have now included a new Figure 11 in the Discussion section that also shows changes in extreme events due to changes in the mean state in addition to changes in variability. In addition, we now use the term 'extreme variability events' to clearly distinguish extreme events that are caused by only variability changes from extreme events that are caused by long-term ocean acidification and variability changes.

The study of marine heatwaves by Froelicher et al. (2018) included the change in the mean state as part of the analysis, so it is even more unclear to me as to why the same was not done here for the analysis of extreme events in [H+] and $\Omega_A$. Moreover, the relative importance of the change in the mean state relative to other temporal fluctuations (subannual, monthly, interannual) seems to be much larger for [H+] and $\Omega_A$ relative to SST. Its prominent role needs greater emphasis in this study by Burger et al.

$\rightarrow$ We are well aware of earlier studies that investigate changes in marine heatwaves under global warming. A co-author of our study is first author of the mentioned heatwave study. In general, there are different approaches how to define extreme events: (i) defining these events relative to ocean conditions during a fixed period of time (i.e. as was done in the Froelicher et al. study), or (ii) relative to a shifting baseline

as was done here. When following approach (i) and taking the secular change into account, the increase in the number of extreme days is much larger than following approach (ii) owing to the high signal to noise ratio in the ocean's carbonate chemistry under anthropogenic carbon uptake (see for example Froelicher et al. 2016). We have now clarified this with an additional paragraph and Figure 11 in the discussion section. The figure puts the changes in extreme variability events in context with changes obtained when analyzing extreme events defined with respect to preindustrial thresholds.

We included following paragraph in the discussion section: "In this study, we analyze changes in extreme variability events that are defined relative to a shifting baseline. If the long-term increase in ocean acidity and decrease in $\Omega_A$ is taken into account, i.e. defining the extremes with respect to a fixed preindustrial baseline (here the preindustrial 99[th] percentile for $[H^+]$ and the preindustrial 1[st] percentile for $\Omega_A$), the changes in $[H^+]$ and $\Omega_A$ extremes are much larger (cyan lines in Figure 11). Under the RCP8.5 scenario, every day becomes an extreme event day in year 2051 at surface and in year 2067 at 200 m depth (Figure 11a). The model also projects year-round extreme conditions for $\Omega_A$ at the surface and at 200 m by the end of the 21[st] century under RCP8.5 (Figure 11b). Comparing the two frameworks for surface $[H^+]$ extremes under present-day conditions, the annual number of extreme event days as defined in this study (i.e. with shifting baseline; black line in Figure 11) is on global average only 3.8 % of that also including the mean changes (i.e. with fixed preindustrial baseline; cyan line in Figure 11). This fraction differs regionally and reaches more than 10 % in the North Pacific, the North Atlantic, and the Arctic Ocean. Interestingly, the GFDL ESM2M projects that surface mean $[H^+]$ overshoots the preindustrial 99[th] percentile in year 1975 on global average. Thereafter, higher variability actually reduces the number of extreme event days that are above the preindustrial percentile. Surface mean $\Omega_A$ falls below the preindustrial 1[st] percentile in year 1990. After that, lower variability further increases the number of extreme event days below the preindustrial percentile. "

One reason why we have focused our main analysis on extreme variability events is the fact that even changes in variability can have deleterious consequences for marine organisms as we state in the introduction section.

**Specific comments:**

L2: Please define what is meant by by "acidity". Acidity is a term used by aquatic chemists to refer to base neutralizing capacity, just as alkalinity is used to refer to acid neutralizing capacity. Acidity does not a priori refer to [H+] just as alkalinity does not refer to [OH-]. The meaning of the authors ([H+]) differs and has to be defined. If acidity is to be used, I would be more comfortable with the term "free acidity" when referring only to [H+] because that is only part of the total acidity that is not bound up in other ions that react with [OH-].

$\rightarrow$ We thank the reviewer for bringing this up. When writing acidity, we actually refer to the hydrogen ion concentration on the total scale that includes sulfate ions

$$[H^+]_T = [H^+]_F + [HSO_4^-].$$

We added a clarifying sentence to the manuscript: "$[H^+]$ is on the total scale and hence the sum of the concentrations of free protons and sulfate ions."

L3: It is unclear what is meant by "mean ocean acidification". Ocean acidification involves changes in many $CO_2$ system variables simultaneously.

$\rightarrow$ We replaced 'mean ocean acidification' by the hopefully less ambiguous expression 'long-term ocean acidification'.

L28: Please define "short term". Also add a hyphen (see Global changes section below).

$\rightarrow$ We clarified: 'Superimposed onto the long-term decadal- to centennial-scale ocean acidification trend are short-term extreme variability events on daily to monthly timescales, during which ocean pH and/or $\Omega$ are extremely low (Hofmann et al., 2011; Joint et al., 2011; Hauri et al., 2013)'

L61-62: The sentence refers to changes in [H+] and $\Omega_A$ and cites references most of which only addressed the seasonal cycle of $pCO_2$. Please separate the references so that the readers know which one(s) actually addressed [H+] and $\Omega_A$.

→ Also in response to reviewer 3, we now write: 'In addition to the changes in the mean, recent studies suggest that the seasonal cycles in [H$^+$] and $\Omega$ are also strongly modulated under elevated atmospheric $CO_2$. Higher background concentrations of dissolved inorganic carbon and warmer temperatures produce stronger departures from mean state values for a given change in pertinent physical or chemical drivers for [H$^+$] and weaker departures for $\Omega$ (Kwiatkowski and Orr, 2018; Fassbender et al., 2018). Other studies have also addressed the changes in the seasonal cycle of $pCO_2$ (Landschützer et al.,2018; Gallego et al., 2018; McNeil and Sasse, 2016; Rodgers et al., 2008; Hauck and Völker, 2015).'

L76: "daily output" is ambiguous. Please say "daily mean output" if that is what you mean. If more frequent, say something like "6 hourly output". This is an important point because the analysis does not seem to include the potentially large diurnal cycle if it is based on daily mean output.

→ Yes, we meant daily mean output. We now refer to daily mean output and daily mean data throughout the manuscript.

L77: It seems ambiguous to write about "daily variability". That could be misunderstood by readers to mean "diurnal variability", which the authors did not consider because they are presumably using daily mean output.

→ We replaced 'residual daily variability' by 'subannual variability' throughout the manuscript.

L79-81: Please delete the last 3 lines. It is much nicer to end the Introduction with the aim of the study. Moreover, subsequently diluting the aim by following that with an outline of the paper makes no sense, particularly when that outline follows the standard IMRAD format (Introduction, Methods, Results, and Discussion) which is what is expected anyway.

→ Following the reviewers suggestion, we deleted the last three lines of the paragraph.

L98: I suspect that the GFDL-ESM2M model does not simply use the K1 and K2 directly from Mehrbach et al. (1973), unlike what is stated, but rather the K1 and K2 from the Mehrbach data after being converted to the total hydrogen scale by either Dickson and Millero (1987) or Lueker et al. (2000). Please clarify. Another important point that is not mentioned which concerns the model's total alkalinity equation. Does the total alkalinity equation in the GFDL-ESM2M model include contributions from phosphoric and silicic acid systems. Many models do not, and this can bias $pCO_2$ in the high latitudes by 10 ppm or so. [H+] would also be affected.

→ We thank the reviewer for this pointer. The model indeed uses K1 and K2 from Dickson and Millero (1987) (Table 4 therein) based on Mehrbach et al. (1973). Total alkalinity in ESM2M includes contributions from phosphoric and silicic acid and their conjugate bases. We clarified in the manuscript: 'The ocean carbonate chemistry is based on the OCMIP2 parametrizations (Najjar and Orr, 1998). The dissociation constants for carbonic acid and bicarbonate ions are from Dickson and Millero (1987), which are based on Mehrbach et al. (1973), and the carbon dioxide solubility is calculated according to Weiss (1974). Total alkalinity in ESM2M includes contributions from phosphoric and silicic acids and their conjugate bases.'

L100: This mention of diurnal variability seems irrelevant if the authors are using only daily mean output, which seems to be the case. Please delete.

→ We decided to keep this, as we do not want to hide this information from the reader, who might be interested in this caveat. No changes are made to the manuscript.

L140: The mention of the software seems too vague. What function was used from the library?

→ We specified the software in the text: 'calculated using the *measure.label* function from the *scikit-image* library for *Python*'

L148-149: It is not clear to me why each ensemble member is detrended with the ensemble mean trend

rather than the trend in the individual ensemble member. I understand that it is easy conceptually just to calculate the ensemble mean trend and use that. But one could also use a spline to detrend each ensemble member. Anomalies relative to the ensemble mean trend will be larger than those relative to the individual member trend. They would contain differences due to the different trends in the ensemble members as well as differences due to the other variability.

$\rightarrow$ Our approach is based on the assumption that all ensemble members share the same anthropogenic trend in $[H^+]$ and $\Omega_A$ (they share the same prescribed atmospheric $CO_2$ evolution) and that deviations from this trend are only a consequence of internal variability. Therefore, we chose to subtract the ensemble mean (with the seasonal cycle removed by an additional running mean) instead of splines because we did not want to make an assumption about the shape (linear, cubic, etc.) of the secular trend in $[H^+]$ over the 240yr period from 1861 to 2100. For a large ensemble, averaging the individual ensemble members yields an ensemble mean that only contains the forced evolution. In our case, due to the relatively small ensemble size, we likely also remove some variability from the ensemble members as discussed in appendix A. However we also think that, when subtracting an individual polynomial spline from each ensemble member, one also introduces some error by either a) adding interannual to decadal variability by subtracting a too-low-order polynomial that deviates from the actual forced evolution or b) subtracting a too-high-order polynomial that misinterprets interannual to decadal variability as forced evolution. Therefore, we kept our method as is.

L151: Please reword. It is not that the mean state is constant. Rather, the trend was removed.

$\rightarrow$ We clarified: 'The removal of the secular trend ensures that the mean state in the processed data stays approximately constant while day-to-day to interannual variability can change over the simulation period (depicted for one grid cell in Figure 1).'

L159: It says that both the standard deviation and variance are used. As mentioned in the general comments, I think that the comparison should be made only with the variance.

$\rightarrow$ Also in response to the general comment above, we removed any reference to standard deviation from the manuscript.

Figure 1: caption - change "subtracted" to "removed". If the ensemble mean were simply subtracted, the mean of the preindustrial period would be zero rather than 7.3. Using "removed" gives more leeway in the meaning.

$\rightarrow$ Also in response to reviewer 1, we clarified: '(b,d) Same as (a,c), but the ensemble-mean change with respect to the average of the 500-year long preindustrial control simulation has been subtracted'.

L164: "daily variability" is a confusing term. It could be taken to mean "diurnal variability" by some readers. I think the authors should stick with the term "subannual" and define that carefully as meaning "with the seasonal cycle removed but ignoring diurnal variability."

$\rightarrow$ Following the reviewer's proposal, we replace the term 'residual daily variability' by 'subannual variability' throughout the manuscript. Many thanks for this suggestion.

L166: It is dangerous to simply compare contributions from the different standard deviations, which do not add up linearly and give a biased impression of their contributions to the total variability. Rather, use the variance of the different components, which add linearly to make up the total variance.

$\rightarrow$ Following the general comment above, we removed any reference to standard deviation from the manuscript.

L168: Please clarify what is meant by "changes".

$\rightarrow$ In response to a comment by reviewer 1, we rewrote this sentence. It now states: 'In order to assess whether changes in low or high frequency variability cause changes in extreme variability events and their characteristics, we use three steps to decompose the total variability in $[H^+]$ into interannual, seasonal, and subannual variability.'

L172: Equation 1 does not seem to be used. Please just delete it and remove any mention of it in the text.

→ We prefer keeping Equation 1 in the text for clarity as it helps to understand Equation 2.

L182: Equation 2 is wrong. In the first 4 terms, the partial derivatives should be squared. This fix will change the balance of their contributions.

→ We thank for bringing up this typo and squared the partial derivatives.

L186: Since your Eq. 2 is wrong, it is not surprising that the terms do not sum up to match the total simulated variance. This match is key if we are to believe the deconvolution analysis.

→ This comment is discussed above in the general comments.

L209: Table A1 should be brought into the main body of the text and given as Table 1.

→ We thank the reviewer for this comment and we agree that this has, to our knowledge, not been published elsewhere yet. Also reviewer 1 has suggested the same thing. Therefore, we move the Table from the Appendix to the main text.

L215-221: It is unclear how trends and the error bars were computed for the data based estimates

→ We added an additional appendix section on the methodology for estimating trends and their confidence intervals as well as on testing for difference between trend estimates.

L215-221: What is the statistical significance of the model vs. data-based comparisons? In some cases, they do not appear to differ statistically, e.g., for $\Omega_A$ in the northern high latitudes.

→ We now tested all trend estimates for difference. Furthermore, we modified the uncertainty of the estimates for the simulated trends, as we had a mistake in our calculations for the uncertainty of the simulated ensemble-mean trend. With the corrected simulation trend uncertainties, it turns out that the difference for $\Omega_A$ in the northern high latitudes is in fact significant. Further details on the methodology can be found in the added Appendix D.

L229: Is there not some question about the data-based estimate of seasonal variability in the Southern Ocean where winter data is sparse?

→ This is a good point. We added an additional comment on line 203: 'An exception is the Southern Ocean where data-based $pCO_2$ products are uncertain due to sparse data in winter (Gray et al.,2018).' We also included at the end of the section: 'Nevertheless, the observation-based trends in the northern and especially southern high latitudes are rather uncertain because winter time data is sparse there.'.

L230: I would recommend being more cautious, by saying something like "it appears that the GFDL-ESM2M model is adequate to assess..."

→ Many thanks. We followed the recommendation and changed the sentence accordingly: "Even though we lack the daily observational-based data to undertake a full assessment, it appears that the GFDL ESM2M model is adequate to assess changes in open ocean ocean acidification extreme events."

L238: What is meant by intensity?

→ We clarified that we mean 'maximal intensity' as introduced in the method section.

L239: Table A2 should be brought into the main body of the paper and given as Table 2. This would allow the authors to simplify the text to mention only the mean changes and avoid giving the uncertainty ranges in parentheses. That simplification would make reading easier. Alternatively they could just give the ranges and not the mean changes. More generally, the first 4 paragraphs of this section (3.1) suffer from trying to cram too much information into every place where any numerical fact is mentioned. Writing complex things such as "0.20 nmol kg-1 (0.19-0.21 nmol kg-1; 18

→ We moved Table A2 to the main text. Furthermore, we followed the recommendation and removed the ensemble ranges from the text to make it more accessible. We only kept the ranges for the comparison of RCP2.6 and RCP8.5 since these are not given in the table.

L240: Please tell us what fraction of the total volume above 200 m is represented by the 2.7 x$10^3$ km$^3$.

→ We added the requested information and changed the sentence to: "Ocean acidity extremes in the upper 200 m occur with a typical volume of 2.7·$10^3$km$^3$, which is about 0.004 % of the total ocean volume in the upper 200 m (Figure 3g)."

L254-255: "is connected to" is vague. The increased contribution of interannual variability is not a mechanistic explanation. Previous work has discussed why changes in [H+] in the subsurface are larger than those at the surface (Orr, 2011; Resplandy, 2015). Perhaps their chemical explanation explains the longer duration of extreme variability mentioned here as well.

→ L254-255 was intended to state that the longer duration of events at depth compared to the surface is connected to the higher relative contribution of interannual variability to total variability at depth (63% at 200m depth vs 11% at the surface). We rewrote the sentence to clarify that we compare 200m depth duration to surface during preindustrial times: "The longer duration is connected to the more pronounced contribution from interannual variability (see Section 3.3).".

Differences in changes within our simulations between the surface and 200m depth are similar to the differences reported by Resplandy et al., 2015 for the comparison between stratified tropical waters compared to mode and intermediate waters: Although the mean DIC changes between preindustrial and RCP8.5 2081-2100 are larger at the surface than at 200m depth ($184\mu$mol kg$^{-1}$ vs $154\mu$mol kg$^{-1}$), the change in mean [H$^+$] is larger at 200m depth (11.4nmol kg$^{-1}$ vs 9.8nmol kg$^{-1}$). Yet, total [H$^+$] variance at surface quadruples between the two periods while variance at 200m depth only increases by a third. In our understanding, this is due to the fact that [H$^+$] variability changes (in contrast to [H$^+$] mean changes), not only depend on mean changes in the drivers (most of all DIC), but also on variability changes in the drivers. This is discussed in section 3.4 in the manuscript and we hence do not go further into detail in section 3.1.

L270-271: Given the large uncertainty between ensemble members, can the authors really say that there is a statistically significant increase in extreme days per year, intensity, and volume of events? I see no subplot for the Volume for the 200-m analysis.

→ We have not stated that there is a significant increase, but just say 'considerably'. However, we have now tested the significance with a simple t-test and found that the differences between 2041-2060 and 2081-2100 in maximal intensity, event days, and volume are all significant with p values smaller 1e-7 (testing the 20 ensemble-mean annual averages of the first period against those of the second). The text has been changed accordingly.

The volume of event metric describes the volume covered by events within the first 200m - it is by construction a column-integrated metric and not connected to a specific depth level. We changed the wording to clarify this.

Section 3.2 and 3.3: There are many statements particularly in these sections that make no sense because when addressing "extreme events", one must include the dominant contribution from changes in the mean state.

→ To clarify that we investigate extreme events that arise only from variability, we replaced 'extreme event' by 'extreme variability event' throughout the manuscript.

Section 3.4: This section will need to be completely rewritten after the fixing the first 4 terms in Eq. 2 and reevaluating the associated contributions. In addition that analysis should include a more quantitative assessment than just showing maps of the different contributions. A comparison of zonal means would be a good start. One could also compare contributions to the variance with stacked bar charts with different parts of each bar indicating different temporal components, similar to those shown in Kwiatkowski and Orr (2018).

→ We completely rewrote section 3.4. More detail on the revised decomposition can be found in the response to the third general comment by reviewer 1. As proposed by the reviewer, we added zonal mean plots to the figures that allow a more quantitative assessment.

L390: It is not clear what "nonlinear dependence" refers to. The authors use a 1st order Taylor expansion, which is linear in terms of the variances.

$\rightarrow$ We use the first order Taylor expansions to decompose variance in a period. When we analyse changes in variance between two periods, we analyse the change in this first order Taylor expansion. The changes in the partial derivatives in fact represent the contribution from nonlinearity in our belief. However, we agree that the sentence, as it was in the text, might confuse and now write instead "The increase in $[H^+]$ extreme variability events is a consequence of increased sensitivity of $[H^+]$ to variations in its drivers.".

L396-405: The comparison of the results for the 99th and 99.9 percentiles of "extremes" is also confused by the general neglect of the effect of the mean state, which is dominant. This paragraph should be rewritten with that in mind or otherwise deleted.

$\rightarrow$ We do not understand the reviewer comment here. Yes, as we stated above, we do neglect the changes in the mean state in our main analysis and focus on changes in extreme variability events. But here we investigate how different definition of extreme events may impact our results. No changes are made to the manuscript.

L410-415: There seems a missed opportunity to compare the authors' results in terms of the effect of changes in variability to those from McNeil and Matear (2008) who assumed that seasonal variability was unchanged but advanced the time at which the $\Omega_A = 1$ threshold was reached relative to previous work that did not address seasonal variability.

$\rightarrow$ We thank for the great hint and added a comparison of our results to the study by McNeil and Matear: "Assuming unchanged seasonality, McNeil and Matear (2008) found that seasonal aragonite undersaturation of surface waters in the Southern Ocean may occur 30 years earlier than annual mean aragonite undersaturation. However, our simulation shows that the reduction in $\Omega_A$ variability delays the onset of undersaturation by about 10 to 15 years in the Southern Ocean relative to a hypothetical simulation where variability does not change. Therefore, changes in variability need to be taken into account when projecting the onset of seasonal undersaturation, especially in the high latitudes and in the thermocline of the tropics."

L416-418: The discussion about arbitrary thresholds for [H+] is too vague. More details would be needed if it is to be kept.

$\rightarrow$ We agree with the reviewer and have deleted these two sentences.

L424-425: The text states, "It is therefore critical to uses daily temporal output to assess extreme events in ocean biogeochemistry." I believe that this statement tends to oversell the importance of the variability of the daily-mean output variability. Currently the authors results indicate that daily-mean variability of [H+] and $\Omega_A$ is generally a minor, second-order concern in surface waters. That is actually good news for future analysis, because saving daily mean output is not feasible on a routine basis, especially considering the many CMIP models and scenarios.

$\rightarrow$ We somewhat disagree with the reviewer given that the average duration of an acidity extreme event is less than a month. We clarified this in the text: 'In addition to earlier studies, we also show that changes in subannual variability contribute to changes in extreme $[H^+]$ variability events under increasing atmospheric $CO_2$ and that the average duration of extreme variability events at the surface and at present-day is about 15 days.'

On the other hand, the contribution to the total variance from diurnal variations, a component that the authors do not quantify, may be much larger especially in coastal regions. An assessment of that variability in the observations and in high-resolution models should remain a priority.

$\rightarrow$ We mention that in the caveat section: "Future studies with Earth system models that resolve diurnal processes are needed to quantify changes in diurnal variability and the impacts of these changes on extreme acidity events."

L447-449: The discussion about the "partial" diurnal cycle in TOPAZv2 (GFDL-ESM2M) does not seem relevant because by using daily means, the authors do not assess diurnal variability. This sentence should

simply be deleted. It will only confuse the reader.

→ We somewhat disagree with the reviewer. This statement about the diurnal cycle is included in caveat section as we think this is an important caveat of our study that needs to be discussed. No changes are made to the manuscript.

L455-456: It seems misleading to lead off this sentence with "While coastal species may be adapted to large variability in ocean acidity...". Although already large variability of [H+] is seen in coastal regions, that variability will also grow as atmospheric $CO_2$ continues to invade the ocean and coastal-water buffer capacities also decline.

→ We agree with the reviewer and have deleted this sub-sentence.

L464-465: As written, this first statement in the final paragraph was known already before this study was undertaken based on previously published papers concerning the future increase in the mean state and the future growth in seasonal variations of [H+]. This new study confirms those findings. It should not say "our analysis reveals", although it could say something like it "confirms previous findings ..."

→ We have extensively discussed previous findings in the discussion section as well as in the introduction. Therefore, we would like to highlight in this last paragraph some of the main findings of our paper. We therefore changed the sentence to: 'In conclusion, our analysis shows that marine organisms and ecosystems are projected to be exposed to less stable $[H^+]$ conditions in the future with more frequent occurrences of variability-driven short-term extreme $[H^+]$ conditions.'

L467: What is "mean" ocean acidification?

→ We replaced 'mean ocean acidification' by the hopefully less ambiguous expression 'long-term increase in ocean acidity'.

**Technical concerns:**

Global Changes:

- change "200 m depth" to "200 m" → Changed throughout the manuscript.

- change "point in time" to "time" → Changed throughout the manuscript.

- change "points in time" to "times" → Changed throughout the manuscript.

- MISSING HYPHENS: the terms "short term", "deep water", and "sea ice" should all include a hyphen when they modify a noun → Changed throughout the manuscript.

L38: change "The vast majority of" to "Most of the". Delete "so far"

→ Changed.

L44:

- delete "one might expect that". → Changed.

- change "exhibit" to "have" → Changed.

- It is not "carbonate chemistry" that will cross critical thresholds. That is like saying "physics will cross thresholds". You could instead say "key CO 2 system variables" → Changed and acknowledged.

L46: change "negatively impacted" to "adversely affected".

→ Changed.

L49: change "undergo a decline in" to "exhibit reduced"

→ Changed.

L63:

- change "suggest" to "project". → Changed.

- change "is projected to" to "will" → Changed.

L69:

- change "extreme" to "extremes" → We kept it as is.

- Please separate citations so that we know which ones address heatwaves and which ones address sea-level rise. → Done.

L78: change "imprint on the occurrence" to "affect"

→ Changed.

L84: change "are performed" with "were made"

→ Changed.

L104:

- change "1000 year" to "1000-year". → Changed.

- "a new computing infrastructure" is vague. Please clarify your intended meaning. → Also in response to reviewer 3, we have deleted the detailed information about the spin-up procedure as it is a distraction and not needed for understanding the model simulation setup.

L108: insert "the average" before "atmospheric"

→ Changed.

L111: insert "average" before "atmospheric"

→ Changed.

L120:

- change "500 year" to "500-year" → Changed.

- delete "long" → Changed.

- what is "potential vegetation" → Also in response to other reviewer comments, we deleted this additional information as it is not key to understand the results of the study.

L121: make the same first 2 changes as in the previous line but for "220 year long"

→ Changed.

L124:

- write "daily mean" instead of "daily" → Changed.

- add "the aragonite saturation state" before the symbol $\Omega_A$ → Changed.

- delete the 2nd sentence → Changed.

L202: change "is" to "appears to be"

→ Changed.

L238: change "typical" to "average"

→ Changed.

L266: delete "blue lines in". That color information should only be given in the Figure or its caption.

→ We kept this information in the main text as, in our opinion, this facilities the understanding of the text and its link to the figure.

Figure 2: The choice of colors for the lines could be improved. It is hard to distinguish light green vs. dark green and blue vs. purple. Yellow is difficult to see on a white background. Are these colors good for colorblind people?

→ The colors were taken from the viridis color scale and should be color-blind safe (see `https://cran.r-project.org/web/packages/viridis/vignettes/intro-to-viridis.html`). We changed the yellow color to orange color to make it more visible.

Figure 3: There is too much white space and repeated information. To remedy these problems, I'd suggest to start by labeling the subpanels as a matrix, i.e., with row labels and column labels. Thus you could remove the title for each subpanel and use that as the row label of the matrix. The row labels should only be given on the left (the connection with the right column is obvious). And column headings "Surface" and "200 m" should only be given at the top. The numbers on the tick marks can be kept for all subpanels, but the x-axis label should only be kept on the bottom row. The current y-axis labels should all be removed. The same approach should be taken for all subsequent figures with more than 2 subplots. The figures will then be more compact, less verbose and redundant, and readers will intuitively understand the setup of your multi-panel figures (without going to the figure caption, repeatedly).

→ We decided not to present the subpanels as a matrix in Figure 3 since we think this might lead to confusion with subpanel g) that does not fit the structure of 'Surface' as left column and '200m' as right column, since volume is a 3D metric, describing the volume of extremes in the entire first 200 m of water column - so it is neither belonging to 'Surface' nor to '200 m'. Row labels have been introduced in Figures 5 and A1.

L283-284: delete "but show distinct spatial patterns" (redundant).

→ Changed.

L295: the meaning of "reoccur" is unclear.

→ We replaced 'reoccur annually' by 'occur each year' which is hopefully clearer.

L421: Please provide separate citations for the papers that addressed pCO$_2$ and those that addressed [H+].

→ We adressed this comment and now write "Previous studies have shown that the seasonal cycle of surface ocean $p$CO$_2$ will be strongly amplified under increasing atmospheric CO$_2$ (Gallego et al., 2018; Landschützer et al., 2018; McNeil and Sasse, 2016) and that a similar amplification is expected for surface [H$^+$] (Kwiatkowski and Orr, 2018)."

L476: I like the choice of the authors in Appendix A to use "subannual" instead of "daily" variability. Please do the same in the main body of the paper.

→ Many thanks. We have changed it throughout the text.

Appendices: Please provide an equation number for each equation listed in the appendices.

→ We followed the comment and provided numbers for all equations in the appendix.

Appendix C: The equations should be written in terms of variances rather than standard deviations.

→ As we have removed any notion of standard deviation throughout the manuscript, these equations are now written in terms of variances.

Tables A1 to A3 do not follow the formatting standards used by Biogeosciences. See the author guidelines and previously published BG papers.

→ We changed the format of the tables and believe they follow the BG guidelines now.

Colorbar: On all the maps with diverging scales, there seems to be an excessive amount of yellow and it is difficult to assess where the "zero" line is. A better choice would be a color bar with very pale blue and very pale red next to each other in the center of the color distribution.

→ As the focus of our analysis is not on the sign of changes, but rather on the magnitude of changes, we kept the color bar as is.
* * *
**Reviewer 3: Sarah Schlunegger**

Burger et. al. presents a clear and logical assessment of projected changes in the variability of ocean acidity and their impact on acidic extremes. The study uses an Earth System Model (ESM) which they demonstrate has historical fidelity with observed acidification trends. The study contrasts future acidity extremes under low- and high-emissions scenarios, all relative to 99th percentile extremes as defined in the preindustrial control simulation. Changes in the duration, frequency, intensity and volume extent of extreme events are presented.

→ We thank the reviewer for the positive and encouraging review.

Drivers of change in [H+] variability are partitioned into contributions from changes in the mean state and variability of carbon concentrations, temperature, alkalinity and salinity. As noted by other reviewers, the exact decomposition used is flawed and needs revision before publication.

→ We have revised the methodology for the carbonate chemistry decomposition and the sections on the decomposition. We would like to refer to the reply to the third general comment by reviewer 1 for more details.

Other than this obvious issue, I have only three broad suggestions which would ready the manuscript for publication in Biogeosciences.

1. As a first assessment of changes in OA extremes, this paper has the opportunity to present simple, conceptual explanations for why those changes are occurring. This is done adequately for the case of $[H^+]$ but not for the case of Omega. Why is Omega variability decreasing? Is this largely driven by changes in carbonate chemistry, the way it is for $[H^+]$, or are changes in ocean dynamics involved? This has implications for the robustness of the results across other ESMs.

$\rightarrow$ We thank the reviewer for pointing out this missing piece. We now also decompose and discuss the variance changes in $\Omega_A$. We also added a new Figure 10 to Section 3.4 that shows the zonal mean decomposition of $\Omega_A$ variance change into the contributions from sensitivity changes, variability changes, and correlation changes. Furthermore, we show that changes in dissolved inorganic carbon are the dominant driver.

2. Further discussion is needed of why variability changes at the surface differ from those at 200m depth and of why there is strong compensation between the contributions of increasing carbon concentrations and decreasing carbon variability to $[H^+]$ variability at depth (9e vs 9f, and to a lesser extent at the surface, 8e and 8f). The striking spatial patterns in Figure 9e and 9f should also be explained.

$\rightarrow$ With the revised carbonate chemistry decomposition, we can now understand the striking compensating patterns we saw in the previous Figures 9e and 9f as well as 9b and 9c. The large compensating contribution arises from mixed changes in the sensitivities due to changes in mean DIC and DIC variability: The large variance increase from the mean increase in DIC is weakened by the simultaneous decrease in DIC variability there. Variability changes in DIC alone would have a much smaller effect (compare Figure 9c and 9d in the new manuscript version.). The striking pattern in (previous) Figure 9e itself is mainly a consequence of background $[H^+]$ variability: DIC mean changes have a much larger imprint on $[H^+]$ variability in regions that are already more variable (see preindustrial variability in $[H^+]$ in Figure A3a). We revised section 3.4 accordingly.

3. Include discussion of model uncertainty. For example, results that are a direct chemical consequence of invading anthropogenic carbon (e.g. increasing sensitivity of $[H^+]$ to drivers) are more likely to be more robust across models than results that are a consequence of changes in ocean physics. ESM2M has less warming and a stronger ocean carbon sink than the majority of CMIP5 models – how might the results presented compare to a model with higher climate sensitivity and less ocean carbon uptake models?

$\rightarrow$ We included a discussion of model uncertainty in the discussion section: "Secondly, our results, in particular at the local scale, might depend on the model formulation. As the mean increases in $C_T$ mainly drive the increases in extreme $[H^+]$ variability events (see Figure 8f), we expect that models with larger oceanic uptake of anthropogenic carbon show larger changes in extreme variability events than models with lower anthropogenic carbon uptake. The GFDL ESM2M matches observation-based estimates of historical global anthropogenic $CO_2$ uptake relatively well, but still has difficulties in representing the regional patterns in storage (Frölicher et al., 2015). Therefore, the exact regional patterns of $C_T$ changes may differ from model to model and further studies focusing on the physical processes that lead to the regional $C_T$ changes may help to better constrain the regional patterns in changes of acidity extremes. In addition, it is currently rather uncertain how $[H^+]$ and $\Omega_A$ variability changes as a result of changes in the drivers' variabilities. We have demonstrated that this factor is particularly important at depth for $[H^+]$ and for $\Omega_A$. It is well known that current Earth system models have imperfect or uncertain representations of ocean variability over a range of timescales (Frölicher et al., 2016; Resplandy et al., 2015; Keller et al., 2014). A possible way forward would be to assess changes in ocean acidity extreme events within a multi-model ensemble, which would likely provide upper and lower bounds of future changes in these events."

**Additional comments and stylistic suggestions**

L5: number of days where? At each point? On average? Overall occurrence anywhere?

$\rightarrow$ Changed to 'Globally, the number of ...'

L28: replace 'or' with 'and/or'

$\rightarrow$ Changed and acknowledged.

L33: define saturation horizon – e.g. Saturation generally decreases with increasing depth, with transition from saturated to undersaturated referred to as the saturation horizon.

$\rightarrow$ We clarified: 'saturation horizon (i.e. the depth between the supersaturated upper ocean and the undersaturated deep ocean (Feely et al.,2008; Leinweber and Gruber, 2013))'

L33: add 'can' in front of 'also' . . . 'can also'

$\rightarrow$ Changed.

L38: remove 'vast'

$\rightarrow$ Removed.

L40: add 'also' in front of necessary. . . 'also necessary'

$\rightarrow$ Included.

L45: . . .'critical threshold, like undersaturation.'

$\rightarrow$ Changed to: 'e.g. from calcium carbonate saturation to undersaturation'

L50: sentence rearrangement for clarity: 'In laboratory experiments in which deep-water corals are exposed to low-pH waters for a week. . .'

$\rightarrow$ Changed and acknowledged.

L53: Start sentence with 'Therefore,'

$\rightarrow$ Changed.

L61: '...at which higher background concentrations of dissolved inorganic carbon (DIC or $C_T$) and warmer temperatures produce stronger departures from mean state values for a given change in pertinent physical or chemical drivers.'

$\rightarrow$ Many thanks for this suggestion which we added to the text. Also taking into account a comment by reviewer 2, we now write: "In addition to the changes in the mean, recent studies suggest that the seasonal cycles in [H$^+$] and $\Omega$ are also strongly modulated under elevated atmospheric $CO_2$. Higher background concentrations of dissolved inorganic carbon and warmer temperatures produce stronger departures from mean state values for a given change in pertinent physical or chemical drivers for [H$^+$] and weaker departures for $\Omega$ (Kwiatkowski and Orr, 2018; Fassbender et al., 2018). Other studies have also addressed the changes in the seasonal cycle of $p$CO$_2$ (Landschützer et al.,2018; Gallego et al., 2018; McNeil and Sasse, 2016; Rodgers et al., 2008; Hauck and Völker, 2015)."

L103: Detail about spin-up and control is distracting.

$\rightarrow$ Also in response to reviewer 2, we have deleted the detailed information about the spin-up simulation.

L110: omit 'and', replace with comma

$\rightarrow$ Changed.

L120: 'year-long' – however these detail is distracting so potentially remove all together.

$\rightarrow$ We agree. We have modified and shortened it accordingly.

Figure 1. The figure caption (and methods) say that the ensemble mean is removed – however what is removed is actually the departure of the ensemble mean from the preindustrial state. If the ensemble mean itself were removed, then all values at year 1861 would be $\sim$ zero. Update description or replot.

$\rightarrow$ We thank for the pointer. We have clarified: "(b,d) Same as (a,c), but the ensemble-mean change with respect to the average of the 500-year long preindustrial control simulation has been subtracted."

Figure 1. Panel b is presumably derived from panel a, however it is confusing that the variability in panel a contracts during the 21st century, however it is shown to increase in panel b. Please add an explanation in the caption or text.

→ As the reviewer correctly says, panel b is derived from panel a. However, we would like to note here that the variability is actually increasing and not decreasing in panel a. No changes are made to the manuscript in response to this comment.

L230: remove 'we consider'

→ Removed.

L240-245: Further and continued clarification that values given are a global average of local changes. With each presentation of a value, indicate it is a global average.

→ We clarified at the beginning of section 3 that global values are grid cell based characteristics that are aggregated globally.

L244: change 'single events' to 'individual events'

→ Changed.

L255: Why is high-frequency variability larger at the surface whereas low-frequency variability larger at depth? ... Presumably because the surface has direct contact with the atmosphere, and its chaotic, high-frequency (weather) variability, whereas the deeper ocean experiences stronger low-frequency variability as the ocean acts as a low pass filter on the atmospheres' stochastic forcing (e.g. Hasselmann 1976). Add/include explanation.

→ At preindustrial, subannual variance is five times larger at depth than at surface ($0.062$ nmol$^2$kg$^{-2}$ vs $0.012$ nmol$^2$kg$^{-2}$), also in percent of total variability (14.6% (200m depth) compared to 8.3% (surface)). So the model does not simulated larger high-frequency variability at surface than at depth. In fact, the seasonal variance is most important at surface (81% of total) while interannual variability is most important at depth (63% of total) (at preindustrial). Therefore, the stronger relative contribution from interannual variability at depth is leading to longer-lasting events. We added a short paragraph on preindustrial variability in section 3.3: "For the preindustrial, the model simulates overall larger [H$^+$] variance at depth than at the surface ($0.42$ nmol$^2$kg$^{-2}$ vs. $0.15$ nmol$^2$kg$^{-2}$, not shown). Seasonality has the largest contribution at the surface (81 % of total variance). At 200 m, interannual variability has the largest contribution (63 %), and also subannual variability is more important compared to the surface (15% vs. 8%)."

L298: insert 200 meters... ' In contrast to the surface, 200 meter [H+] extremes'...

→ Changed and acknowledged.

L357-361: C$_T$ increases due to invasion of anthropogenic carbon. But why does ALK change? ALK is not influence by gas-exchange, but by biology or circulation. Briefly explain ALK changes and give reference.
→ We added an explanation: "The changes in A$_T$ are largely due to changes in freshwater cycling that also manifest in salinity changes (Supplementary Figure A4, Carter et al., 2016)."

378-379: [H+] variability changes at depth are larger than at surface – is this mostly due to the fact that background C$_T$ concentrations increase with depth, however increased C$_T$ itself is associated with decreased susceptibility to external variability? Figure 8 e) vs f) indicates this is the case. Provide commentary.

→ Yes, the [H$^+$] variability changes (from PI to 2081-2100 under RCP8.5) from mean changes in C$_T$ are larger at 200m depth compared to the surfaced due to the higher preindustrial background carbon concentration (global mean: $2078$ $\mu$mol kg$^{-1}$ (200m depth) vs. $1959$ $\mu$mol kg$^{-1}$ (surf.)), although changes in C$_T$ are smaller at depth (global mean: $154$ $\mu$mol kg$^{-1}$ (200m depth) vs. $184$ $\mu$mol kg$^{-1}$ (surf.)). The impression that increased C$_T$ is associated with decreased variability in C$_T$ was an artefact of the method. We thank the reviewer for bringing this up. We rewrote most of section 3.4.

Figure 9. Panel 9b and 9c bear strikingly similar patterns and magnitudes – where the changes in the meanstate of the drivers is largest and positive, changes in the variability are also largest and negative. And this is mostly coming from $C_T$ mean (increasing) and $C_T$ variability (decreasing). Is this directly understandable through carbonate chemistry? In a higher DIC world, DIC variations are actually smaller? Or is it that the physical, surface to deep gradient in DIC is weakening with surface invasion of anthropogenic carbon and therefore mixing-related variability is reduced? Explain this near-perfect anti-correlation. Also, the structures of strong mean and variability changes look dynamical. Potentially the expansion of the subtropical gyres. Explain the spatial structure of the patterns of strong drivers.
→ As we have modified the Taylor decomposition method, these patterns have changed. Please see above.

In the appendix, include figures of the pre-industrial and year 2100 concentrations of $C_T$ at surface and 200m depth. The patterns in 9e and 9f are likely related to both the starting concentrations of $C_T$ and the storage of $C_T$. Figure A4a and A4e show the change in $C_T$, which resembles the strong patterns of 9e and 9f. Synergies of anthropogenic carbon and background concentrations of $C_T$ may help explain strong structures of 9e and 9f. The strongest temperature changes (A4g) also resemble the strong patterns of 9e and 9f, indicating either dynamical changes or thermal changes are involved.
→ We agree that the patterns of variability changes in 9e and 9f are related to the preindustrial starting points. The preindustrial variability in $[H^+]$ (see Figure A3a - event intensity is a good indicator for variability) already shows the patterns that are also visible in in 9e and 9f. But what's the reason for these preindustrial patterns of high $[H^+]$ variability? Focusing on DIC, this could have in principle two reasons - (a) because of large sensitivity w.r.t drivers variations arising from high mean DIC and (b) because of large variations in DIC. It is actually more connected to (b), the patterns resemble the preindustrial patterns of DIC variability, not so much the patterns of mean DIC (see figure below). We decided not to add an analysis of preindustrial $[H^+]$ variability, which would likely add even more complexity to the manuscript. But we added a sentence in section 3.4 to highlight that the patterns seen in Figure 9 are connected to the preindustrial background variability in $[H^+]$: "There, the preindustrial background $[H^+]$ variability is also the largest (Figure 3a). As a result, an increase in the sensitivities due to an increase in mean $C_T$ has the largest effect there.".

[Figure]

[Figure]

Line 393: Is the decrease in Omega variability a direct consequence of carbonate chemistry? Or of changes in the variability of the drivers or physical state (e.g. vertical gradient hypothesis stated before)? This is an important distinction, as carbonate chemistry is well constrained (little model uncertainty) whereas dynamical changes have much larger model uncertainty. If reductions in Omega variability are chemically deterministic, then this feature will be relatively robust across models, as most use the same (e.g. OCMIP2, Najjar et al. 2007) protocols for carbonate chemistry calculations. But if it is dynamical, it may not be. Furthermore, if it is chemically driven, and if it is related to the storage of anthropogenic carbon and/or concentration of natural carbon, then contemporary observations and reconstructions of the distribution of natural and anthropogenic carbon in the ocean could reveal where changes in Omega variance are likely to occur.
→ We added a new paragraph in section 3.4 that shows the carbonate chemistry decomposition of variability changes in $\Omega_A$ and a new Figure A6. Changes in the variability of the drivers have a larger impact on $\Omega_A$ compared to $[H^+]$. Furthermore, $\Omega_A$ variability changes are even more determined by changes in DIC. Changes in the other potential drivers are less important (dashed vs. solid lines in Figure A6). As a result, projected changes in $\Omega_A$ variability largely depend on changes in DIC variability. We added a sentence to the Discussion section: "
[revised manuscript text omitted]

---

## Author Response (AR2)

GENERAL COMMENTS
====================

I thank the authors for responding to my previous remarks in their revised manuscript. Largely the changes that they have made are satisfactory. They have fixed equation (2), using variances instead of standard deviations. They have also changed their figures to show variances rather than standard deviations.

We thank the referee again for the careful review, which lead to substantial improvement of our manuscript.

Furthermore, they now go beyond just mentioning the mean state, touching on how results change when that is included in the analysis. Unfortunately, their demonstration of the importance of the mean state comes too late and is too brief. Appearing only as a glimpse in the middle of the Discussion section, many readers may well miss it. I think it should be expanded upon further and presented in the first part of the Results section. Right now it seems to come as only an afterthought in the Discussion, a peculiar place given that it is the change in the mean state that is the dominant factor controlling the change in actual extreme events for both [H+] and $\Omega A$.

We now present the results of extremes relative to a fixed baseline at the beginning of the results section (section 3.1) and have expanded the associated paragraph. Furthermore, we now included these results in the abstract and added a paragraph to the introduction that outlines the different existing definitions of extremes and added relevant references:

"Changes in extremes can arise from changes in the mean, variability, or shape of the probability distribution (Coles, 2001). There exists no general accepted definition of an extreme event beyond the common understanding that an extreme is rare (Pörtner et al., 2019). As a result, many different approaches exist to define extreme events (Smith, 2011). If a relative threshold (e.g. quantile) is used to define an extreme event, it is important to distinguish between extreme events that are defined with respect to a fixed reference period or baseline, or if the reference period or baseline moves with time. If the baseline is fixed, the changes in the mean state as well as changes in variability and higher moments of the distribution contribute to changes in extreme events (e.g. Fischer and Knutti 2015; Frölicher et al. 2018, Oliver 2018). However, if a shifting baseline is used, changes in the mean state do not contribute to changes in extreme events (Stephenson, 2008; Seneviratne et al., 2012; Zscheischler and Seneviratne, 2017; Cheung and Frölicher, 2020; Vogel et al., 2020). In this case, changes in extremes arise solely due to changes in variability and higher moments of the distribution (Oliver et al., 2019). This latter definition ensures that values are not considered extreme solely because the baseline changes under climate change (Jacox, 2019; Oliver et al., 2019). Whether extreme events should be defined with respect to a fixed baseline or with respect to a shifting baseline depends on the scientific question. For example, the shifting-baseline approach may be more appropriate when the ecosystems under consideration are likely able to adapt to the mean changes, but not to changes in variability (Seneviratne et al., 2012; Oliver et al., 2019). Here, we use both approaches, with a special focus on the analysis of ocean acidity extremes with respect to shifting baselines."

In an attempt to fix previous confusion, another reviewer suggested that the authors should generally replace "extreme event" with "extreme variability event", and the authors have complied. Although well intentioned, when I read the revised manuscript, the ambiguity of this term, or rather what the authors mean by it, seems to stand out. While "variability" refers to an oscillation over a given period involving both highs and lows, "extreme event" refers to the time or times at which a high or low threshold is surpassed.

A search with Google Scholar reveals no precedent for use of "extreme variability event" in climate science. Although it has been used in astrophysics in several published papers, the meaning is different. It is only used to refer to events with high variability. It is never used, in any field as far as I can tell, to refer to a maximum or a minimum generated by that variability crossing a upper or lower limit (after removal of the mean), i.e., the meaning intended by the authors. Due to this confusion and lack of precedent, that term should be avoided here. More careful wording is still needed.

We carefully revised all sentences that used this expression. In the newly written paragraph of the introduction, we present the different existing definitions of extreme events. Previous studies defined extreme events either relative to a fixed or shifting baseline, e.g. IPCC Special Report on Extremes (Seneviratne et al. 2012) or Jacox (2019). Therefore, the term 'extreme events' was used when using a fixed baseline as well as when using a shifting baseline. To avoid any confusion, we now clearly state throughout the manuscript including the subsection headers, if extremes are defined to a fixed or shifting baseline. We also reduced the usage of the term 'extreme event' to the necessary minimum.

Unfortunately to fix this ambiguity is not so simple. Sentences and sometimes full paragraphs will need to be revised on a case-by-case basis. In the specific comments below, I give a couple of examples of how the authors could rewrite a sentence (L33-34) and a paragraph (L346-352) to avoid this confusing 3-word term. Other specific comments also address how to improve some related passages, while elsewhere I leave it to the authors make the relevant improvements.

We thank the reviewer for providing us some concrete suggestions on how to rewrite the sentences. We have carefully revised the entire manuscript.

Overall then, I recommend that the revised manuscript be further improved by expanding on the section concerning relative importance of the changing mean state, moving that expanded subsection up to the beginning of the Results section, and clarifying all passages where the authors currently used "extreme variability event" or similar ambiguous terminology (see specific comments).

We followed the reviewers' recommendations by expanding the discussion of acidity extremes defined with respect to a fixed baseline, by moving this section to the beginning of the results section, and by improving the terminology. We also added a paragraph to the introduction that embeds our extreme event definitions (fixed and shifting baseline) in the existing literature.

SPECIFIC COMMENTS
==================

GLOBAL changes:
- In some places the authors still use "daily" instead of "daily mean" (e.g., L133, 134, 159)

Changed in L133, L134, additionally in L217, L258, L496, L506, L507 (previous manuscript version), and in the caption of figure 1. We haven't found the word 'daily' on L. 159.

- replace "GFDL ESM2M" with "the GFDL ESM2M model". Added clarity is needed at least for non-modelers. Readers tend to skip around in a paper and may well miss the original definition.

Changed throughout the manuscript.

TITLE
I am confused by the title. The use of "extremes" appears misleading because the paper does not actually focus on extreme events.

We think that using the term 'extremes' is appropriate. Extremes can be defined relative to a fixed or shifting baseline. This has been described and highlighted in several IPCC reports including

chapter 3 of the IPCC Special Report on Managing the Risks of Extreme Events and Disasters to Advance Climate Change Adaptation (SREX; Seneviratne et al. 2012). We added a new paragraph to the introduction to clarify the terminology, but kept the word in the title.

ABSTRACT

L1: I'd suggest to change "extremely" to "relatively". The sentence then becomes more subtle and avoids word repetition.
Changed.

L2-3: For a clearer and less verbose sentence, I'd suggest to change the last half to "not only due to changes in the long-term mean but also changes in short-term variability."
Changed.

L5: The phrase "changes in variability on variability-driven ocean acidity extreme events" is not clear. A common but wrong interpretation from readers could be how variability contributes to extreme events in the context of the dominant role of the changing mean state. The term "variability-driven ... extreme events" does not clearly indicate that the changes in the mean state are ignored and that the focus is actually not on extreme events but rather changes in extremes generated by variability alone.
We now clarify in the abstract and throughout the manuscript that when extremes are defined with respect to the preindustrial baseline the mean changes are the main reason for changes in extremes. Furthermore, we clarify that mean changes do not contribute to changes in extremes when defining extremes with respect to a shifting baseline. We write "When defining extremes relative to a fixed preindustrial baseline, the projected increase in mean [H+] causes the entire surface ocean to reach a near-permanent acidity extreme state by 2030 under both the low and high CO2 emission scenario. When defining extremes relative to a shifting baseline (i.e., neglecting the changes in mean [H+]), ocean acidity extremes are also projected to increase because of the simulated increase in [H+] variability, (...)"

L8: same problem as just above with the term "variability driven extreme events"
We now avoid this wording and call them 'extreme events'. In addition, we clarified throughout the manuscript if the extremes are defined to a fixed or shifting baseline.

L10: ambiguous use of "extreme event".
We now clearly state if the extreme events is defined to a fixed or shifting baseline.

L12: change "variability extremes" to something like "the associated increased maxima from that variability alone"
We have now removed the reference to extreme events in this sentence and just write "Increases in H+ variability arise predominantly from increases in the sensitivity of H+ to variations in its drivers (i.e., carbon, alkalinity, and temperature) due to the increase in oceanic anthropogenic carbon."

L13-14: This sentence regarding OmegaA is open to misinterpretation due to the ambiguous term "variability driven extremes" and the opposing effects from changes in variability and changes in mean state on actual OmegaA extreme events. As space seems to be lacking to get into enough detail, I would suggest simply to delete this sentence.
We deleted the sentence.

L15: change "will" to "may"

Changed.

1. INTRODUCTION:

L23: change "formation of [HCO3-] from [CO32-]" to "conversion of [CO32-] to [HCO3-]"
Changed.

L33-34:
* What qualifies as "extremely low"? Would it not be better to say "much lower than usual" because it is the relative difference that matters?
Changed.

* Does not variability also lead to extremely high values of pH and OmegaA not only low values? As phrased, the statement is one-sided.
Yes, an increase in variability also leads to extremely high values of pH and OmegaA. We now explicitly write that we define extreme events to feature low values: "...during which ocean pH and/ or OmegaA are much lower than usual".

The problem is the term "extreme variability events", which implies changes in both highs and lows. While variability refers to oscillation over a given period, "extreme" and "event" refer to a moment in time when a threshold is surpassed (as mentioned in the General Comments).
To remedy the problem, I would suggest to change the following sentence from

"Superimposed onto the long-term decadal- to centennial-scale ocean acidification trend are short-term extreme variability events on daily to monthly timescales, during which ocean pH and/or Ω are extremely low (...). These events ..."

to something like

"Superimposed onto the long-term trend are lows and highs in pH and ΩA that will be modified as short-term variability changes (...). Lows from extreme variability …"
We thank for pointing out the difficulties with the sentence and rewrote it to: "Superimposed onto the long-term decadal- to centennial-scale ocean acidification trend are short-term extreme events on daily to monthly timescales during which ocean pH and/or OmegaA are much lower than usual". We think that avoiding "extreme variability event" and defining extremes to feature low values solves the problem.

L42: This sentence does not seem to make sense since "extreme variability events" are anomalies relative to a long-term mean and mean pH conditions are not anomalies. Perhaps the authors mean "changes in mean pH"? In any case, "extreme variability events" is ambiguous and should be avoided.
We replaced 'extreme variability events' by 'extreme events' . We now write: "Such extreme events may have pH levels that are much lower than the mean pH conditions projected for the near future (Hofmann et al., 2011)."

L46: change "variability extremes" to "variability". That is, "variability extremes" refers to when variability is high or low but not the min or max generated by that variability (anomaly relative to the mean).
We replaced 'variability extremes' with 'variability and extremes'

L50: delete "events"

We now write "Furthermore, if the frequency and intensity of short-term extreme events strongly increase, …".

L51: "saturation" is when OmegaA = 1. "Supersaturation" is the word used when OmegaA > 1.
We changed it to "from calcium carbonate supersaturation to undersaturation".

L53:
* insert "ion" after "carbonate"
Done.

* "several days of aragonite undersaturation" is unclear and open to misinterpretation. Change to "several days of being exposed to waters which are undersaturated with respect to aragonite".
Changed.

L60:
- Please tone it down a bit by changing "of critical importance" to "important"
Changed.

- suggest to change "and the changes therein" to "and how that will change"
Changed.

L75: change "impose changes in" to "affect"
Changed.

2 METHODS
2.1 MODEL & EXPERIMENTAL DESIGN

L120: delete "long"
Changed.

2.2 ANALYSIS METHODS

The word "METHODS" seems unnecessary since this subsection is already in the METHODS section.
Changed.

2.2.1 EXTREME EVENT DEFINITION AND CHARACTERIZATION

L132: "sulfate" ions? The authors' definition of the [H+] total scale is wrong.
Changed 'sulfate ions' to 'hydrogen sulfate ions'.

L133:
* change "daily" to "daily mean"
Changed.

* change from "a one-in-a-hundred days event" to "occurring once every 100 days."
Changed.

L135-136:
* I do not buy this statement, as written, as justification for using relative thresholds over absolute threshold. Absolute thresholds can be determined as a function of a regionally (or grid-cell) varying baseline (preindustrial conditions here).

Absolute thresholds are, to our knowledge, usually used when referring to a fixed, often global threshold, such as for example a calcium carbonate saturation state of 1. A relative definition of extreme events based on quantiles of the local distributions ensures the same preindustrial event frequency across regions. We clarified: "In contrast to absolute thresholds, relative thresholds, such as those used here, take into account regional differences in the variables mean state, variance, and higher moments. Events that are defined based on relative thresholds have the same occurrence probability across the globe in the period in which they are defined (e.g. preindustrial period; see also Frölicher et al. (2018))."

\* "statistical properties" is vague.
The term was removed from the manuscript (see above).

L136-138: I don't follow. Biases in simulated variables also alter the definition of model-defined absolute thresholds if they are regionally varying.
We deleted the sentence to avoid confusion.

L138: "extreme events" is ambiguous.
The sentence was removed from the manuscript (see above).

L135-138: It seems that the authors are trying to use the last 4 lines of this paragraph to justify their choice of studying only the effects from variability and ignoring the changes in the mean state, but the text is not clear nor convincing. Moreover, the text only exposes the advantages. The other side of the story is that an analysis of the relative thresholds has little to do with actual "extreme events" (the authors' own term in the last sentence) when changes in the mean state are dominant (the case here for [H+] and $\Omega A$).
We think that the paragraph was not written in a clear way. The paragraph intended to distinguish extreme events based on relative thresholds from those based on absolute thresholds. Our intention was not to discuss the (dis)advantages of applying a fixed or shifted baseline for the analysis. We clarified: "In contrast to absolute thresholds, relative thresholds, such as those used here, take into account regional differences in the variables mean state, variance, and higher moments. Events that are defined based on relative thresholds have the same occurrence probability across the globe in the period in which they are defined (e.g. preindustrial period; see also Frölicher et al. (2018))."

**2.2.3 TAYLOR DECONVOLUTION METHOD TO IDENTIFY MECHANISTIC CONTROLS OF [H+] AND $\Omega A$ VARIABILITY CHANGES**

This 13-word subsection title is too detailed. Please change it to something simpler such as "Taylor expansion". The first sentence of this section provides the more detailed complementary information.
We changed the subsection title to: "Taylor expansion of H+ and OmegaA variability changes".

L198: twice in this sentence, we see "drivers" modifying a noun. It should actually be "drivers' " because in both cases it is a plural possessive.
Changed.

L199: The sentence "We do so by calculating the full Taylor series ... up to the fifth order." will confuse readers for 2 reasons:
\* no Taylor series is "full" (in practice there is always truncation of higher order terms)
Since Equation 2 is a polynomial of fifth order in the sensitivities, standard deviations, and correlation coefficients, its Taylor series only has five orders (all higher orders are zero). The expression 'full' was intended to express that all non-vanishing orders were taken into account. We

removed the details about the Taylor expansion as it may be rather confusing at this point. See also below.

\* Mentioning an "order" in a sentence about a Taylor series should refer to the order of the Taylor series. But the authors' Taylor series equation is only 1st order. It does not even include terms with 2nd derivatives as needed for a 2nd order Taylor series. After looking at Appendix C, I understand a little more about what the authors might be trying to say, but it would be necessary to separate the two different meanings for "order" in 2 separate sentences. Better yet, avoid all confusion by not even mentioning fifth order here. It is not critical to the text, at least not in this section.
We agree with the reviewer that mentioning orders might lead to confusion in this context since Equation 2 builds on Equation 1 that is a first order Taylor expansion. We removed the reference to 'order' at this point.

L200-204: Appendix C should be mentioned earlier. Its first mention does not come until ten lines later (L214), so readers will struggle trying to make sense out of the authors' introduction of the 4 new terms, which are not so simple after all.
We included a reference to Appendix C in L199 (previous manuscript version, now L225).

L207: Eq 3 is verbose. Eliminating the H+ subscript would be an improvement and make it more general. Making that change would mean that the authors should also generalize parts of the previous text, including for example changing the end of the sentence on line 201 to "overall change in variance for each of H+ and $\Omega A$ ($\Delta s\sigma 2$)"
While we agree that the formula would be more elegant and easier to read without the subscript, we think that it is more consistent and precise to keep the subscript for two reasons: 1) we denote the drivers variance with the variable name in the subscript and think that it is more consistent to do it the same way for H+ and OmegaA. 2) We decompose variance changes in H+ and in OmegaA in section 3.5. We think that it is easier to distinguish the two from each other when keeping the subscript.

L208-210: This sentence seems out of place. First it mentions Figs 8-10 too early (before Fig. 3 is introduced). More importantly, the concern turns out to be minor since the authors show that the grey and black lines generally agree well. I suggest to move the idea of the agreement to the the caption of Fig. 8 or to the text that discusses the results from those figures.
As suggested we moved the sentence to the caption of Figure 8 (which is now Figure 9).

L212-214: The mathematical terms are becoming even more verbose. Again, removing the H+ subscript would help.
As suggested below, we now also show the contributions to the four terms in Equation 3 arising from AT and CT as well as from AT, CT, and T. To not introduce more symbols, we decided to drop the formalism introduced in L212-214. We now write "We also assess the contributions to the four components from CT alone, from CT and AT, and from CT, AT, and T." Similarly, the corresponding section in appendix C was revised.

2.3 MODEL EVALUATION

L227: "co2sys" should be capitalized (all letters).
Changed.

L236: simplify the parenthetical statement to just "(Appendix D)"
Changed.

~L247:

Fig 3 (contents): The Landschutzer data product does not extend all the way up to 90°N. Please modify the label of the northernmost band accordingly. Also the Southern Ocean does not extend all the way down to 90°S.

The labels were changed from 80°N to 75°S.

Fig 3 (caption): Change the cumbersome phrase ". (c,d) The same as (a,b), but for ΩA." to "along with the same for c) data-based ΩA and d) simulated ΩA." Other figure captions also have the same type of cumbersome phrase and should be improved in the same way.

We changed all figure captions accordingly.

L249: insert "seasonal amplitude of" before "ΩA".

Changed.

L257: delete "rather".

Changed.

3 RESULTS

L264: change "was" to "were"

Changed.

L262-265: Four "we" on 4 consecutive lines is a bit too much.

This paragraph was changed in the revised manuscript.

3.1 GLOBAL CHANGES IN OCEAN ACIDITY VARIABILITY EXTREMES

Inconsistent terminology:
Fig. 4:
* panel a title: "Yearly extreme days"
* caption: "frequency"
--vs.--
Fig. 11:
* caption: "number of extreme event days per year"
--vs--
Fig. 5: "Yearly OmegaA Extreme Days"

We now consistently name this extreme event metric as 'yearly extreme days' throughout the manuscript.  In Fig. 5, we write Yearly Extreme Days (OmegaA) so that the reader can distinguish the figure more easily from those for [H+].

Line 297: delete "only"

Changed.

line 303: change "long" to "longer"

Changed.

line 307: insert "period" after "historical" and insert "during" between "and" & "the"

Changed.

line 310: delete "It should be noted that".

Deleted.

3.2 REGIONAL CHANGES IN OCEAN ACIDITY VARIABILITY EXTREMES

L314-327: This paragraph is weak because it only quantifies absolute changes. We also need to know the numbers for what were the percent changes relative to the preindustrial reference. We now also state the relative changes.

L342: change "similar as" to "similar to"
Changed.

L340-345: It would be nice to remind readers that you are discussing results for [H+].
We added a reference to [H+] in L343 (previous manuscript version, now L386).

(2) L346-352: This paragraph is particularly confusing because (a) it relies on the ambiguous term "extreme variability event" and (b) its conclusions (an "improvement" with time due to declining variability alone) are opposite to those for a real "extreme event" (a "worsening" with time, i.e., for ΩA where the long-term declining mean dominates the signal). Here is the authors' text:

"While the decline in mean ΩA generally leads to lower values in ΩA, extreme variability events in ΩA are projected to become less frequent throughout most of the ocean (89% of surface area under RCP8.5 at the end of the 21st century; Figure 5c). In many regions, extreme variability events in ΩA are projected to disappear by 2081-2100 under the RCP8.5 scenario (grey regions in Figure 5c). However, the frequency of surface ΩA variability extremes is projected to increase by 10 or more days per year in the subtropical gyres, especially in the western parts of the subtropical gyres. At depth, no extreme variability events are projected for most of the ocean during 2081-2100 under RCP8.5 (Figure 5d)."

I would suggest that the authors try to lead readers more by the hand while avoiding the term "extreme variability event". As food for thought, here is an attempt to make this paragraph clearer:

"Although the long-term mean of surface ΩA declines, so generally does its projected variability. Thus after removing the mean, there is also a decline in the number of days per year when the variations lead to levels that are below the reference threshold (1st percentile from the preindustrial variability distribution with the mean removed). That decline in frequency is evident over 89% of the ocean; by 2081-2100 under RCP8.5 the low reference threshold is never reached over most of the ocean (grey regions in Figure 5c). Conversely, in the subtropical gyres the low threshold is crossed with increasing frequency, reaching 10 or more days per year during 2081-2100 under RCP8.5. Simultaneously, at 200 m, the corresponding low reference threshold is never reached over most of the ocean (Figure 5d)."
The paragraph has been revised to: "While the decline in mean OmegaA generally leads to lower values in OmegaA and therefore extreme events are becoming more frequent when defined with respect to a fixed preindustrial baseline (Section 3.1), extreme events in OmegaA are projected to become less frequent throughout most of the ocean when defined with respect a shifting baseline (89% of surface area under RCP8.5 at the end of the 21st century; Figure 6c). In many regions, extreme events in OmegaA are projected to disappear by 2081-2100 under the RCP8.5 scenario (grey regions in Figure 6c) when defined with respect to a shifting baseline. However, the number of yearly extreme days in OmegaA is projected to increase by 10 or more in the subtropical gyres, especially in the western parts of the subtropical gyres. At 200 m depth, no extreme events are projected  for most of the ocean during 2081-2100 under RCP8.5 (Figure 6d)."

3.3 DECOMPOSING [H+ ] VARIABILITY CHANGES INTO INTERANNUAL, SEASONAL, AND SUBANNUAL VARIABILITY CHANGES

This subsection title is verbose. It should be changed to something like "Decomposition of temporal variability"

We changed it to: "Decomposition of temporal variability in [H+]"

L355: change "We therefore decompose" to "Thus we decomposed"

Changed.

L356: change "overall" to "generally"

Changed.

L360: "preindustrial" is a period, while "the end of this century" is a point in time. "Preindustrial" is also an adjective and should be modifying something? This issue seems a common mistake in this manuscript.

We rewrote it to "Over the 1861-2100 period following the RCP8.5 scenario from 2006 to 2100".

L364:
- "extreme variability event days" is a confusing term.
- Do the authors really mean "extreme events"?

We have removed all references to extreme events in section 3.3 (except for the introductory sentence to the section)

L360-367: It would be easier to understand if the authors would just refer to increasing variability or variance rather than getting into the messy "extreme" terms, which they use currently. This paragraph only discusses Fig. 7, and that figure only shows the variance. The word "extreme" seems entirely unnecessary throughout this paragraph.

We have removed the references to extreme events.

For the same reasons as mentioned just above, I would also suggest the following:
* L370: change "variability extremes" to "variability"
* L374 and 378: change "extreme variability events" to "variability"
* L377: delete "extremes"

We have removed the references to extreme events.

Also
* L373: change "types of variability" to "time components of variability"
* L377: change "variability types" to "time components of variability"

Changed.

3.4 DRIVERS OF [H+] AND ΩA VARIABILITY CHANGES

L390: delete "northern and southern". As no other "high latitudes" exist other than in the north and south. Thus, these 3 words are unnecessary.

Agreed and deleted.

L394: insert "The" before "GFDL" and insert "model" after "ESM2M"

Inserted.

L396:
* insert "to" after "leads"
* change "very" to "more"

Changed.

L397: "This" what?
We changed the sentence to: "In the low-to-mid latitudes and in particular in the Atlantic Ocean, mean surface A_T is projected to increase (Figure A4) and therefore dampens the overall increase in..".

L403: "that following" is confusing. Changing those 2 words to "increases" would be better if that is what is meant. Unclear.
We replaced "that following" by "the increases"

L407: "would else"? What does that mean?
We rewrote: "The latter contribution is large in the high latitudes, where mean changes alone would lead to a strong increase.".

L409: simplify "a large part" to "much"
Changed.

L410: change "golden" to "gold"
Changed.

Figure 8: Panel (h), the zonal mean plot, is very important but also very busy. The authors try to plot too much information, 8 lines, which is even more confusing because of some overlap (green and gold). I would suggest to make a separate zonal mean figure, where a first panel shows only the first 6 lines, and subsequent panels break down contributions to each term (total, s, sigma, s-sigma, rho) from the different components (CT, AT, T, and S). This would also help the authors to clarify their text on L403-412, which currently (without such added information) seems to suffer from the common a priori concept that only CT matters. If the authors feel that adding a separate zonal mean figure, would result in too many figures, I would suggest that the suggested zonal mean figure should be kept in the main body of the paper and that the maps should then be moved to an Appendix or Supplement.
We followed the referees' recommendation and created separate zonal-mean figures for Figures 8 and 9 (now Figures 9 and 10). The first panel shows the simulated variance change and the contribution of the four components as well as the sum of the four contributions. Panels 2-5 show the contribution to the four components from CT alone, from CT+AT, and from CT, AT, and T. We removed the maps from the manuscript.

Figure 9: Panel (h) the same comments made just above concerning Fig. 8 also apply to Fig. 9. In addition, it is hard to distinguish the 2 blue colored lines, especially because they are generally so close together. Furthermore, it is bothersome that the choice of axis limits cuts off the largest variability seen across large zonal bands.
We readjusted the y-axis limits and changed the colors.

Figure 10: The line color for each of the different components should be made entirely consistent with those used in Figs. 8 and 9. Currently they are not.
We changed the colors in the panels a) of the new zonal-mean versions of Figures 8 & 9 (which are now Figures 9 & 10) so that they agree now with the colors in this figure.

L434: change "similar relative" to "secondary". Otherwise, the authors are imposing on the readers to recall what was said about [H+], which they may not have read or may have forgotten amidst a wealth of other information.

Changed.

DISCUSSION AND CONCLUSIONS

L436: I would recommend to delete "and conclusions" in this section title. By default, the Discussion includes the conclusions if there is no separate Conclusions section. Actually, modern science readers seem to often prefer to have a separate Conclusions section because with more and more papers to read, the tendency is to read only parts of a paper.
We have seen recently published papers in Biogeosciences using 'Discussion and Conclusions'. We therefore keep it as is, but leave it to editor to make the final decision.

L437: Please rephrase without using the confusing term "extreme variability events".
We replaced it by "extreme events".

L439: No, the manuscript does not focus on "extreme events" unlike what the authors state here. Rather it focuses on periods where variability is high. More precisely, after removing the dominant contribution from the change in the mean state it focuses on the resulting upper value (for H+) or the lower value (for OmegaA).
We added that we consider high [H+] and low OmegaA events. Please note that the first two paragraphs of the discussion section have been revised.

L440: suggest change "extreme variability events" to "extremes in variability"
Changed to "extreme events" as in other places in the manuscript.

L442: change "event characteristics" to "variability".
We keep it as is since the characteristics are mentioned in the sentence before and we refer to those.

L443-444: change "extreme variability events" to "variability"
Changed.

L446:
* change "Extreme variability events" to "Extremes in variability". It would be even clearer to replace this sentence with "Variability of OmegaA is projected to decline in the future."
We rewrote this part to: "In contrast to [H+], variability of OmegaA is projected to decline in the future. Therefore, extreme events in OmegaA are projected to become less frequent in the future when defined with respect to a shifting baseline. The reason for the decline in variability is that OmegaA, unlike [H+], becomes less sensitive to variations in the drivers with the mean increase in CT. Furthermore, the projected reductions in the drivers' variabilities, mainly in CT, further reduce OmegaA variability."

* change "it is because" to "The reason is that"
The sentence has been rewritten in response to the comment above.

L449:
- change "significantly add to the reduced occurrence of ΩA variability extremes" to "further reduce ΩA variability"
- To avoid ambiguity, "significantly" should not be used except when discussing statistical results. Even then it should be elaborated upon, with something along the lines of "statistically significant".
Changed as proposed.

L450: use of "extreme variability events" should be avoided. A more precise first sentence would be

"Here we analyzed how extremes driven only by variability change, i.e., after removing the long-term mean."
Changed.

L459: insert "model" before "projects"
Done.

L466: delete "very"
Done.

L469-470: I do not understand this sentence starting with "In other words". Either delete it or be clearer about what is being referring to. The authors have talked about 2 criteria, but it is not stated which criterion each of the numbers mentioned here is referring to.
Deleted.

Figure 13: (caption) delete "temporal"
Done.

L477: It is strange to cite Bednarsek for the meaning of $\Omega A = 1$. The meaning was defined well before that paper.
We agree and now cite Morse and Mackenzie, 1990

L479: change "the ones" to "those"
Done.

L482: insert "the" before "surface"
Done.

L492-493: The authors state the following:
"Here we show that the changes in the seasonal cycle of [H+] translate into large increases in short-term extreme acidity events, at surface as well as at 200 m."
But where do the authors actually show this? From looking at their results, my impression is that the change in the mean state is by far the main reason why the extreme events in [H+] increase, and that the effect from changes in the seasonal cycle is of second order. Please be quantitative.
We now write: "Here we show that the changes in the seasonal cycle of [H+] translate into large increases in short-term extreme acidity events at surface as well as at 200m, when these events are defined with respect to a shifting baseline."

L495: "extreme variability events"? What is meant here? Rephrase.
We now write 'extreme events'.

L496:
* What is meant by "temporal"?
Deleted.

* Can the authors be more precise about what is meant by critical? Have not they shown that most of the variability will be captured with only monthly mean output?
We rewrote the sentence to: "In addition to earlier studies, we also show that changes in subannual variability, which are only partially resolved by monthly mean data, contribute to changes in extreme events in [H+] under increasing atmospheric $CO_2$. Furthermore, show that the average duration of extreme events at the surface and in recent past conditions (1986-2005) is about 15 days. To resolve such events that last for days to weeks, it is critical to use daily mean output."

L502, L503, 521, ...
- replace occurrences of "GFDL ESM2M" with "the GFDL ESM2M model". Some readers will not be modelers. Any added clarity is worthwhile here. Acronyms are confusing and some readers will reach the Discussion before seeing the authors' definition of GFDL ESM2.
Changed throughout the manuscript.

L508, 509, 527, etc: Please rephrase around all occurrences of "extreme variability events"
L508, L509: replaced by 'variability'
L527: replaced by 'extreme events'

L512: Break this sentence into 2 sentences, splitting it just after the 2nd "model".
Changed.

APPENDIX B

L559: I would recommend to avoid footnotes entirely. They are generally frowned upon by journal editors and not even allowed in many journals for good reason.
In this case, we see complications when the footnote is split across pages, half of it appearing under another Appendix, and it does not get the Equations numbers correct. An easy solution is just to move the text and equations that are currently in the footnote to the body of the appropriate Appendix to which the footnote refers to. Adding a separate paragraph as an aside is not a problem.
We moved the footnotes to the text body.

L578: The authors will confuse readers by saying that their Taylor series is 5th order. It includes only 1st derivative and is thus a 1st order Taylor expansion. Saying things like "its Taylor series has five orders" is ambiguous. That should just be deleted. The order of a differential equation has a particular meaning. What the authors mean by order is not carefully defined. Perhaps they can find another term or carefully define what is meant.
As pointed out above, Equation 2 (and also Equation C4) are based on a first order Taylor expansion of H+ with respect to the drivers. However, Equation 2 is also a fifth order polynomial in the sensitivities, standard deviations, and correlation coefficients. We then analyzed the change in H+ variance (represented by Equation 2 and thus based on a first order Taylor expansion) resulting from the changes in the sensitivities, standard deviations, and correlation coefficients of the drivers. To do so, we calculated  the Taylor series of Equation 2 in these variables. This series has five non-vanishing orders. It contains only first derivatives of H+ with respect to the drivers because the derivative itself is treated as a variable (the sensitivity). We revised the paragraph on the decomposition in Appendix C to: "We use Equation C4 and decompose the variability change between the preindustrial and 2081-2100 into the contributions from changes in s, sigma, and rho based on a Taylor expansion. Since [H+] variance represented by Equation C4 is a polynomial of fifth order in these variables, its Taylor series has five non-vanishing orders." and further "Furthermore, it should be noted that the resulting decomposition of [H+] variance change only approximates the simulated variance change because it is based on Equation C4 that itself is based on a first order Taylor expansion of [H+] with respect to the drivers.".

APPENDIX D:

L631: I think that this type of Data availability statement is no longer adequate for the journal Biogeosciences. My recent experience is that some model output will actually need to be made available up front without the requirement to first come back to the authors. The authors might want to check the latest author guidelines and consult with the editor.

We will make all data that are shown in the figures publicly available on ZENODO. Data availability statement has been changed accordingly.

L635: "significantly" is ambiguous.
Deleted.

Table A1: the table title should be placed at the top of the table. It should be short, unlike a figure caption.
We moved the title to the top of the Table and changed it to: "Simulated global ensemble-mean OmegaA extreme event characteristics, when extremes are defined with respect to a shifting baseline. Values in brackets denote ensemble minima and maxima."

Figure A1: "extreme [H+] days" is ambiguous as is "extreme [H+] variability events".

[revised manuscript text omitted]